# The mitochondrial long non-coding RNA *lncMtloop* regulates mitochondrial transcription and suppresses Alzheimer's disease

Wandi Xiong[1,2,3,10], Kaiyu Xu [ID][4,10], Jacquelyne Ka-Li Sun[5,10], Siling Liu[4], Baizhen Zhao[6], Jie Shi[2], Karl Herrup [ID][7], Hei-Man Chow [ID][5✉], Lin Lu [ID][1,2,8✉] & Jiali Li [ID][2,4,6,9✉]

## Abstract

Maintaining mitochondrial homeostasis is crucial for cell survival and organismal health, as evidenced by the links between mitochondrial dysfunction and various diseases, including Alzheimer's disease (AD). Here, we report that *lncMtDloop*, a non-coding RNA of unknown function encoded within the D-loop region of the mitochondrial genome, maintains mitochondrial RNA levels and function with age. *lncMtDloop* expression is decreased in the brains of both human AD patients and 3xTg AD mouse models. Furthermore, *lncMtDloop* binds to mitochondrial transcription factor A (TFAM), facilitates TFAM recruitment to mtDNA promoters, and increases mitochondrial transcription. To allow *lncMtDloop* transport into mitochondria via the PNPASE-dependent trafficking pathway, we fused the 3'UTR localization sequence of mitochondrial ribosomal protein S12 (MRPS12) to its terminal end, generating a specified stem-loop structure. Introducing this allotropic *lncMtDloop* into AD model mice significantly improved mitochondrial function and morphology, and ameliorated AD-like pathology and behavioral deficits of AD model mice. Taken together, these data provide insights into *lncMtDloop* as a regulator of mitochondrial transcription and its contribution to Alzheimer's pathogenesis

**Keywords** Mitochondrial Homeostasis; Alzheimer's Disease; mtDNA; *lncMtDloop*; TFAM
**Subject Categories** Neuroscience; Organelles; RNA Biology

## Introduction

Mitochondria constitute a vital and distinct functional entity within the cells of complex organisms. Traditionally hailed as energy powerhouses, these organelles play a pivotal role in generating over 90% of the cellular ATP, which in turn fuels a plethora of essential cellular processes (Fecher et al, 2019; Reddy et al, 2010). Recent evidence has expanded our understanding, revealing mitochondria's involvement in multifaceted cell signaling cascades. This participation facilitates bidirectional communication between the mitochondrial reticulum and other cell components, thereby regulating pivotal aspects like fuel metabolism, cell cycle progression, developmental processes, and programmed cell death (McBride et al, 2006). The diminishing quality and activity of mitochondria have long been implicated in the natural aging process, as well as the emergence of diverse age-related ailments like Alzheimer's disease (AD). During the aging process, factors such as accumulated reactive oxygen species (ROS), DNA damage, and impaired mitophagy can lead to mitochondrial dysfunction (Balaban et al, 2005; Linnane et al, 1989; Wallace, 1992). While the connection between malfunctions in nuclear genome-encoded genes responsible for mitochondrial function and brain aging is well-established, the potential impact of the intracellular regulatory network on mitochondrial DNA (mtDNA) remains insufficiently characterized (Hirai et al, 2001; Kujoth et al, 2005; Tsai et al, 2022; Wolf, 2021).

The mitochondrial genome contains the blueprints for 13 pivotal proteins that play an indispensable role in ATP production and respiratory functions within the oxidative phosphorylation (OXPHOS) chain. Additionally, this genome comprises two rRNAs and 22 tRNAs that are intricately involved in mitochondrial transcription, replication, and in situ translation. Despite their limited count, these genes hold paramount importance in upholding mitochondrial biogenesis, metabolic processes, and overall homeostasis (Cai et al, 2021; Lane and Martin, 2010; Sabouny and Shutt, 2021). Furthermore, recent discoveries have unveiled several non-coding RNAs (ncRNAs) originating from the mitochondrial genome. These ncRNAs actively engage in intercellular communication between the cytosol and nuclear compartments. For instance, specific DNA sequences complementary to

[1]Peking-Tsinghua Center for Life Sciences, Beijing, China. [2]National Institute on Drug Dependence, Peking University, Beijing, China. [3]Key Laboratory of Tropical Biological Resources of Ministry of Education, School of Pharmaceutical Sciences, Hainan University, Haikou, China. [4]Key Laboratory of Animal Models and Human Disease Mechanisms of Chinese Academy of Sciences & Yunnan Province, Kunming Institute of Zoology, the Chinese Academy of Sciences, Kunming, Yunnan, China. [5]School of Life Sciences, The Chinese University of Hong Kong, Hong Kong, China. [6]JFK Neuroscience Institute, Hackensack Meridian Health JFK University Medical Center, Edison, NJ, USA. [7]Department of Neurobiology, The University of Pittsburgh School of Medicine, Pittsburgh, PA, USA. [8]Institute of Mental Health, National Clinical Research Center for Mental Disorders, Key Laboratory of Mental Health and Peking University Sixth Hospital, Peking University, Beijing, China. [9]Department of Neurology, Hackensack Meridian School of Medicine, Nutley, NJ, USA. [10]These authors contributed equally: Wandi Xiong, Kaiyu Xu, Jacquelyne Ka-Li Sun. ✉E-mail: heimanchow@cuhk.edu.hk; linlu@bjmu.edu.cn; jiali.li@hmhn.org

genes like the mitochondrially encoded NADH: ubiquinone oxidoreductase core subunit 5 (MT-ND5), NADH-ubiquinone oxidoreductase chain 6 (MT-ND6), and cytochrome b are responsible for generating three antisense mitochondrial ncRNAs (ncmtRNAs). The regulation of these ncmtRNAs hinges upon nuclear-encoded proteins (Mercer et al, 2011; Rackham et al, 2011). Conversely, these ncmtRNAs possess the intriguing capacity to reciprocally influence processes beyond the mitochondrial realm. Notably, they can exert an impact on extramitochondrial biological events, including cellular proliferation status (Gao et al, 2017; Villegas et al, 2007), and even extend their influence to tissue-level phenomena such as cardiac remodeling (Kumarswamy et al, 2014).

Mounting evidence underscores the pivotal roles of non-coding RNAs (ncRNAs), particularly long non-coding RNAs (lncRNAs), in governing a diverse array of biological processes within the human brain (Nie et al, 2019). The dysregulation of lncRNAs during brain aging leads to various neurodegenerative disorders, especially AD (Faghihi et al, 2008). Our team previously embarked on characterizing the dynamic expression patterns of all discernible lncRNAs in brain tissue obtained from rhesus macaques, spanning both the developmental and aging stages (Liu et al, 2017). From there, the lncRNA AC027613.1, originating from the mtDNA D-loop, emerged as a previously undiscovered, evolutionarily conserved entity with age-associated characteristics (Liu et al, 2017).

In this study, we have rechristened AC027613.1 as "lncMtDloop" and illuminated its substantial role in orchestrating TFAM-dependent mtDNA transcription and overall mitochondrial equilibrium. Furthermore, we delve into a comprehensive understanding how disrupted homeostasis of a mitochondrial-derived lncRNA—lncMtDloop, disrupts the normal functioning of mitochondria in neurons and even contributes to Alzheimer's pathogenesis.

# Results

## LncMtDloop is a mitochondrial lncRNA with a conserved noncoding nature

In our previous report, cluster analysis revealed that AC027613.1 was one of the newly identified lncRNAs in the rhesus macaque brain, with important relevance in both brain development and aging (Liu et al, 2017). In this study, the extended characterization of the AC027613.1 gene sequence reveals that it is located within the D-loop region of the mitochondrial genome (Fig. 1A), and therefore we renamed the gene as lncMtDloop. By the RNAscope in situ hybridization (RISH), specific probes set against the mouse lncMtDloop were compared against the mitochondrial ATP synthase F1 subunit alpha (Atp5a) signals obtained in immunocytofluorescence (ICF) assays. This comparison revealed that the lncMtDloop is predominantly mitochondrially localized in primary hippocampal neurons (Fig. 1B,C). Such localization was indeed replicated even in cells of different mammary species (i.e., monkey Cos-7, human SH-SY5Y, and mouse N2a cells) (Fig. 1D–G), hinting that the lncMtDloop is likely conserved during the mitochondrial evolution.

Next, through a thorough BLAST search and mapping of this sequence with the NONCODE and UCSC databases, we have conclusively affirmed that AC027613.1 originates from the mitochondrial DNA (mtDNA), specifically located at chrM: 15,356–16,294 bp (Fig. EV1A). While analysis of the various lncMtDloop gene sequences indicated a slight size variance among the human (572 bp), macaque (695 bp), and mouse (939 bp) homologs, the key block sequences are still highly conserved among various species (Fig. EV1B). To confirm that the lncMtDloop is a true, novel mitochondrial lncRNA species without protein-coding potential, the Phylogenetic Codon Substitution Frequencies (PhyloCSF) algorithm, which allows comparative genomics analysis was performed on the UCSC Genome Browser (Fig. EV1C) (Mudge et al, 2019). The corresponding coding potential scores (CPC) and coding potential assessment tool (CPAT) coding probabilities (Fig. EV1D,E) were both at minimum as compared to a set of positive-control reference coding transcripts, confirming that the lncMtDloop has no protein-coding potentials. Since the function of a wide range of existing lncRNAs is largely dependent on their secondary structures (Bugnon et al, 2022), based on the minimum free energy (MFE) model, lncMtDloop is predicted to form a complex double-stranded stem-loop (Fig. EV1F), a common structure among well-documented lncRNAs (Yan et al, 2016).

## Trafficking of allotropic lncMtDloop into mitochondria

To date, no established methodologies, including CRISPR/Cas9-based techniques and conventional siRNA technology, have proven effective in targeting mtDNA sequences, mainly due to the inherent inefficacy of nucleic acid import into mitochondria. To address this challenge and facilitate gain-of-function or loss-of-function investigations within the mitochondria, we have devised an innovative expression construct tailored for lncMtDloop.

In the context of allotropic expression, the genes encoded by wild-type mtDNA are transcribed within the nucleus. Consequently, resulting non-coding RNAs (ncRNAs) or proteins can be transported into the mitochondria, thereby salvaging mitochondrial functionality. This transportation is facilitated by mitochondrial targeting sequences (MTSs) and 3′ untranslated regions (3′UTRs), which have been harnessed to guide the influx of proteins and RNAs into the mitochondria (Chin et al, 2018; Wang et al, 2012b). Certain specific 3′UTRs exhibit the capability to guide RNA molecules to the external surface of the mitochondria. An illuminating example is the mRNA encoding human mitochondrial ribosomal protein S12 (MRPS12), which originates from the nucleus and bears regulatory sequences within its 3′UTR that facilitate localization to the mitochondrial surface. To ascertain whether the 3′UTR of MRPS12 could effectively guide the localization of allotropic lncMtDloop to the mitochondrial surface, we strategically fused the 3′UTR localization sequence of MRPS12 to the terminal end of lncMtDloop (Fig. 2A). The RISH-ICH assay revealed that this allotropic expression of lncMtDloop, accompanied by the inclusion of MRPS12's 3′UTR, indeed resulted in a heightened localization of lncMtDloop to the mitochondrial surface (Appendix Fig. S1). Moreover, the incorporation of the 3′UTR element from MRPS12 led to a greater enrichment of lncMtDloop within the mitochondrial fraction (Fig. 2B). These results indicate that this construct exhibits an allosteric expression pattern, effectively reinstating the expression levels of lncMtDloop and thus enabling functional interventions.

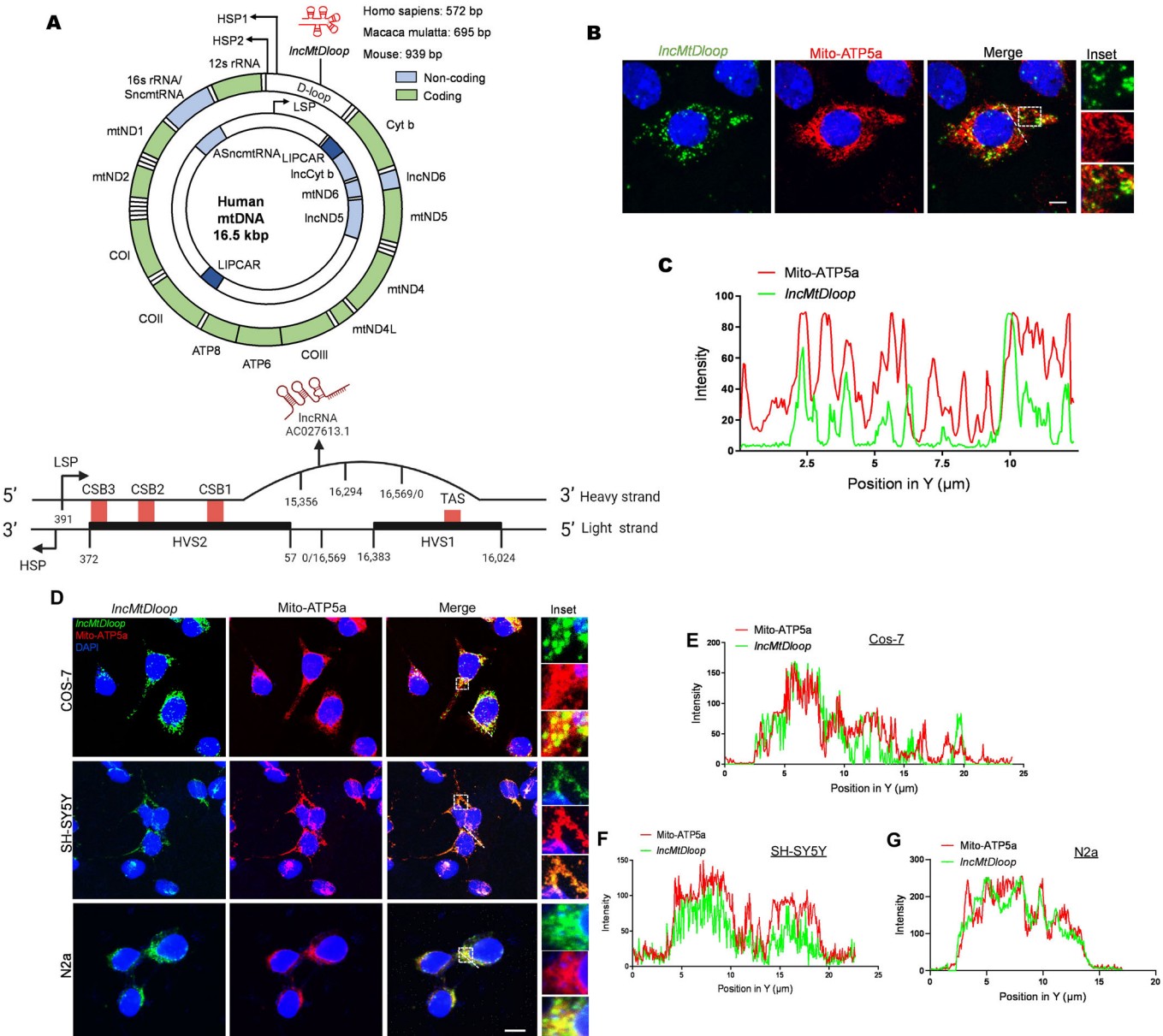

**Figure 1. *LncMtDloop* is characterized as a conserved long noncoding RNA derived from the mitochondrial DNA D-loop.**

(A) Schematic illustration depicting the intricate structure of the non-coding region within the mitochondrial genome known as the D-loop and the transcripts of the *lncMtDloop* in humans, macaques, and mice. The two strands of mtDNA are denoted as heavy and light strands. And *lncMtDloop* (AC0276613.1) is originated from the heavy strand of the mitochondrion. D-loop displacement loop, HSP heavy strand promoter, LSP light strand promoter. (B) Representative images of RNAscope ISH (in situ hybridization) coupled with immunofluorescence, highlighting the cellular localization of *lncMtDloop* and mitochondria within neurons. The scale bars represent a length of 5 μm. (C) Quantification of the relative fluorescence intensities of *lncMtDloop* and mito-ATP5a signals, as indicated by the white dotted line in (B). The analysis was performed using Image J software. (D) Representative images depicting the subcellular localization of *lncMtDloop* (green) and mitochondria (red) using RNAscope ISH and immunostaining. Images are shown for various cell lines: N2a (mouse), SH-SY5Y (human), and Cos-7 (monkey). Scale bars, 10 μm. (E–G) Quantification of fluorescence signal intensities for *lncMtDloop* and mitochondrial markers along the white dotted line in panel (D) was performed using Image J software. Source data are available online for this figure.

Although with the element of 3′UTR of *MRPS12* allows more allotropic *lncMtDloop* to attach to mitochondria, there is still free recombinant *lncMtDloop* distributed in the cytosol. Subsequent to conducting enrichment analyses on candidate proteins within both the cytosolic and mitochondrial compartments, a multitude of functional pathways emerged, primarily linked to protein

localization and transportation. Within this array of proteins, we undertook an expansive protein–RNA association prediction, evaluating the interaction tendencies between polypeptide and nucleotide chains through the implementation of the catRAPID algorithm. In the cytosolic fractions, our scrutiny identified the Aly/ REF export factor (Alyref), tubulin polymerization-promoting

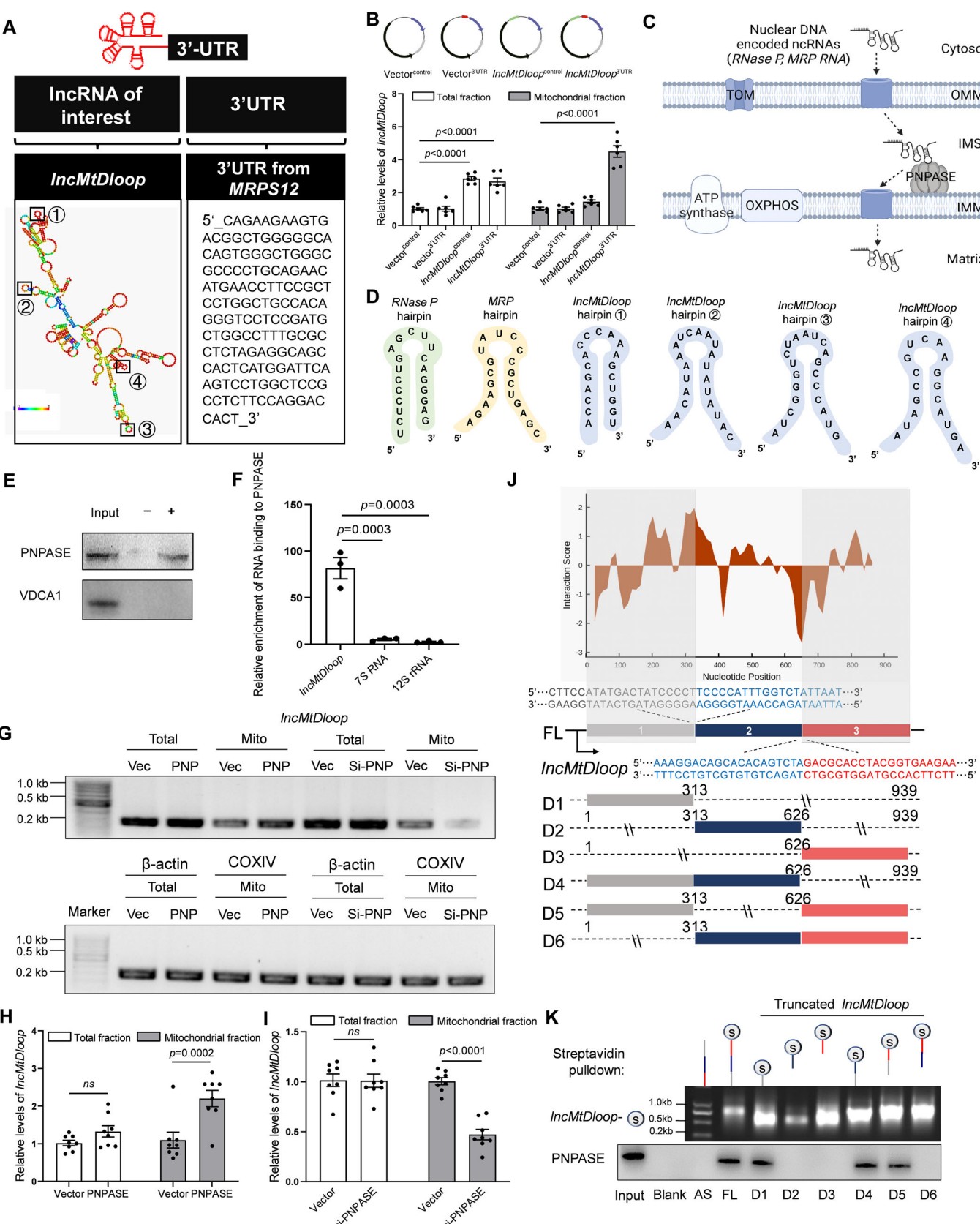

**Figure 2.   PNPASE binds to allotropic *lncMtDloop* and facilitates its mitochondrial import.**

(A) Illustration of the expression construct structure featuring the addition of the 3'UTR localization sequence of *MRPS12* to the 3' end of *lncMtDloop*. (B) Upper: Plasmid expression constructs. Lower: RT-qPCR analysis of alternate *lncMtDloop* expression upon treatment with various plasmids in both total and mitochondrial fractions. Error bars indicate mean ± SEM, with $n = 6$ repetitions per group, one-way ANOVA followed by Dunnett's multiple comparisons test. (C) PNPASE's role in governing the import of nucleus-encoded *RP* and *MRP* RNAs into the mitochondrial matrix through binding to specific stem-loop structures in the RNAs. (D) Four minimized RNA stem-loop structures within *lncMtDloop* are potentially capable of directing *lncMtDloop* into the mitochondria. (E) RNA pull-down assay utilizing biotinylated *lncMtDloop* and antisense-*lncMtDloop*, followed by Western blot analysis of PNPASE. The voltage-dependent anion-selective channel protein 1 (VDAC1) was used as mitochondrial quality control. "+": *lncMtDloop*; "−": anti-*lncMtDloop*. (F) RT-qPCR assessment of detectable *lncMtDloop* from RIP analysis using PNPASE antibody. 7S RNA and 12S rRNA was used as control. Results were normalized to the control IgG RIP group. Error bars represent mean ± SEM, with $n = 3$ independent experiments per group, unpaired *t*-test. (G) RT-qPCR was conducted with primers amplifying *lncMtDloop* (170 bp), *β-actin* (250 bp), and *COX IV* (184 bp). RNA was isolated from total cell lysate or mitochondrial lysate of N2a cells expressing PNPASE (PNP), PNPASE siRNA (si-PNP), and an empty vector (Vec), which were pre-transfected with the expression construct including the 3'UTR of *MRPS12* appended to *lncMtDloop*'s 3' end. (H) RT-qPCR analysis of *lncMtDloop* expression. RNA was isolated from total cell lysate or mitochondrial lysate of N2a cells expressing PNPASE (PNP) and an empty vector (Vec), pre-transfected with the expression construct appended with the 3'UTR of *MRPS12* to *lncMtDloop*'s 3' end. Error bars denote mean ± SEM, with $n = 8$ repetitions per group, unpaired *t*-test. (I) RT-qPCR analysis of *lncMtDloop* expression. RNA was isolated from total cell lysate or mitochondrial lysate of N2a cells expressing PNPASE siRNA (si-PNP) or an empty vector (Vec), pre-transfected with the expression construct appended with the 3'UTR of *MRPS12* to *lncMtDloop*'s 3' end. Error bars represent mean ± SEM, with $n = 8$ repetitions per group, unpaired *t*-test. (J) Heatmap illustrating the interaction propensity of *lncMtDloop* binding to PNPASE, as predicted by catRAPID. The X-axis represents the RNA nucleotide sequence location, while the Y-axis represents the protein residue location. Red shades in the color scale bars indicate predictions with high interaction propensity. At the bottom, a schematic map outlines the construction of six truncated *lncMtDloop* mutants intended to identify its protein-binding motifs. (K) The indicated six *lncMtDloop* truncates were transcribed in vitro and tagged with RNA streptavidin for RNA pull-down assays. Source data are available online for this figure.

protein (Tppp), and p32 as the foremost interacting candidates (Fig. EV2A). To ascertain the validity of these interactions, we proceeded with western blot and confirmed the direct interaction between *lncMtDloop* and all of these candidates, (Fig. EV2C). Notably, a selection of proteins with relevance to protein localization, encompassing syntaxin 1B (Stx1b), Atp5d, and p32, ranked prominently in terms of potential interaction scores and their distinguishing capabilities with *lncMtDloop* within the mitochondrial fraction. Western blots were performed and verified the physical interactions between *lncMtDloop* and the aforementioned proteins with the exception of Stx1b (Fig. EV2B,D).

p32, a well-conserved eukaryotic protein that predominantly localizes in mitochondria (Barna et al, 2019; Gotoh et al, 2018), serves as a recognized RNA-binding protein within the mitochondrial context. Its functions extend to influencing the regulation of OXPHOS, orchestrating communication between the nucleus and mitochondria, and participating in other mitochondrial activities (Jiang et al, 1999; Leucci et al, 2016; Yagi et al, 2012). As illustrated, p32 was detected from both fractionation experiments (Fig. EV2C,D) with similar localization patterns as that of the *lncMtDloop* (Fig. EV2E,F). Furthermore, employing a combination of RNA immunoprecipitation (RIP) assay and qPCR analyses with the anti-p32 antibody verified the reverse binding interaction between *lncMtDloop* and p32 protein (Fig. EV2G). Given the strong binding of in vitro-transcribed-*lncMtDloop* with p32 protein in both cytosolic and mitochondrial fractions, it is plausible that endogenous p32 could potentially facilitate the transportation of ectopically expressed *lncMtDloop* into mitochondria. To explore this possibility, experiments were conducted in N2a cell lines pre-infected with lentiviral particles (LV) to overexpress *lncMtDloop*. Remarkably, knocking down p32 expression effectively counteracted the ectopic expression effects, resulting in significant reductions of *lncMtDloop* within mitochondria (Fig. EV2H–J). Importantly, this effect was distinct from p32, as similar knockdowns of other potential binding partners, such as ATP5d and Stx1b did not produce the same outcome (Fig. EV2K). The effect of p32 on *lncMtDloop* mitochondrial localization was further verified by the RISH-ICF

assay (Fig. EV2L,M), underscoring the critical nature of the interaction between the two in governing the mitochondrial localization of *lncMtDloop*. Next, through the utilization of catRAPID, an algorithm designed to predict the binding affinity of protein–RNA pairs (Agostini et al, 2013), critical regions within the 5′ and 3′ terminal domains of *lncMtDloop* were identified for potential protein–RNA interactions (Fig. EV2N). Next, a series of truncated *lncMtDloop* constructs were synthesized, encompassing *lncMtDloop*-FL (full length), *lncMtDloop*-D1 to D6, which were subsequently employed as bait in RNA pull down assays. Of note, the 3′-terminal fragment spanning 637–939 nt exhibited robust interactions with the p32 protein, akin to the positive control, i.e., full-length *lncMtDloop* (Fig. EV2N,O). Nonetheless, the mechanisms underlying the import of *lncMtDloop* into mitochondria remain shrouded in ambiguity.

In certain instances, select small ncRNAs, including *RNase P* RNA and *MRP* RNA, have been shown to rely on a mitochondrial intermembrane space-located RNA import regulator known as polynucleotide phosphorylase (PNPASE) (Wang et al, 2010; Wang et al, 2012a). PNPASE orchestrates the import of specific nucleus-encoded RNAs into the mitochondria through its recognition of distinct RNA stem-loop structures (Fig. 2C). Indeed, within *lncMtDloop*, four ~20 ribonucleotide stem-loop structures have been identified. Notably, these structures bear resemblance to the predicted stem-loop structures found in *RNase P* and *MRP*, albeit with differing sequences (Fig. 2D), suggesting the potential role of PNPASE in facilitating the transport of *lncMtDloop* into the mitochondrial matrix. Using RNA pull-down and RNA immunoprecipitation (RIP), we verified the interactive binding between *lncMtDloop* and PNPASE (Fig. 2E,F). Of note, the augmentation of PNPASE expression (PNP) led to a twofold increase in the import of allotropic *lncMtDloop* into mitochondria, a significant elevation when compared to the empty vector control (Vec) (Fig. 2G,H). Interestingly, in the aging mouse brain, PNPASE exhibited a consistent level of abundance (Appendix Fig. S2A). Nevertheless, when the expression of PNPASE was suppressed (si-PNP), the augmented import of allotropic *lncMtDloop* into mitochondria was effectively hindered (Fig. 2G,I; Appendix Fig. S2B). Moreover,

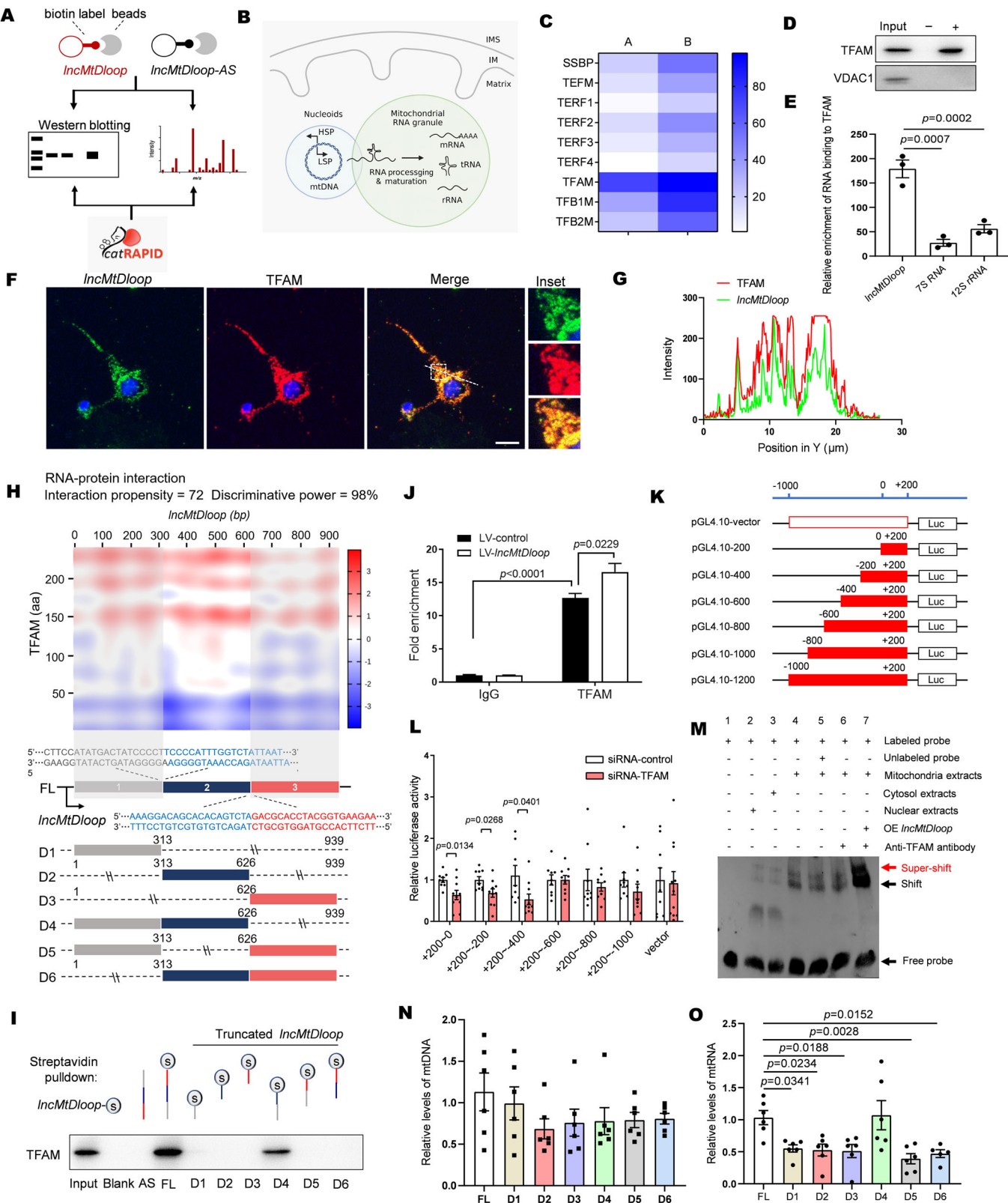

**Figure 3.  *LncMtDloop* is required for the recruitment of TFAM and mitochondrial transcription initiation.**

(A) Schematic illustrating the procedure of *lncMtDloop* RNA pull-down and mass spectrum (MS). (B) Schematic representation of mtDNA transcription, RNA processing, and maturation occurring between nucleoids and mitochondrial RNA granules. IMS stands for the mitochondrial intermembrane space, while IM refers to the inner mitochondrial membrane. (C) Prediction of *lncMtDloop*'s interaction propensity with potential binding proteins in the mitochondrial nucleoid using catRAPID. The left row displays interaction propensity scores, while the right row indicates the proteins' discriminative power in a centesimal system. Notable proteins include SSBP (mitochondrial single-stranded DNA-binding protein), TEFM (mitochondrial elongator factor), TERF1-4 (mitochondrial transcription termination factors 1–4), TFAM (mitochondrial transcription factor A), and TFB1M/TFB2M (mitochondrial transcription factors B1/B2). (D) Utilization of RNA pull-down with biotinylated *lncMtDloop* and antisense-*lncMtDloop* in mitochondrial preparations, followed by Western blotting to detect TFAM. VDAC1 was used as mitochondrial quality control. "+": *lncMtDloop*; "−": *anti-lncMtDloop*. (E) RT-qPCR to detect *lncMtDloop* from RIP analysis using a TFAM antibody. 7S RNA and 12S rRNA was used as control. The results were normalized to the control IgG RIP group. Error bars indicate mean ± SEM, with n = 3 independent experiments per group, unpaired *t*-test. (F) Representative images displaying RNAscope ISH combined with immunostaining, highlighting the co-localization of *lncMtDloop* (green) and TFAM (red) in neurons. Scale bars represent 10 µm. (G) Quantification of relative fluorescence signals of *lncMtDloop* and TFAM, as depicted along a white dotted line in panel (F), using Image J software. (H) Heatmap illustrating the interaction propensity of *lncMtDloop*'s binding to TFAM, as predicted by catRAPID. The X-axis represents the location along the RNA nucleotide sequence, while the Y-axis corresponds to the protein residue location. Red shades in the color scale bars indicate predictions with high interaction propensity. At the bottom, a schematic map outlines the construction of six truncated *lncMtDloop* mutants aimed at identifying its protein-binding motifs. (I) The indicated six *lncMtDloop* truncates were transcribed in vitro and tagged with RNA streptavidin for subsequent RNA pull-down assays. (J) Confirmation of TFAM binding to the mitochondrial promoter within the D-loop region using ChIP analysis. Error bars represent mean ± SEM, with n = 3 independent experiments per group, unpaired *t*-test. (K, L) Incorporation of six distinct loci from the mitochondrial promoter within D-loop regions into the pGL4.10 vector, followed by luciferase reporter assays in TFAM-silenced N2a cells. Error bars denote mean ± SEM, with n = 8-10 repetitions per group, unpaired *t*-test. (M) EMSA assay demonstrating the direct binding of TFAM to the promoter within the mitochondrial D-loop region. (N, O) RT-qPCR results reflecting mtDNA and mtRNA levels upon treatment with LV-*lncMtDloop* in primary cultured neurons. Error bars signify mean ± SEM, with n = 5–6 repetitions per group, one-way ANOVA followed by Dunnett's multiple comparisons test. Source data are available online for this figure.

insights gleaned from the catRAPID algorithm highlighted the potential interactions between PNPASE and several fragments of allotropic *lncMtDloop* (Fig. 2J). Leveraging this prediction, a series of truncated *lncMtDloop* constructs—namely *lncMtDloop*-FL (full length), *lncMtDloop*-D1 to D6—were synthesized for employment as baits in RNA pull down assays (Fig. 2J). Notably, the binding propensity between PNPASE and the 5′ terminal 1–313 nt fragment, which encompasses the stem-loop structures similarly predicted with *RNase P* and *MRP*, was identified (Fig. 2K). These findings propose the involvement of the *MRPS12* 3′UTR in steering *lncMtDloop* toward the vicinity of mitochondria. The RNA import sequence within the stem-loop at the 5′ terminal of *lncMtDloop* appears to enable the PNPASE-dependent import of allotropic *lncMtDloop* into mitochondria (Appendix Fig. S2C).

### *LncMtDloop* is required for TFAM recruitment to initiate mitochondrial gene transcription

To explore the potential biological partners of *lncMtDloop*, we performed an RNA pull-down. With both cytosolic and mitochondrial fractions enriched from brain tissue lysates of WT mice, full-length sense or antisense of in vitro-transcribed *lncMtDloop* sequences conjugated with biotin probes were used as baits to capture any potential binding partners (Fig. 3A). In contrary to the antisense bait, full-length sense *lncMtDloop* bait was able to pull down a group of interacting proteins and their identities were further revealed by the silver staining-mass spectrometry (SS-MS) analysis (Appendix Fig. S3A). Overall, we obtained a group of proteins that were potentially associated with *lncMtDloop*. Then, Metascape was used to pick out essential biological processes and functional pathways of the candidate proteins through Gene Ontology (GO) and Kyoto Encyclopedia of Genes and Genomes (KEGG) enrichment analysis. In cytosolic fractions, the candidate proteins are mainly involved in the mRNA metabolic process and biosynthetic process (Appendix Fig. S3B). In mitochondrial fraction, the candidate proteins were mainly enriched in pathways in metabolism, cellular respiration, mitochondrial transcription, and protein localization (Appendix Fig. S3C).

The mitochondrial genome homeostasis is essential for mitochondrial function, and its deleterious mutations are implicated in metabolic disorders, aging, and neurodegenerative disorders (Antonyova et al, 2020; Sanchez-Contreras et al, 2023). The mitochondrial D-loop region is essential for mtDNA replication and transcription. Hence, the lncRNA *lncMtDloop* transcribed from this region is highly like to be responsible for the regulation of the mitochondrial genome. Hinted from the SS-MS analysis results with the mitochondrial fraction, *lncMtDloop* may potentially interact with the mitochondrial transcription factors located within the transcription complex as well, serving as a guide to regulate mitochondrial genome homeostasis and function. The mtDNA is usually packaged in DNA-protein complexes named nucleoids, which contain various proteins required for mtDNA transcription, replication, translation, and degradation (Long et al, 2021). The mtDNA transcripts undergo processing and maturation in the mitochondrial RNA granule (Fig. 3B) (Xavier and Martinou, 2021). With all existing and known mtDNA transcription factors, prediction using the catRAPID algorithm was performed for potential *lncMtDloop* interactions based on the physical-chemical properties between the binding partners (Agostini et al, 2013; Bellucci et al, 2011; Livi et al, 2016). From there, TFAM, an mtDNA promoter-binding protein revealed the highest interaction scores whereas other proteins identified had lower catRAPID scores (Fig. 3C), and further experimental analyses with western blot, RIP, and RISH-ICF confirmed such prediction (Fig. 3D–G). Combining the catRAPID algorithm and RNA pull-down of truncated *lncMtDloop*, the *lncMtDloop*-D4 region was again validated to be important for the TFAM interaction (Fig. 3H,I).

TFAM is essential for mtDNA genome maintenance and initiates mitochondrial transcription by recruiting mitochondrial RNA polymerase (POLRMT) and mitochondrial transcription factor B (TFBM) to specific promoter regions on mtDNA (Hillen et al, 2017; Hillen et al, 2018). Therefore, to explore whether *lncMtDloop* is helpful in the recruitment of TFAM to the promoter locus of mtDNA, we performed the chromatin immunoprecipitation (ChIP) assay for the analysis of mitochondrial genome remodeling. Consistently, overexpression of *lncMtDloop* did

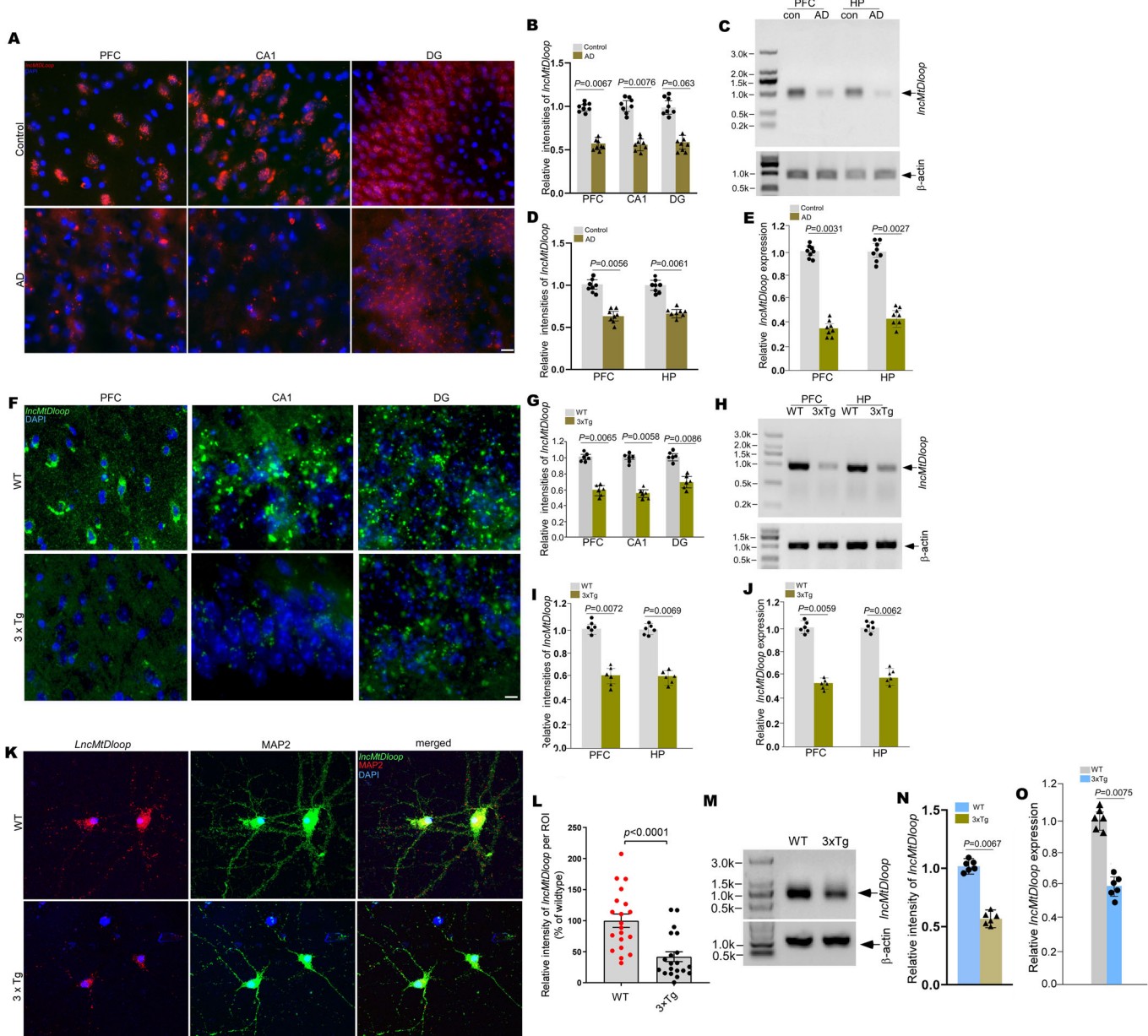

enhance the binding ability between TFAM and the D-loop locus of mtDNA (Fig. 3J). Subsequent refined analyses with luciferase reporter assays under the condition of knocking down TFAM for the identification of TFAM-mtDNA interacting region revealed that the mtDNA segment at $+200 \sim -400$ bp within the D-loop region was critical and sufficient (Fig. 3K,L). To further examine whether *lncMtDloop* truly promotes the recruitment of TFAM to mtDNA promoter, the electrophoretic mobility shift assay (EMSA) was performed. Based on previous protein structural studies (Hillen et al, 2017; Hillen et al, 2018), biotinylated probes against TFAM-mtDNA binding pocket were designed and incubated with nuclear, cytosol, and mitochondrial extracts harvested from primary cultured neurons. While in both cytosolic and nuclear fractions, little signals resulting from size shifting of probes were found,

strong band shifting signals were, however, found in the mitochondrial extract reaction. Such a phenomenon could be competitively disrupted by an unlabeled probe or reversibly enhanced when *lncMtDloop* was overexpressed (Fig. 3M; Appendix Fig. S4). Functionally, *lncMtDloop*-FL and *lncMtDloop*-D4, which constitute the interacting site for TFAM significantly promoted mitochondrial genome transcription but not its replication (Fig. 3N,O). To investigate the regulatory role of *lncMtDloop* in mtDNA transcription, N2a cells were treated with ethidium bromide (EB) for 8 weeks, a standard method for inducing mtDNA depletion. Following EB treatment, both mtDNA and mtRNA copy numbers significantly decreased (Fig. EV3A,B), along with reduced levels of MitoTracker signaling (Fig. EV3C,D) and mitochondrial complex (OXPHOS subunit proteins) (Fig. EV3E,F). Interestingly,

**Figure 4.   Loss of *lncMtDloop* expression in neurons of Alzheimer's disease.**

(A) Illustrative images from RNAscope ISH, demonstrating *lncMtDloop* expression. RNAscope ISH was performed on 15 μm paraffin sections of the prefrontal cortex (PFC) and hippocampus (CA1 and DG) obtained from age- and sex-matched control and AD individuals using human-specific probe sets targeting *lncMtDloop*. Scale bars, 10 μm. (B) Quantification of the relative RNAscope fluorescence intensity shown in A), performed using Image J software. Error bars indicate mean ± SEM, with *n* = 8 patients per group, unpaired *t*-test. (C) Northern blot images displaying *lncMtDloop* transcript in postmortem frozen PFC and hippocampal tissues from control and AD individuals. (D) Quantification of northern blot signal intensities shown in (C), analyzed using Image J. Error bars denote mean ± SEM, with *n* = 8 individuals. *P < 0.05, unpaired *t*-test. (E) RT-qPCR was conducted to validate the *lncMtDloop* expression. Total RNAs were extracted from postmortem frozen PFC and hippocampal tissues of age- and sex-matched control and AD individuals. Error bars represent mean ± SEM, with *n* = 8 patients per group, unpaired *t*-test. (F) Representative images illustrating RNAscope ISH results for *lncMtDloop* expression. RNAscope ISH was performed on 10 μm cryostat brain sections from 12-month-old wild type (WT) and 3xTg mice, using mouse-specific probe sets targeting *lncMtDloop*. Scale bars, 15 μm. (G) Quantification of RNAscope fluorescence intensity shown in (F), analyzed using Image J software. Error bars indicate mean ± SEM, with *n* = 6 mice per genotype, unpaired *t*-test. (H) Northern blot images of *lncMtDloop* transcript in fresh PFC and hippocampal tissues from wild type and 3xTg mice. (I) Quantification of northern blot signal intensities shown in (H), assessed using Image J. Error bars denote mean ± SEM, with *n* = 6 mice per genotype, unpaired *t*-test. (J) RT-qPCR validation of *lncMtDloop* expression. Total RNAs were extracted from fresh PFC and hippocampal tissues of 12-month-old wild-type and 3xTg mice. Error bars represent mean ± SEM, with *n* = 6 mice per group, unpaired *t*-test. (K) Representative images from RNAscope ISH depicting *lncMtDloop (Red)* expression in wild-type (WT) and 3 × Tg primary hippocampal neurons (MAP2, Green) at DIV14, using mouse-specific probe sets. Scale bars, 10 μm. (L) Quantification of *lncMtDloop* fluorescent intensity shown in (K), measured using Image J software. Error bars indicate mean ± SEM, with *n* = 20 images per group, unpaired *t*-test. (M) Northern blot images displaying *lncMtDloop* transcript in primary hippocampal neurons of wild type and 3 × Tg at DIV14. (N) Quantification of northern blot signal intensities shown in (M), analyzed using Image J Error bars denote mean ± SEM, with *n* = 6 per group, unpaired *t*-test. (O) RT-qPCR analysis of *lncMtDloop* expression in hippocampal neurons of wild type and 3 × Tg at DIV14. Error bars represent mean ± SEM, with *n* = 6 per group, unpaired *t*-test. Source data are available online for this figure.

ectopic expression of *lncMtDloop* notably restored mtRNA levels without affecting mtDNA copy number and further boosted OXPHOS subunit protein levels. These findings suggest that l*ncMtDloop* enhances the affinity and efficacy of TFAM interaction with the mtDNA promoter, thereby promoting transcription (Appendix Fig. S3D).

## Loss of *lncMtDloop* found in the brains of human AD and the mouse model

Mitochondria exhibiting diminished mtDNA content may engage in genetic compensation or continue to merge with other mitochondria as compensatory mechanisms to uphold neuronal survival and functionality (Dooley et al, 2019; Swaegers et al, 2020; Youle and van der Bliek, 2012). Nevertheless, persistent and prolonged exposure to additional stressors, such as pathologies linked to AD, can exacerbate these deficiencies or intensify loss-of-function mutations in mitochondrial genes, consequently deteriorating mitochondrial function. Thus, our inquiry delved into the potential involvement of *lncMtDloop* in the pathogenesis of Alzheimer's. Our observation encompassed the assessment of human brain tissue, revealing a noteworthy reduction in *lncMtDloop* expression within the prefrontal cortex (PFC), as well as the hippocampal CA1, and dentate gyrus (DG) regions among AD patients, in comparison to age- and gender-matched non-dementia controls (Figs. 4A,B, EV4A; Appendix Table S1). This observation was further substantiated through northern blotting and RT-qPCR utilizing the same collection of human brain tissue samples (Fig. 4C–E). Analogous findings were replicated within aged (12-month-old) 3xTg mice—a triple transgenic AD animal model characterized by classic Aβ pathology and tauopathies—employing identical analytical methodologies (Fig. 4F–J). Notably, *lncMtDloop* expression in the whole brain of 3xTg mice decreased during aging, with a marked decline beginning at 9–12 months (Fig. EV4B; Appendix Fig. S5A–C). In contrast, mtRNA copy numbers showed a delayed reduction, starting at 12 months (Fig. EV4C). Next, we investigated whether alterations in the expression of other mitochondrial noncoding RNAs occur in 3xTg mouse brains. Notably, *lncND5* displayed reduced expression in the

brains of 3xTg mice (Appendix Fig. S5D–F). Interestingly, 7S RNA, a polyadenylated non-coding RNA transcribed from a region immediately downstream of the mitochondrial light strand promoter, exhibited significant accumulation in AD brains, while 12S rRNA expression showed minimal changes (Appendix Fig. S5G–I).

Next, given that *lncMtDloop* was initially identified from brain tissue, it was logical to investigate its cell-specific expression patterns across various brain cell types. Using the RISH-ICF assay, *lncMtDloop* was prominently detected in primary mouse neurons, microglia, and astrocytes. Notably, the reductions in *lncMtDloop* observed within brain tissue were specific to neurons and not present in glial cells (microglia and astrocytes), as confirmed through multiple assay approaches (Figs. 4K–O and EV4D–I).

## Allotropic *lncMtDloop* expression reverses loss of mitochondrial homeostasis in neurons of 3xTg mice

Dysregulated mitochondrial biogenesis, alterations in mitochondrial dynamics (fusion and fission), and compromised genome integrity mechanisms collectively contribute to the breakdown of mitochondrial health and function. These factors are intricately linked to various disorders, notably AD (Baloyannis et al, 2004; Zhunina et al, 2020). To explore whether the loss of *lncMtDloop* is connected to decreased mtDNA content and reduced mtRNA expression, we initially examined age-related alterations in mtDNA content and mtRNA expression within the hippocampal and prefrontal cortex (PFC) tissues of both rhesus macaques and mice across different age groups. While minimal changes were observed in mtRNA expression during normal aging (Fig. 5A,B), diminished mtDNA transcripts were evident in 12-month-old 3xTg mice, which experience age-associated changes (Fig. 5C).

Consequently, we delved deeper into the relationship between *lncMtDloop* and mtDNA levels. To explore the potential regulatory role of *lncMtDloop* in TFAM (mitochondrial transcription factor A) and its influence on mtDNA content and mtRNA expression, we executed *lncMtDloop* restoration through allotropic expression, including the incorporation of 3′UTR of *MRPS12*, in neurons affected by AD. Notably, this intervention resulted in a significant

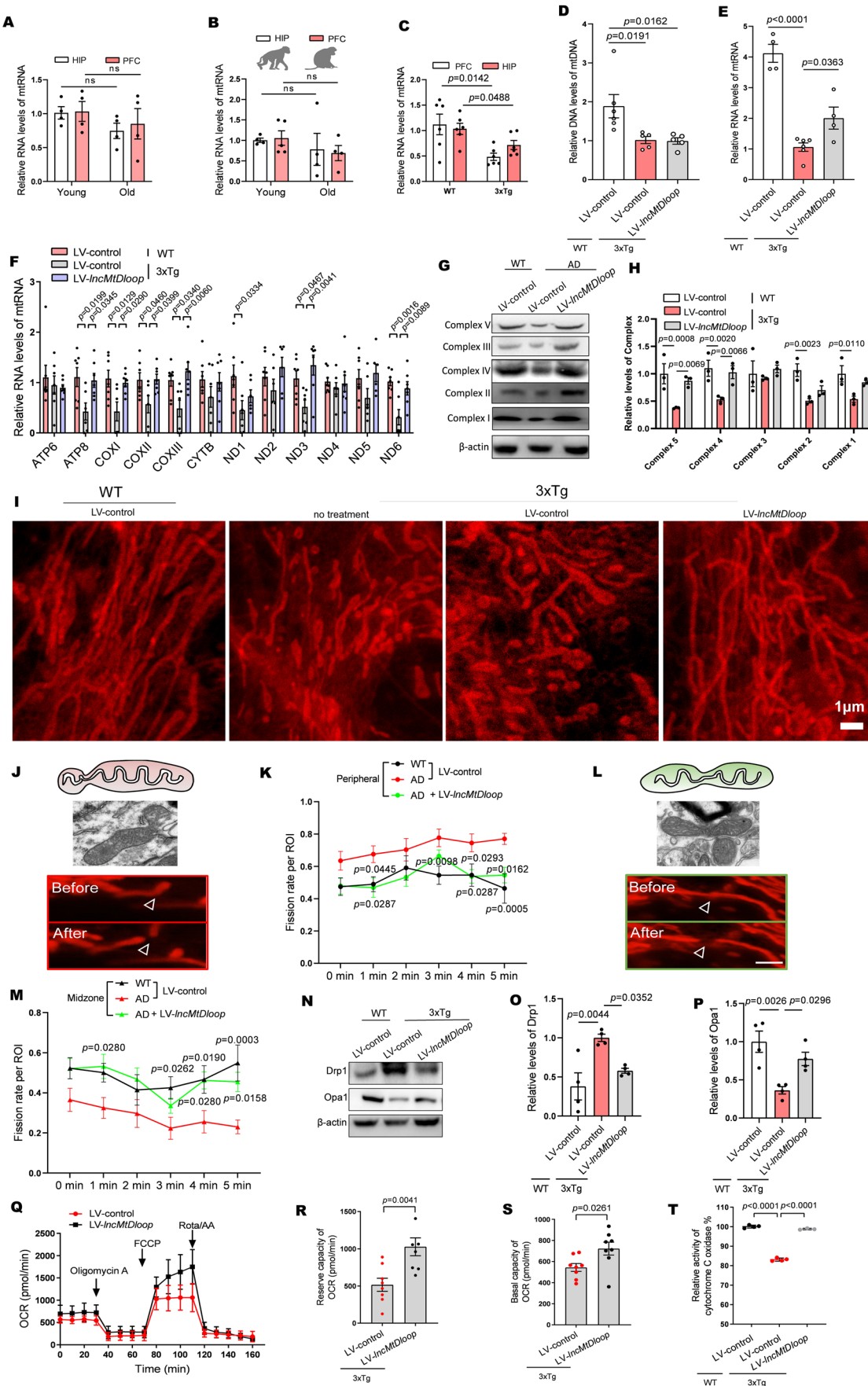

**Figure 5.   Allotropic *lncMtDloop* expression restores mitochondrial homeostasis in AD neurons.**

(A) RT-qPCR results illustrating mtRNA levels in the hippocampus and prefrontal cortex (PFC) of young (4 to 6 years) and old (16 to 20 years) *Macaca mulatta*. Error bars indicate mean ± SEM, with $n = 4$–5 per group, n.s. no significance, unpaired *t*-test. (B) RT-qPCR results depicting mtRNA levels in the hippocampus and PFC of young (3 months) and old wild-type mice (15 months). Error bars denote mean ± SEM, with $n = 4$ per group, n.s. no significance, unpaired *t*-test. (C) RT-qPCR analysis confirming RNA levels of mtRNA. Total RNAs were extracted from fresh hippocampal and PFC tissues of 12-month-old wild-type (WT) and 3xTg mice. Error bars represent mean ± SEM, with $n = 6$ mice per group, unpaired *t*-test. (D) qPCR results indicating mtDNA levels upon treatment with LV-*lncMtDloop* in primary cultured neurons. Error bars denote mean ± SEM, with $n = 5$–6 repetitions per group, one-way ANOVA followed by Dunnett's multiple comparisons test. (E) RT-qPCR results showing mtRNA levels following treatment with LV-*lncMtDloop* in primary cultured neurons. Error bars signify mean ± SEM, with $n = 4$–6 per group, one-way ANOVA followed by Dunnett's multiple comparisons test. (F) Relative mitochondrial RNA abundances quantified by RT-qPCR. RNA levels are normalized against WT control. Error bars represent mean ± SEM, with $n = 6$–8 repetitions per group, two-way ANOVA followed by Dunnett's multiple comparisons test. (G) Western blot analysis depicting the levels of OXPHOS subunit proteins upon treatment with LV-*lncMtDloop* in primary cultured neurons. (H) Relative intensities of signals for OXPHOS subunit proteins. Error bars denote mean ± SEM, with $n = 3$ repetitions per group, one-way ANOVA followed by Dunnett's multiple comparisons test. (I) STED imaging showcasing mitochondrial dynamics using MitoESq-635 staining. Scale bars, 1 μm. (J) Mitochondrial fissions within 25% from the top (red) are termed "peripheral fission". The mitochondria marked with a white arrow represent the morphology before and after peripheral fission. Scale bars, 2 μm. (K) Peripheral fission rates in WT and AD neurons. Error bars indicate mean ± SEM, with $n = 7$–8 videos per group, two-way ANOVA followed by Tukey's multiple comparisons test. (L) Mitochondrial fissions within the central 50% (green) are termed "midzone fission". The mitochondria marked with a white arrow represent the morphology before and after midzone fission. Scale bars, 2 μm. (M) Midzone fission rates in WT and AD neurons. Error bars indicate mean ± SEM, with $n = 7$–8 videos per group. For midzone fission rates: two-way ANOVA followed by Tukey's multiple comparisons test. (N) Western blot analysis depicting the levels of Opa1 and Drp1 upon treatment with LV-*lncMtDloop* in primary cultured neurons. (O, P) Relative intensities of signals for Opa1 and Drp1. Error bars denote mean ± SEM, with $n = 4$ repetitions per group, one-way ANOVA followed by Dunnett's multiple comparisons test. (Q) Assessment of mitochondrial oxygen consumption rate (OCR) in primary hippocampal neurons. 3xTg primary hippocampal neurons were infected with LV-*lncMtDloop* or LV-control at DIV5 and subjected to seahorse assay at DIV14. Error bars denote mean ± SEM, with $n = 8$ repetitions per group, unpaired *t*-test. (R, S) Restoration of *lncMtDloop* resulted in increased reserved capacity and basal OCR levels in primary hippocampal neurons, as described above. Error bars represent mean ± SEM, $n = 8$ repetitions per group, unpaired *t*-test. (T) Influence of *lncMtDloop* restoration on cytochrome c oxidase activity assessed using isolated mitochondria from primary cultured neurons as described above. Error bars denote mean ± SEM, $n = 4$ repetitions per group, one-way ANOVA followed by Dunnett's multiple comparisons test. Source data are available online for this figure.

elevation in mtRNA expression, although mtDNA content remained unaffected (Fig. 5D,E). Furthermore, the reintroduction of *lncMtDloop* expression in primary neurons from AD model mice effectively augmented the transcription levels of multiple mitochondrial genome-encoded OXPHOS-related genes that are under the control of TFAM. These genes included *Atp8, CoxI, CoxII, CoxIII, ND1, ND3*, and *ND6*. Interestingly, the levels of mtDNA remained unchanged following this restoration (Fig. 5F). This observation was further validated by Western blot analysis (Fig. 5G,H).

Throughout the cellular lifecycle, the dynamic behavior of mitochondria gives rise to a diverse array of morphologies, often characterized by unpredictability and heterogeneity. The process of mitochondrial fission plays a pivotal role in governing metabolism, proliferation, and mitophagy. Notably, newly emerged daughter mitochondria originating from peripheral fission typically lack nucleoids and replicative mitochondrial DNA (mtDNA); instead, they tend to encapsulate damaged constituents and are marked for eventual mitophagic degradation (Kleele et al, 2021; Kraus et al, 2021). Furthermore, an excess of peripheral fission events, unaccompanied by timely turnover, can result in the accumulation of impaired mitochondria. This accumulation subsequently disrupts the integrity of mitochondrial genomes and overall homeostasis. In our investigation into whether *lncMtDloop* has a regulatory influence on the outcomes of mitochondrial fission in AD neurons, we monitored the varied fission destinies of mitochondria using the MitoTracker probe MitoESq-635. This probe utilizes enhanced squaraine dye and is excited at both 635 and 775 nm for stimulated emission depletion (STED) super-resolution microscopy, thereby offering high temporal and spatial resolution capabilities (Vicidomini et al, 2018; Yang et al, 2020). Employing this advanced technique (Appendix Fig. S6A), our findings disclosed that neurons from the 3xTg model exhibited an elevated frequency of peripheral fission instances, juxtaposed with a

reduced occurrence of midzone fission during mitochondrial division. This combination ultimately led to an increased presence of fragmented mitochondria in comparison to their wild-type (WT) counterparts (Fig. 5I–M; Appendix Fig S6B; Movies EV1–3). Combining with western blot, we showed a significant increase of Drp1 (dynamin-related protein 1) and a decrease of Opa1 (optic atrophy 1) in 3xTg neurons (Fig. 5N–P). Nevertheless, allotropic *lncMtDloop* expression brought about a substantial improvement in this scenario.

mtDNA encodes essential components of the OXPHOS complexes, which play a pivotal role in the synthesis of cellular ATP. Emerging research underscores the significance of mitochondrial architectural dynamics in adapting to varying metabolic requirements (Liesa and Shirihai, 2013). Through global assessments of cellular oxygen consumption rates using the Mito Stress Test, it was evident that the allotropic expression of *lncMtDloop* in primary cultured neurons from 3xTg-AD mice led to notable enhancements in both basal and reserve respiratory capacities (Fig. 5Q–S). Furthermore, the analysis of mitochondrial cytochrome c oxidase activities, a critical enzyme engaged in oxygen consumption during mitochondrial oxidative phosphorylation (Wilson et al, 2014), unveiled a substantial decline in the hippocampal tissues of 3xTg mice when contrasted with age-matched WT controls. This perturbation, however, was effectively counteracted and reversed by the introduction of allotropic *lncMtDloop* expression (Fig. 5T).

## Allotropic *lncMtDloop* expression reverses mitochondrial morphology defects in 3xTg mouse neurons

Accumulation of damages to mtDNA content and integrity can give rise to mitochondrial morphology abnormalities (Hirai et al, 2001). To systematically investigate the potential contributions of *lncMtDloop* to mitochondrial structure, we administered injections of AAV-PHP.eB carrying allotropic *lncMtDloop* fused with the

3'UTR of MRPS12 (designated as SYN: AAV-PHP.eB-*lncMtDloop*-3'UTRMRPS12) via the tail vein of 11-month-old 3xTg-AD mice and their age-matched controls. Six weeks after the initial injection, brain tissues were collected, sectioned, and examined under a microscope to confirm the successful expression of eGFP-tagged *lncMtDloop* from tail injections (Appendix Fig. S7A,B). Subsequently, transmission electron microscopy was performed on these tissue sections to assess mitochondrial ultrastructure (Fig. 6A,D). Classification of quality was systematically carried out based on the number of cristae, specifically Class I (>3 cristae), Class II (2–3 cristae), or Class III (none or 1 crista), along with matrix densities categorized as Class A (dense matrix) or Class B (swollen and hypodense matrix). In 3xTg mice that were injected with AAVs carrying an empty vector, a noteworthy prevalence of Class III and Class B mitochondria was observed in neurons located within the hippocampus and prefrontal cortex regions (Fig. 6B–F), signifying that the mitochondria exhibited small, abbreviated, and fragmented morphologies (Fig. 6G–J). However, these suboptimal conditions regarding mitochondrial cristae and matrix quality were notably eased with the introduction of allotropic *lncMtDloop* expression. Furthermore, the accumulation of abnormal mitochondria without timely clearance indicates mitophagy dysfunction within AD brain neurons (Du et al, 2010; Sorrentino et al, 2017). Of note, the reduced protein levels of several crucial modulators associated with mitophagy, including PINK1, Parkin, and LC3B, observed in the hippocampal and prefrontal cortex tissues, were substantially restored upon the administration of allotropic *lncMtDloop* expression throughout the entire brain of 3xTg mice (Appendix Fig. S8A–C).

## Allotropic *lncMtDloop* expression alleviates Aβ pathology, enhances synaptic plasticity, and improves behavioral deficits in 3xTg mice

A reciprocal relationship has been proposed to exist between AD-related pathologies and the extent of mitochondrial dysfunction (Kerr et al, 2017; Kingwell, 2019). Building upon the observed impacts on mitochondrial morphology and function, we hypothesized that changes in *lncMtDloop* expression might extend to the realm of brain neuropathology. Hence, we embarked on a mouse tail vein injection regimen involving AAV-PHP.eB-*lncMtDloop*-3'UTR$^{MRPS12}$ to examine whether a systemic strategy could be a viable means to comprehensively enhance the overall brain mitochondrial status for potential clinical application. Additionally, we aimed to scrutinize any potential adverse effects that could arise (Fig. 7A). To ascertain the therapeutic potential of this systemic approach, we employed immunofluorescence (IHF) techniques, unveiling the successful introduction of AAV viral particles, carrying either the control construct or one facilitating the allotropic expression of *lncMtDloop*, across the entire brain, including crucial regions like the hippocampus (Appendix Fig. S7A,B). Notably, this systemic *lncMtDloop* expression within the brains of 3xTg-AD mice yielded substantial reductions in the sizes and intensities of Aβ deposits within the hippocampal CA1, CA3, and DG regions, as well as the prefrontal cortex regions (Fig. 7B–D). Consistently, ectopic expression of *lncMtDloop* in the same 3xTg-derived culture neurons reduced the accumulation of Aβ and phosphorylated Tau (Fig. EV5D–I). These findings were further verified through quantification of Aβ$_{1-40}$ and Aβ$_{1-42}$ levels in freshly collected

tissues utilizing the ELISA assay, effectively underscoring the efficacy of the systemic intervention approach (Fig. 7E–H).

Neurons, with their rapid firing of action potentials, axonal nutrient transport, and synaptic activities, inherently demand substantial energy. This energy demand is actively met through mitochondrial bioenergetic processes. Conversely, in the context of AD, mitochondrial dysfunction is intimately tied to synaptic dysfunction and cognitive impairment (Kapogiannis and Mattson, 2011; Khacho et al, 2017). Given that the hippocampus stands as the most vulnerable region to pathological changes in AD, we proceeded with localized stereotaxic injection of AAV9-*lncMtDloop*-3'UTR$^{MRPS12}$ to the hippocampal area, directly evaluating the effects of this allotropic intervention on hippocampal neuron firing (Appendix Fig S7C). Following a span of 6 weeks, an electrophysiological recording was undertaken (Fig. 7I). The electrophysiological recordings yielded significant insights. Impairments in the field excitatory postsynaptic potentials (fEPSPs) of the Schaffer collateral pathway, as well as the long-term potentiation (LTP) typically observed in 3xTg mice, were effectively restored through the specific and localized hippocampal expression of AAV9-*lncMtDloop*-3'UTR$^{MRPS12}$ (Fig. 7J,K). Next, behavioral evaluation on social, cognitive, and memory-associated behavior, which are mainly associated with the prefrontal cortex and hippocampus, was performed after 6 weeks of administrated injections of AAV-PHP.eB-*lncMtDloop*-3'UTR$^{MRPS12}$ into the mouse tail vein. Notably, in tasks such as nest-building (Appendix Fig. S9A,B), open-field exploration (Appendix Fig. S9C,D), and novel object recognition (Appendix Fig. S9E), the performance of 3xTg-AD mice demonstrated deficiencies when compared to WT controls, as anticipated. Nevertheless, the allotropic expression of *lncMtDloop* via systemic AAV particle injections led to significant improvements. Furthermore, a contextual and cued fear conditioning test, which involves hippocampus-dependent learning and memory tasks, was carried out. Rescuing *lncMtDloop* expression significantly improved both context and cued freezing time in 3xTg mice (Fig. 7L,M). To further assess the impact of allotropic *lncMtDloop* overexpression on spatial memory, the Morris water maze (MWM) test was employed. While the performance of 3xTg mice was compromised, the systemic allotropic expression of *lncMtDloop* throughout the brain notably ameliorated the primary deficits observed in the MWM, effectively outperforming the control group (Fig. 7N–R).

# Discussion

In this study, we have unveiled a previously undiscovered yet evolutionarily conserved lncRNA originating from the mitochondrial DNA D-loop region, aptly named *lncMtDloop*. Our data has illuminated its pivotal regulatory role in upholding TFAM-dependent transcription of the mitochondrial genome and ensuring overall homeostasis. Employing meticulous in vitro and in vivo methodologies, we meticulously dissected the molecular intricacies of *lncMtDloop*'s localization and interactions, further unraveling its influence on mitochondrial genome transcription and balance. Notably, the influence of *lncMtDloop* on the preservation of mitochondrial morphology, metabolism, and mitophagy has been underscored, with its absence leading to deficits in neuronal firing and impairments in cognition and memory within the 3xTg mice.

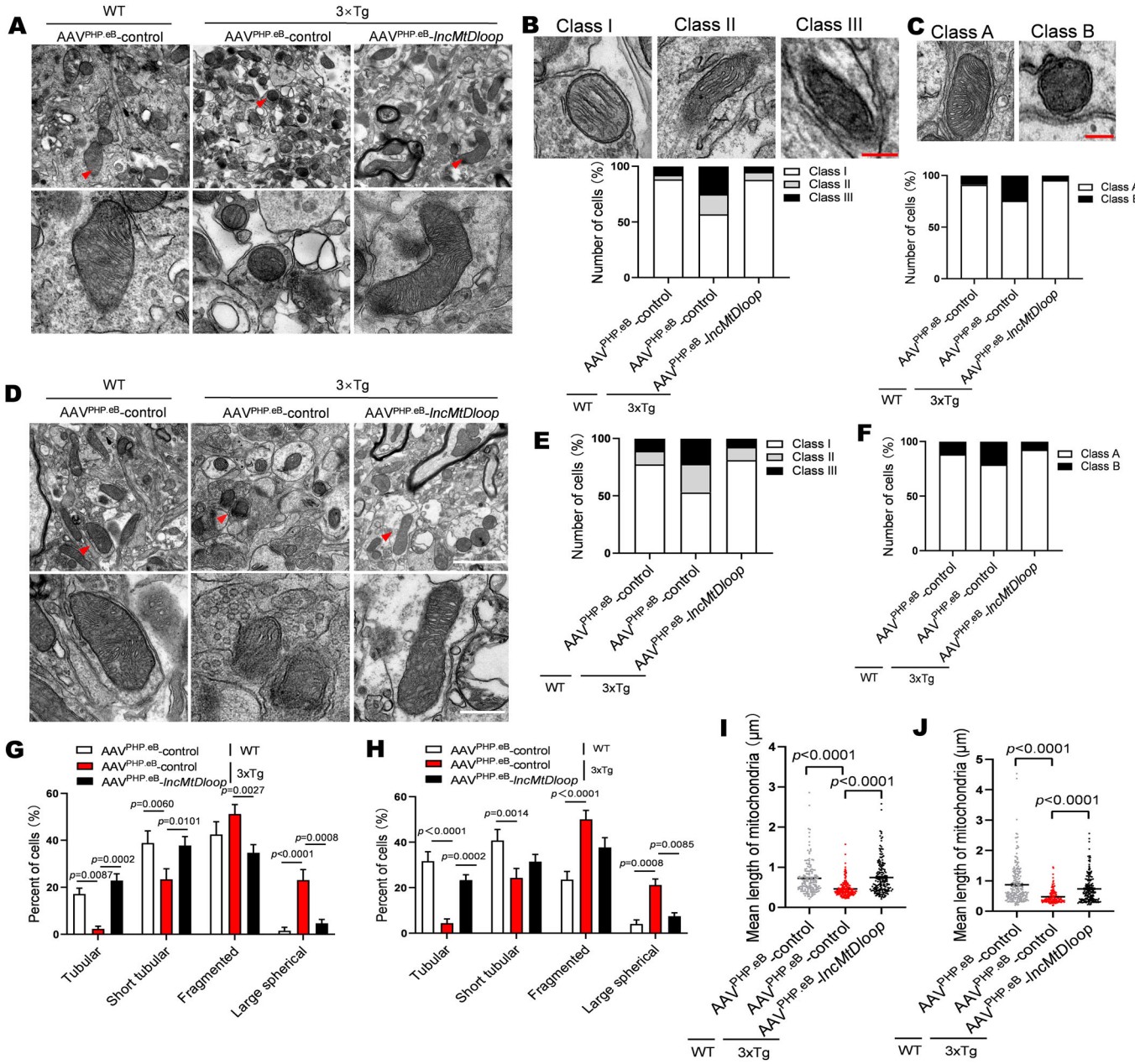

**Figure 6. Improvement of mitochondrial morphology through allotropic *lncMtDloop* expression in the brains of 3xTg mice.**

(**A**) Representative electron micrographs showcasing mitochondrial ultrastructure in hippocampal neurons of 12-month-old wild type and 3xTg mice, with or without (w/o) *lncMtDloop* restoration. $N = 15$ random fields from brain slice samples of 6 animals per group. The mitochondria marked with red arrow were magnified at the bottom. Scale bars are 2 μm and 500 nm. (**B**) Percentage distribution of cells in different categories: Class I (more than three cristae), Class II (two to three cristae), and Class III (no or one cristae), in hippocampal neurons as shown in (**A**). $N = 15$ random fields from brain slice samples of 6 animals per group. Scale bars are 500 nm. (**C**) Percentage distribution of cells in different categories: Class A (dense matrix) and Class B (swollen mitochondria with hypodense matrix) in hippocampal neurons as shown in (**A**). $N = 15$ random fields from brain slice samples of 6 animals per group. Scale bars are 500 nm. (**D**) Representative electron micrographs showcasing mitochondrial ultrastructure in prefrontal cortex (PFC) neurons of 12-month-old wild type and 3xTg mice, with or without (w/o) *lncMtDloop* restoration. $N = 15$ random fields from brain slice samples of 6 animals per group. The mitochondria marked with red arrow were magnified at the bottom. Scale bars are 2 μm and 500 nm. (**E**) Percentage distribution of cells in different categories: Class I, Class II, and Class III, in PFC neurons as shown in (**D**). $N = 15$ random fields from brain slice samples of 6 animals per group. (**F**) Percentage distribution of cells in different categories: Class A and Class B in PFC neurons as shown in (**D**). $N = 15$ random fields from brain slice samples of 6 animals per group. Scale bars are 500 nm. (**G, H**) Percentage alteration in mitochondrial morphology (tubular, short tubular, fragmented, and large spherical) in hippocampal and PFC neurons as depicted in A) and D). Error bars denote mean ± SEM, $n = 15$ random fields from brain slice samples of 6 animals per group, two-way ANOVA followed by Tukey's multiple comparisons test. (**I, J**) Measurement of mitochondrial lengths in hippocampal and PFC neurons as shown in (**A, D**). Error bars signify mean ± SEM, with $n = 157$–212 mitochondria per group, one-way ANOVA followed by Dunnett's multiple comparisons test. Source data are available online for this figure.

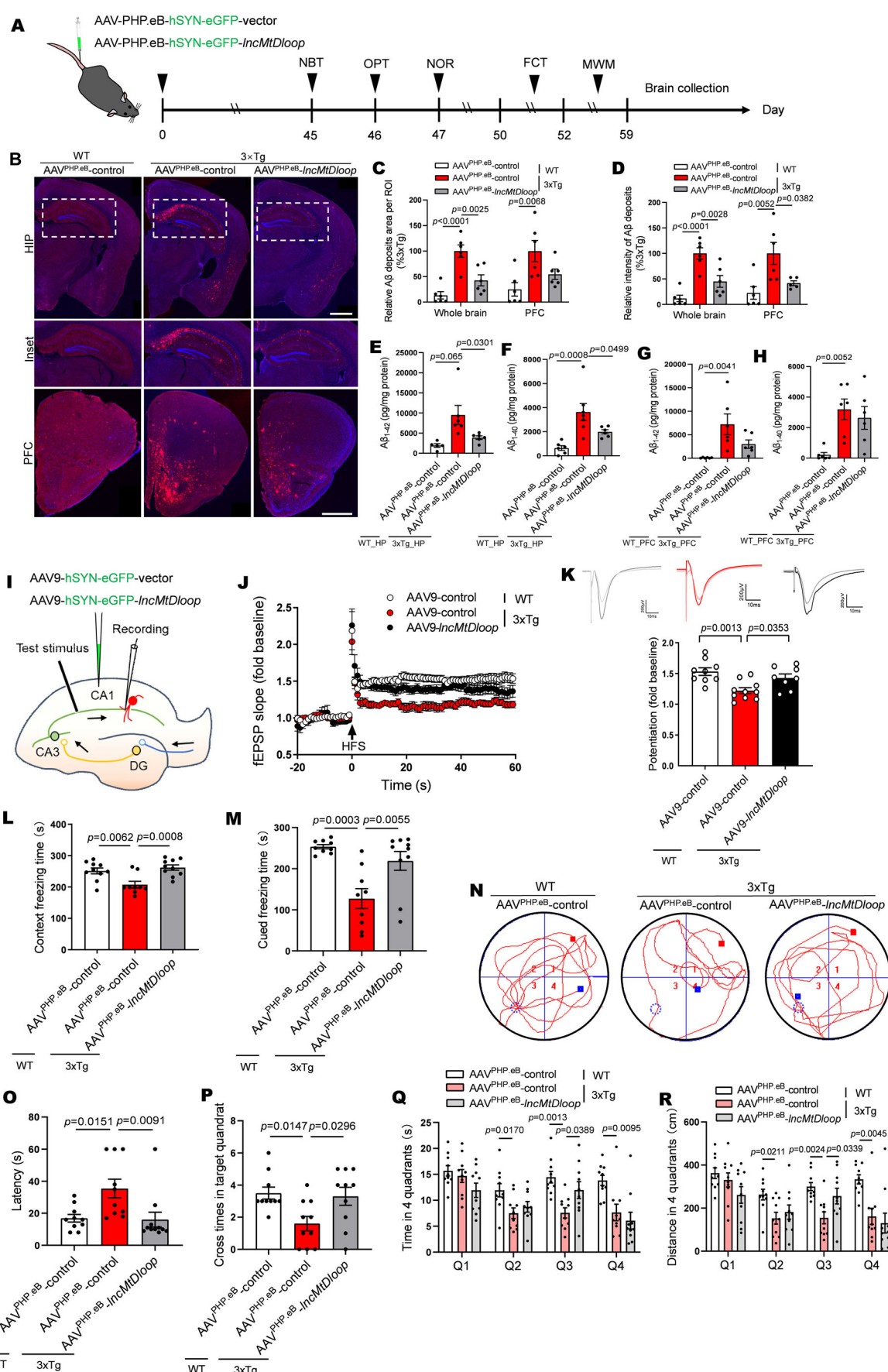

◀ **Figure 7. Allotropic *lncMtDloop* expression improves Aβ pathology, synaptic plasticity impairments, and behavioral deficits in 3xTg mice.**

(A) Timeline illustrating the sequence of behavioral tests conducted following intravenous injections of AAV-PHP.eB into the lateral tail vein of mice. (B) Representative immunohistochemical fluorescence (IHF) images displaying 6E10 staining in the hippocampus and prefrontal cortex (PFC) of WT and 3xTg mice with or without restoration of *lncMtDloop* through AAV-PHP.eB injections. Scale bars, 1 mm. (C, D) Quantification of average Aβ plaque area and fluorescence intensity of 6E10 staining in (B). Error bars denote mean ± SEM, with $n = 6$ mice per group, one-way ANOVA followed by Dunnett's multiple comparisons test. (E, F) ELISA analysis revealed alterations in $A\beta_{42}$ and $A\beta_{40}$ levels in hippocampal tissues (HP) of WT and 3xTg mice w/o *lncMtDloop* restoration as described above. (G, H) ELISA analysis demonstrating changes in $A\beta_{42}$ and $A\beta_{40}$ levels in PFC tissues of WT and 3xTg mice w/o *lncMtDloop* rescue as described above. Error bars signify mean ± SEM, with $n = 6$ mice per group, one-way ANOVA with Dunnett's multiple comparisons tests. (I) Schematic depiction of recording and stimulating electrode placements for synaptic transmission via the Schaffer collateral pathway after AAV9 hippocampal injection. (J) Field excitatory postsynaptic potentials (fEPSPs) evoked by Schaffer collateral pathway stimulation during the LTP experiment in acute hippocampal slices from WT and 3xTg mice without *lncMtDloop* restoration. Error bars indicate mean ± SEM, with $n$ (brain slice) = 8–10 per group from 6 animals per genotype. (K) Representative traces and graphs displaying electrophysiological recordings from acute hippocampal slices of each group (top). Quantification of average fEPSPs during the last 10 min of the time course in (J) (bottom). Error bars signify mean ± SEM, with $n$ (brain slice) = 8–10 per group from 6 animals per genotype, one-way ANOVA with Dunnett's multiple comparisons test. (L, M) Contextual fear conditioning tests and cued fear conditioning tests were conducted in mice as described above. Error bars indicate mean ± SEM, with $n = 9$–10 animals per group, one-way ANOVA with Dunnett's multiple comparisons tests. (N) Escape traces to the hidden platform of each group in the Morris water maze (MWM) on day 6. The red dot indicates the start point. The blue dot indicates the stop point. The blue dotted cycle outlines the hidden target quadrant. (O) Latencies to the target platform of each group in the MWM on day 6. Error bars signify mean ± SEM, with $n$ (brain slice) = 8–10 per group from 6 animals per genotype, one-way ANOVA with Dunnett's multiple comparisons test. (P) Number of crosses in the target quadrant of the MWM on day 6. Error bars signify mean ± SEM, with $n$ (brain slice) = 8–10 per group from 6 animals per genotype, one-way ANOVA with Dunnett's multiple comparisons test. (Q, R) Time spent and distance traveled in the four quadrants of the MWM on day 6. Error bars signify mean ± SEM, with $n = 10$ animals per group, one-way ANOVA with Dunnett's multiple comparisons test. Source data are available online for this figure.

Crucially, our findings illuminate the potential of allotropic *lncMtDloop* expression—achieved through systemic administration of the construct—in ameliorating AD-like pathological hallmarks and functional impairments.

By considering both gain-of-function and loss-of-function strategies within the mitochondrial context, we meticulously refined an allotropic expression construct for *lncMtDloop*. This construct serves the crucial purpose of restoring the expression levels of *lncMtDloop*. Our optimization process involved the fusion of *lncMtDloop* with the 3′UTR of *MRPS12*, facilitating its targeted localization to the mitochondrial surface—a technique supported by previous research (Chin et al, 2018; Wang et al, 2012b). Furthermore, p32, as a binding partner of *lncMtDloop*, can efficiently facilitate free recombinant *lncMtDloop* from cytosol to mitochondria surface. PNPASE emerged as a key player in the RNA import machinery of mammalian mitochondria (Wang et al, 2010; Wang et al, 2012a). Our results revealed that PNPASE aids in channeling *lncMtDloop* into mitochondria through its interaction with the predicted stem-loop structure of the lncRNA. This revelation carries immense significance as it empowers the allotropic expression of *lncMtDloop*, enabling the intervention strategies we employed in this study—namely, the reintroduction of exogenous *lncMtDloop* into mitochondria whenever the endogenous levels falter.

LncRNAs are prevalent within mammalian brains (Palazzo and Koonin, 2020; Pollard et al, 2006), yet a comprehensive understanding of their functions, especially those situated uniquely in mitochondria, remains limited. Unlike better-characterized nuclear-encoded lncRNAs, the synthesis and roles of mitochondria-derived lncRNAs are not well-defined. Nevertheless, their significance in normal brain function and implications in neurodegenerative disorders are hinted at by altered expressions (Faghihi et al, 2008; Mus et al, 2007; Rogaeva et al, 2007). Previous studies suggest the D-loop region interacts with proteins like TFAM, SSBP1, HADHA, ATP5D, and p32, although their roles in mitochondria are unclear. *LncMtDloop*, originating from the mtDNA D-loop region and interacting with various proteins, adds a new dimension (Choi et al, 2011). A previous report also indicates

that the element within the D-loop region plays a crucial role in mediating the transition between mitochondrial DNA replication and transcription events. (Jemt et al, 2015). Given the transcription of *lncMtDloop* originating from the D-loop, it is plausible that it interacts with mitochondrial proteins, thereby participating in the regulation of the mitochondrial genome, which in turn is intricately associated with various cellular processes.

LncRNAs are conventionally recognized for roles in transcriptional, translational, and post-translational regulations. Within mitochondria, devoid of histone proteins, mtDNA mutations occur more frequently compared to nuclear DNA. This culminates in mtDNA loss, diminishing the transcription of vital OXPHOS-related proteins essential for cellular metabolism. Such changes are well-documented in age-related brain disorders like AD and Parkinson's disease. Our study reveals that *lncMtDloop* directly binds to TFAM, inducing mitochondrial gene transcription in a *cis* manner by facilitating TFAM recruitment to mtDNA D-loop region promoters. A recent study indicates that the non-coding 7S RNA triggers POLRMT dimerization and subsequently regulates mitochondrial transcription through a series of in vitro transcription and transcription pulse-chase experiments. (Zhu et al, 2022). We have conducted qPCR, ChIP, and luciferase reporter assays to elucidate the regulatory mechanism of *lncMtDloop* in mitochondrial transcription, nevertheless, future studies should include in vitro transcription assays in the presence of *lncMtDloop* to clarify its direct role in mtDNA transcription. Indeed, allotropic *lncMtDloop* expression effectively restores mtDNA transcription, bolstering mitochondrial genome integrity and respiratory function, even as other *lncMtDloop*-interacting partners beyond TFAM could also play a role.

Due to the energy-intensive nature of the nervous system, which demands substantial ATP synthesis for neural activity and synaptic transmission. Consequently, the neuron's susceptibility to mitochondrial dysfunction in the AD brain is heightened (Swerdlow et al, 2014). Given this context, *lncMtDloop* assumes clinical significance regarding its diminished expression in AD patient brains and 3xTg mice. The most conspicuous decline in *lncMtDloop* occurs within neurons compared to other brain cell

types. Strikingly, restoring *lncMtDloop* not only directly boosts mtDNA transcription but also rejuvenates mitochondrial function and dynamics within AD neurons. Employing high-resolution STED imaging, we scrutinized the adverse impact of *lncMtDloop* loss on mitochondrial morphology and dynamics, revealing significant intra-organelle mitochondrial abnormalities due to reduced *lncMtDloop* levels. Insufficient *lncMtDloop* corresponds with heightened peripheral fission, fragmentary forms, swollen cristae, and hypodense matrix within neurons taken from AD mice. Other reports have corroborated that excessive mitochondrial fission without timely mitophagy-mediated clearance compromises mitochondrial genome integrity, and nucleoid organization, leading to mtDNA loss, and overall mitochondrial dysfunction (Sabouny and Shutt, 2021). Collectively, these findings underscore *lncMtDloop* as an essential player derived from the mitochondrial genome, pivotal for upholding mitochondrial morphology, dynamics, function, and overall homeostasis.

Aligned with the "mitochondrial cascade hypothesis," our investigations highlighted the substantial impact of *lncMtDloop* on Aβ deposition clearance. This beneficial effect at the cellular level translated into improvements in electrophysiological responses and behavioral outcomes. Furthermore, administering the AAV construct systemically through intravenous injection effectively reinstated global brain *lncMtDloop* expression and function. This approach leveraged the AAV-PHP.eB vector delivery system, known for its ability to traverse the blood-brain barrier. This system holds promise as a potent tool for expanding systemic gene therapy, enabling the delivery of constructs to peripheral regions with eventual targeting of the brain (Mathiesen et al, 2020). This innovation presents an innovative and potentially translatable avenue for AD therapeutics.

# Methods

### Human brain samples

Human autopsy paraffin-embedded 10 μm brain sections were from the following sources with approval from the appropriate local regulatory authorities. We examined 16 case-patients graciously provided by the University of Pittsburgh Alzheimer's Disease Research Center (ADRC) brain bank with approval from the Committee for Oversight of Research and Clinical Training Involving Decedents (CORID). Each case had been diagnosed neuropathologically and ranked by Braak stage (Appendix Table S1). Additional frozen tissue was a generous gift of the ADRC at Washington University in St. Louis (Grant P50-AG-05681) with approval from the Neuropathology Core (protocol #T1016) at Hong Kong University of Science and Technology.

### Rhesus macaque specimens

Frozen postmortem brain tissues from rhesus macaque were obtained from the Kunming Primate Research Center of the Chinese Academy of Science (KPRC). Four- to 6-year-old monkeys are characterized as a young group, and 16- to 20-year-old monkeys are characterized as the old group. The hippocampus (including CA1, CA3, and DG) and PFC region were systematically collected from well-characterized rhesus macaque of sex identity.

All the collected brain tissues were stored at liquid nitrogen temperature.

### Primary cells

For primary microglia cultures, WT female mice or 3xTg female mice were mated with the WT male mice or 3xTg male from an unrelated colony. The hippocampus was isolated from newborn mice (P0-P4), and cut into small pieces. The hippocampal tissue was digested with Neurosphere Dissociation Kit (Cat# 130-095-943) and centrifuged for 5 min at 1500 rpm. The purified microglia were resuspended in DMEM/F-12 (Thermo Fisher Scientific) medium with 10% FBS and penicillin-streptomycin (Invitrogen) and then seeded on 6- or 24-well plates with 14–21 days (Lee et al, 2018; Litvinchuk et al, 2018). Cultures were kept in a 37 °C incubator with 5% $CO_2$. About 1/2rd of the medium was replaced with fresh medium every 3 days. For primary astrocyte cultures, WT female mice or 3xTg female mice were mated with the WT male mice or 3xTg male from an unrelated colony. The hippocampus was isolated from newborn mice (P0-P4), and cut into small pieces. The hippocampal tissue was digested with a Neurosphere Dissociation Kit (Cat# 130-095-943). After mechanical dissociation and centrifugation, the cells were resuspended in DMEM/F-12 (Thermo Fisher Scientific) medium with 10% FBS, penicillin-streptomycin (Invitrogen), 15 mM HEPES, 14.3 mM $NaHCO_3$, 1% Fungizone® and 0.04% gentamicin, and then seeded on 6- or 24-well plates pre-coated with poly-L-lysine with 14–21 days (Wyse et al, 2020). Cultures were kept in a 37 °C incubator with 5% $CO_2$. About 1/2rd of the medium was replaced with fresh medium every 3 days. For primary mouse hippocampal neuronal cultures, WT female mice or 3xTg-AD female mice were mated with the WT male mice or 3xTg-AD male from an unrelated colony. The hippocampus was isolated from embryos of day E17 and cut into small pieces. The hippocampal tissue was digested with a Neurosphere Dissociation Kit (Cat# 130-095-943). The primary mice hippocampal neurons were resuspended in Neurobasal plus medium supplemented with B27-P (Thermo Fisher Scientific), Glutamax (Thermo Fisher Scientific), and penicillin-streptomycin (Invitrogen) and then seeded on 6- or 24-well plates pre-coated with poly-L-lysine with 14–21 days (McInnes et al, 2018). Cultures were kept in a 37 °C incubator with 5% $CO_2$. About 1/2rd of the medium was replaced with fresh medium every 3 days. For the $Aβ_{1-42}$ fibrils, $Aβ_{1-42}$ peptide (sigma) was dissolved in sterile PBS, diluted to a final concentration of 250 μM and incubated on a shaker at 37 °C for 84 h. In DIV14 or DIV21, cultures were treated with $Aβ_{1-42}$ fibrils in indicated concentrations for 18 h, and cells were collected for subsequent experiments (Ising et al, 2019).

### Cell line

N2A cells, Cos-7 cells, and SH-SY5Y cells were cultured in DMEM (Thermo Fisher Scientific) medium with 10% FBS and penicillin-streptomycin (Invitrogen). Cultures were kept in a 37 °C incubator with 5% $CO_2$. About 1/2rd of the medium was replaced with fresh medium every 3 days. For the conduction of mtDNA depletion, EB (Ethidium Bromide) was introduced into the medium at a concentration of 100 ng/mL for 8 weeks to induce mtDNA depletion, facilitating further investigation into mitochondrial function.

*Mice*

All mouse lines, including the 3xTg line (B6. Cg-Tg (APPSwe, tauP301L)1Lfa PSEN1tm1Mpm/J; The Jackson Laboratory) and age- and gender-matched WT mice controls were maintained on C57/BL6 background. Mice were group-housed in cohorts of five, separated by sex, in individual home cages under specific-pathogen-free conditions with free access to food and water. The light and dark cycle was 12 h/12 h, and the temperature was kept constant at 23 ± 2 °C and humidity (50 ± 5%). Mice were randomly assigned to the experimental groups. Behavioral experiments were performed during the dark cycle. The ages of mice used for each behavioral experiment were 11–13 months. All animal experiments complied with the Biomedical Ethics Committee for Animal Use and Protection of Peking University.

## Plasmids

AC027613.1 and relevant truncated mutants were cloned into pcDNA3.1 or pGL4.10 (pcDNA3.1-T7-AC027613.1, pcDNA3.1-T7-AC027613.1#1, pcDNA3.1-T7-AC027613.1#2, pcDNA3.1-T7-AC027613.1#3, pcDNA3.1-T7-AC027613.1#4, pcDNA3.1-T7-AC027613.1#5, pcDNA3.1-T7-AC027613.1#6, pGL4.10-1200, pGL4.10-1000, pGL4.10-800, pGL4.10-600, pGL4.10-400, pGL4.10-200).

## Reagents and assay kits

EB (E1385, Sigma), HiScript®II Q RT SuperMix for qPCR (Cat# Q711-02), mMESSAGE mMACHINE T7 ULTRA Kit (Cat# AM1345), RNA 3′ End Desthiobiotinylation Kit (Cat# 20163), Magnetic RNA-Protein Pull-Down Kit (Cat# 20164), Magna RIP Kit (Cat# 17-700), Dual-Luciferase® Reporter Assay Kit (Cat# E1960), SimpleChIP® Enzymatic Chromatin IP Kit (Cat# 9003), Chemiluminescent EMSA Kit (Cat# GS009), Seahorse XF Cell Mito Stress Test Kit (Cat# 103015-100), Cytochrome c Oxidase Assay Kit (Cat# CYTOCOX1), Cell Mitochondrial Isolation Kit (Cat# C3601), Tissue Mitochondrial Isolation Kit (Cat# C3606), Neurosphere Dissociation Kit (Cat# 130-095-943), RNAscopre® H2O2 and Protease Reagents (Cat# 322381), RNAscopre® Multiplex-Fluorescent Detection (Cat# 323110), RNAscopre® Probe Mmu-Mt-AC027613.1 (Cat# 518361), RNAscopre® Probe Mm-Mt-AC027613.1 (Cat# 518381).

## siRNA sequences

si_p32-1: CCCAAGAUGUCUGGAGAUUTT, AAUCUCCAGACAUCUUGGGTT

si_p32-2: UGAACGGCACGGAGGCUAATT, UUAGCCUCCGUGCCGUUCATT

si_p32-3: GAGCCAGAACUGACAUCAATT, UUGAUGUCAGUUCUGGCUCTT

si_TFAM-1: GGGAAGAGCAGAUGGCUGATT, UCAGCCAUCUGCUCUUCCCTT

si_TFAM-2: ACAAAGAAGCUGUGAGCAATT, UUGCUCACAGCUUCUUUGUTT

si_TFAM-3: GGUAAAGAGAAGAGAAUUATT, UAAUUCUCUUCUCUUUACCTT

si_PNPASE: purchased from Santa Cruz (Cat# sc-6137)

The siRNA sequences are seen in Appendix Table S2.

## RNA extraction and real-time RT-qPCR

Total RNA was extracted from cultures or mice tissues using RNAiso plus reagent following the manufacturer's instructions. RNA quantity and quality were evaluated by NanoDrop ND-1000 spectrophotometer. The real-time quantitative reverse transcription PCR (RT-qPCR) was performed using a Reverse Transcription kit (vazyme) and SYBR Green I Master (vazyme). qPCR was performed in a 10 μl reaction system including 1 μl cDNA, 5 μl 2× Master Mix, 0.1 μl of forward primer (10 μM), 0.1 μl of reverse primer (10 μM), and 3.8 μl of double distilled water. β-Actin was used as an endogenous reference for data normalization. Primer sequences are described in the Appendix Table. Cycle threshold values were used to calculate fold changes in gene expression using the $2^{-\triangle\triangle Ct}$ method. The primer sequences of probes are seen in Appendix Table S3.

## Tissue preparation and western blot

Samples from mouse brain and primary cell culture were collected and homogenized in RIPA buffer containing protease inhibitors and phosphatase inhibitors. After centrifugation, supernatants were heated to 95 °C for 10 min and resolved on 6–12% polyacrylamide precast gels. The gel was transferred to the PVDF membrane at 250 mA for 90 min at 4 °C and blocked for 1 h at room temperature in 5% BSA in Tris-buffered saline with 0.05% Tween 20. Membranes were incubated in primary antibody overnight at 4 °C. Antibodies used for western blot were: anti-6E10 (803001, BioLegend), anti-p32 (ab24733, Abcam), anti-Atp5a (ab14748, Abcam), anti-LC3B (ab51520, Abcam), anti-PINK1 (ab23707, Abcam), anti-parkin (#2132, Cell Signaling), anti-β-actin (ab8227, Abcam), anti-COX IV (ab33985, Abcam), anti-Stxb1 (ab183722, Abcam), anti-Tppp (ab92305, Abcam), anti-Aly/Ref (ab202894, Abcam), anti-Atp5d (ab174438, Abcam), anti-TFAM (ab252432, Abcam), anti-PNPASE (sc-271479, Santa Cruz), anti-VDAC (ab15895, Abcam), anti-Drp1 (ab56788, Abcam), anti-Opa1 (ab157457, Abcam), anti-OXPHOS (ab110413, Abcam). Antibodies used in this study are shown in Appendix Table S4. All blots were imaged using HRP-conjugated secondary antibodies. After three washes for 10 min each, the fluorescence signals were quantified using Image J software.

## Immunofluorescence staining

Mice were perfused with PBS, and the brains were fixed with 4% paraformaldehyde (PFA). The brains were sliced into 30 μm thick with a Leica CM1950 freezing microtome. For immunofluorescence staining, brain sections were permeabilized and blocked in PBS with 0.3% Triton X-100 and 10% goat serum at room temperature for 1 h, followed by incubation with the primary antibody: anti-6E10 (803001, BioLegend), anti-p32 (ab24733, Abcam), anti-Atp5a (ab14748, Abcam), anti-LC3B (ab51520, Abcam), anti-PINK1 (ab23707, Abcam), anti-parkin (#2132, Cell Signaling), anti-AT8 (MN1020, Thermo Fisher Scientific), anti-AT180 (MN1040, Thermo Fisher Scientific), anti-HT7 (MN1000, Thermo Fisher Scientific), anti-TAU-5 (AHB0042, Thermo Fisher Scientific). For the confocal microscopic analysis (TCS SP8, Leica), the primary antibodies were visualized with Alexa-Fluor488, Alex-Fluor594, and Alexa-Fluor647 secondary antibodies (Invitrogen). DAPI

(ab104139, Abcam) was used for nuclear counterstaining. For primary cultures, mice hippocampal neurons, microglia, and astrocytes were fixed with 4% PFA, then permeabilized and blocked in PBS with 0.3% Triton X-100 and 10% goat serum at room temperature for 1 h. Primary antibodies were: anti-LC3B (ab51520, Abcam), anti-PINK1 (ab23707, Abcam), anti-parkin (#2132, Cell Signaling), anti-Atp5a (ab14748, Abcam), anti-Iba1 (019-19741, Wako), anti-GFAP (837201, BioLegend), anti-p32 (ab24733, Abcam), anti-TFAM (ab252432, Abcam), anti-MAP2 (ab32454, Abcam). For the confocal microscopic analysis (TCS SP8, Leica), the primary antibodies were visualized with Alexa-Fluor488, Alex-Fluor594, and Alexa-Fluor647 secondary antibodies (Invitrogen). DAPI (ab104139, Abcam) was used for nuclear counterstaining.

## RNAscope assay for *lncMtDloop* detection

Detection of *lncMtDloop* was performed on cryostat brain slices of mice and primary cultured cells. At first, we prepared and pretreated the samples with RNAscope® H₂O₂ and protease reagents (Cat# 322381). Next, we cover each slide or cells cultured on coverslips with the probe sets against human and mouse *lncMtDloop* (Cat# 518381 and 518361). A positive-control probe, *Homo sapiens peptidylprolyl isomerase B* (*PPIB*), and a negative-control probe for *bacterial gene dapB* were included for each group. The probe hybridization was performed in the HybEZ™ for 2 h at 40 °C. The signal amplification was conducted according to the manufacturer's instruction using RNAscope® Multiplex Fluorescent Detection Reagents (Cat# 323110). The fluorescence signals of *lncMtDloop* were visualized with TSA® plus fluorophore. The brain slices and cells cultured on coverslips were imaged on confocal microscopy (TCS SP8, Leica). The sequences of probes are seen in Appendix Table S5.

## STED superresolution for live cells

Primary neuronal cultures were cultured at a suitable density (moderate) and seeded on glass-bottomed dishes. At DIV3, neurons were transfected with a lentiviral vector to evaluate the gain-of-function of *lncMtDloop*. At DIV14, primary neuronal cultures were incubated with MitoESq-635 at a concentration of 1 μM for 5 min in a humid atmosphere with 5% (v/v) CO₂ at 37 °C to label the mitochondrial membrane, then imaged with STED nanoscopy. STED imaging was performed with a Leica TCS SP8 STED 3X system equipped with a white light laser for excitation and a 775-nm pulsed laser for STED depletion. A ×100 oil-immersion objective (Leica, N.A. 1.4) was employed.

## Electron microscopy

The mice were perfused with paraformaldehyde-glutaraldehyde mixed fixative (2/2.5%), and the hippocampus was removed and post-fixed at 4 °C using paraformaldehyde-glutaraldehyde mixed fixative (2/2.5%) in 0.1 M PB (PH 7.4). The hippocampal samples from each group were washed with 0.1 M PB (PH 7.4) three times for 15 min and then post-fixed with 1% OsO₄ for 2 h at 4 °C, followed by serial ethanol dehydration and acetone transition for 5 min, embedding in Epon 812 resin, polymerization and polymerization at 60 °C for 48 h. The hippocampal samples were sliced into 60 nm for ultrathin sections of transmission electron

microscopy with a Leica UC7 ultramicrotome. Ultrathin sections were loaded onto 100-mesh Cu grids and double stained with 2% uranyl acetate and lead citrate before observations employing a JEM 1400 plus transmission electron microscopy.

## RNA pulldown and mass spectrometry (MS)

*LncMtDloop* was in vitro transcribed using the mMESSAGE mMACHINE T7 ULTRA Kit (Ambion, AMB1345), followed by biotinylation with the Pierce RNA 3′ End Desthiobiotinylation Kit (Thermo Fisher Scientific, 20163) RNA pull-down was performed using a Magnetic RNA-protein Pull-Down Kit (Thermo Fisher Scientific, 20164) with the manufacturer's guidelines. The retrieved protein was detected by slivery staining (Thermo Fisher Scientific, 24612). The specific bands were identified by MS and western blot. For MS, specific bands and retrieved protein pull-downed by *lncMtDloop* or antisense probes were collected and digested in 50 mM triethylammonium bicarbonate with chymotrypsin and modified trypsin at 30 °C overnight. After digestion, each peptide sample was extracted and speed vacuum-dried. Dried proteins were resuspended in a 200 μl volume of buffer (0.1% formic acid and 2% acetonitrile) and loaded on a Bruker Autoflex speed TOF/TOF MALDI-TOF system for LC/MS/MS. The acquired peptide-sequencing data were loaded in Proteome Discoverer (version 1.3.0.339, Thermo Scientific) and analyzed with Mascot software (version 2.3.01, Matrix Science) against a *Mus musculus* protein database to identify proteins.

## Northern blot

Total RNA (10 μg for human postmortem PFC and hippocampal frozen tissues of AD and control individuals as well as WT and 3xTg mouse brain fresh tissues, 2 μg for mouse primary neurons) was denatured using NorthernMax®-Gly sample loading dye (Ambion) and resolved on 1.2% agarose gel in MOPS buffer. The gel was soaked in 1 × TBE for 20 min and transferred to a Hybond-N+ membrane (GE Healthcare) for 1 h (15 V) using a semi-dry blotting system (Bio-Rad). Membranes were dried and UV-crosslinked with 150 mJ/cm² at 254 nm. Pre-hybridization was done at 68 °C for 1 h and using DIG Northern Blot Starter KIT (12039672910, Roche). DIG-labeled in vitro transcribed *lncMtDloop*, and control probes were hybridized overnight. The membranes were washed three times in 2× SSC, 0.1% SDS at 68 °C for 30 min, followed by three 30 min washes in 0.2× SSC, 0.1% SDS at 68 °C. The immunodetection was performed with anti-DIG AP-conjugated antibodies. Immunoreactive bands were visualized using a CDP star reagent (Roche) and a LAS-4000 detection system (GE Healthcare). The sequences of primer for T7 template amplification for northern blot probes are seen in Appendix Table S6.

## RNA immunoprecipitation (RIP)

RIP was performed using a Magnetic RIP RNA-Binding Protein Immunoprecipitation (Millipore, 17-700) by the manufacturer's guidelines. The primary cultured neurons were treated with ice-cold PBS and IP lysis buffer (Thermo Fisher Scientific, 87787) supplemented with RNase inhibitors and a protease inhibitor cocktail. The supernatants were incubated with the anti-p32 (ab24733, Abcam) antibody, anti-TFAM (ab252432, Abcam), or

control mouse IgG with rotation at 4 °C overnight, and then protein A/G beads were added to each group. After the incubation at 4 °C overnight, we collected the precipitates and extracted RNA by proteinase K-chloroform method. The enrichment of *lncMtDloop* was detected using RT-qPCR. The primer sequences of RT-qPCR are seen in Appendix Table S7.

## Luciferase reporter gene assays

Luciferase assays were performed using the Dual-Luciferase® Reporter Assay Kit (Promega, E1960). N2a cells were planted in 96-well plates at $2.0 \times 10^4$ cells/well. After culturing for 2 days, siRNA for TFAM was transfected into the cell lines by using Lipofectamine 2000 (Invitrogen, 11669019). After transfection for 2 days, cells were cotransfected with the luciferase reporter constructs (150 ng/well) and internal control plasmid pRL-CMV (30 ng/well). Two days after transfection, the cells were collected to measure the luciferase activity using a luminometer according to the manufacturer's instructions.

## Chromatin immunoprecipitation (ChIP) assays

N2A cells were collected at ~90% confluency from 15 cm dishes, then performed ChIPassay using the SimpleChIP® Enzymatic Chromatin IP Kit (Cell Signaling, 9003). Firstly, cells were crosslinked with 37% formaldehyde for 10 min, quenched, and digested with micrococcal nuclease for 20 min at 37 °C. The digested chromatin was sonicated in 500 μl ChIP buffer on ice using a sonicator (three times, the 20 s each). Chromatin was Sheared chromatin was centrifuged at 9400 × *g* for 10 min at 4 °C. The supernatant was collected and incubated with specific antibodies against TFAM (Abcam, ab252432) or IgG antibody at 4 °C overnight with rotation, then incubated with protein G magnetic beads at 4 °C for 2 h with rotation. After purification, immunoprecipitated DNA was analyzed by quantitative real-time PCR (40 cycles). Primer pairs must range from 150 bp to 22 bp long. The primer sequences of the promoter of mtDNA containing the TFAM binding domain are shown in Appendix Table S8.

## Electrophoretic mobility shift assay (EMSA)

The probes for EMSA were labeled with biotin at their 3′-end using the EMSA Probe Biotin Labeling Kit (Beyotime, GS008). The nuclear extract, cytosolic extract, and mitochondrial extract were prepared using hippocampus tissue lysate using the Nuclear and Cytoplasmic Protein Extraction Kit (Beyotime, P0027) and Tissue Mitochondria Isolation Kit (Beyotime, C3606). The assay was separated by native 4% PAGE gels at 100 V and transferred onto nylon membranes at 380 mA. EMSA was performed using the Chemiluminescent EMSA Kit (Beyotime, GS009). The biotin signs were detected using the Chemiluminescent EMSA Kit (Beyotime, GS009). The primer sequences used for EMSA is shown in Appendix Table S9.

## The lncRNA–protein interaction prediction

The coding probability of *lncMtDloop* was examined with the PhyloCSF analysis in the UCSC browser, Coding Potential Assessment Tool (CPAT, http://lilab.research.bcm.edu/cpat/index.php), and Coding Potential Calculator (CPC, http://cpc.cbi.pku.edu.cn/). The secondary structure of *lncMtDloop* was predicted on RNAfold WebServer (http://rna.tbi.univie.ac.at/cgi-bin/RNAfold.cgi) based on partition function and minimum free energy (MFE). The catRAPID was used for the prediction of protein–RNA binding propensity between *lncMtDloop* fragments and possible proteins (http://service.tartaglialab.com/page/catrapid.group). The soft and algorithms used in this study are shown in Appendix Table S10.

## Measure of extracellular fluxes using Seahorse XF96

About 20,000 primary neurons were seeded in each well of an XF 96-well cell culture microplate in 80 ml of culture media and incubated overnight at 37 °C in 5% $CO_2$. The four corners were left only with medium for background correction. The culture medium is replaced with 180 ml of bicarbonate-free DMEM and cells are incubated at 37 °C for 30 min before measurement. Oxygen consumption rates (OCR) were measured using an XF96 Extracellular Flux Analyzer (Yepez et al, 2018). OCR was determined at four levels: with no additions, and after adding: oligomycin (1 μM); carbonyl cyanide 4-(trifluoromethoxy) phenylhydrazone (FCCP, 0.4 μM); and rotenone (2 μM) (additives purchased from Sigma at highest quality). After each assay, manual inspection was performed on all wells using a conventional light microscope. Wells for which the median OCR level did not follow the expected order, namely, median $(OCR(Int_3))$ > median $(OCR(Int_1))$ > median $(OCR(Int_2))$ > median $(OCR(Int_4))$, were discarded (977 wells, 10.47%). Of note, we excluded from the analysis contaminated wells and wells in which the cells got detached (461 wells, 4.94%).

## Cytochrome c oxidase assay

The hippocampus was isolated from 12-month-old 3×Tg mice and subjected to Cytochrome c oxidase measurement using cytochrome c oxidase assay kits following the manufacturer's instructions. The colorimetric assay of the samples is based on the observation of the decrease in absorbance at 550 nm of ferrocytochrome c caused by its oxidation to ferricytochrome c by cytochrome c oxidase.

## Electrophysiology

LTP studies were performed on hippocampal slices from 12-month-old mice. Mice were anesthetized with isoflurane, decapitated, and brains were rapidly removed in pre-ice-cold ACSF (120 mM NaCl, 2.5 mM KCl, 1.25 mM $NaH_2PO_4$, 26 mM NaHCO3, 2 mM MgSO4, 10 mM D-glucose, 2 mM $CaCl_2$ with 95% O2, 5% $CO_2$ [pH 7.4]). Transverse hippocampal slices were cut at a thickness of 350 μm and placed on infusion chambers in ACSF. The fEPSPs were recorded from the CA1 stratum radiatum region by an extracellular borosilicate glass capillary pipette (resistances of 3–5 MΩ) filled with ACSF. Stimulation of Schaffer collaterals from the CA3 region by a bipolar electrode. Signals were amplified using a MultiClamp 700 B amplifier (Axon), digitized using a Digidata 1440 A Data Acquisition System (Axon) with 1 kHz low pass filter and 2 kHz high pass filter, and analyzed using Clampex

10.7 software. For each experiment, eight to nine sections from three to four animals/genotypes were used.

## Behavioral testing

For nest-building behavior, mice were individually housed with nesting materials of facial tissue strips. Mice were given the nesting material on day 1. Pictures were taken and nests were scored 24 h later. For the open-field test, mice were placed into the open-field arena (made of opaque white plastic material, 35 cm × 35 cm separated into 5 × 5 grids) by a blinded experimenter and allowed to explore the arena for 15 min. The number of crossing grids and percentage of time spent in the center (22 cm × 22 cm) were quantified using video tracking software. The novel object recognition test was performed on 3 consecutive days. On the first day, mice were placed in the center of an empty open box and allowed to explore for 10 min. On the second day, the mice were placed in the open box with two identical objects in the two corners and allowed to explore for 10 min. On the third day, the mice were placed in the same box but with an object replaced by a novel one and allowed to explore for 10 min. The discrimination index refers to the time spent exploring the novel object relative to the time spent exploring both objects. For the fear conditioning test, a fear memory was measured by pairing a conditioned stimulus with an unconditioned stimulus. We used the Video Freeze Conditioning 'Video Freeze' software to record freezing behavior. On day one, mice were placed inside the chambers; after the 120-s baseline, a 30-s CS tone followed, and a US foot shock was given during 120–150 s, 180–210 s, and 240–270 s. On day two, both contextual and cued phases were done. During phase one, mice were placed in the same testing chambers used on day one for 5 min. During phase two, 3 h later, mice were placed into modified chambers with plastic inserts, and after 300 s, the CS tine was played for 30 s at 1 min intervals (five CS tones in total). After 10 min, the mice were returned to their housing cages (Hou, 2018; Iaccarino et al, 2016). For the Morris water maze, the pool (150 cm diameter) is filled with water that is maintained at room temperature. A 12 cm diameter transparent platform was placed 1 cm below the water surface at the first quadrant. Mice were trained for 5 consecutive days, with four trials per day. Each training lasted 60 s until the mouse arrived on the platform. Twenty-four after the last training, the test was conducted without a platform for the 60 s to test their memory performance. The latency time to the area of the removed platform, the time spent in the first quadrant, and the number of crossing the area of the removed platform were recorded. The behavior was recorded and quantified using Ethovision Vision 10.1 software.

## AAV9 microinjections in mice

For bilateral injection of AVV9, sex-matched 11-month-old WT and 3xTg mice were anesthetized with isoflurane. Wiping 70% ethanol was used to sterilize the surgical area, remove the skin above the skull by using aseptic techniques, and drill small holes for the sites of injection. A volume of 0.5 μl of $8 \times 10^{12}$ gc particles of AAV9 per hemisphere was injected at a speed of $0.075\ \mu l\ min^{-1}$. The target sites for AAV injection were marked on the skull (AP −2.0 mm, ML ± 1.5 mm, DV −2.0 mm). After surgery, mice were housed under standardized conditions for 6 weeks.

## Tail-vein injections of AAV-PHP.eB in mice

For tail vein injection of AAV-PHP.eB, sex-matched 11-month-old WT and 3xTg mice were anesthetized with isoflurane. The tail of the mice was wiped with 70% ethanol. The 150 μL of $2 \times 10^{13}$ gc particles of AAV-PHP.eB diluted in PBS were injected into the tail vein by an insulin needle. After tail vein injection, mice were housed under standardized conditions for 6 weeks.

## Statistical analysis

All statistical analysis was performed using GraphPad Prism 7 software. Results are shown as mean ± SEM of three independent experiments. The significance between the two groups was analyzed by a two-tailed Student's test. Significance between three groups or more was analyzed by one-way ANOVA or two-way ANOVA by Tukey's or Dunnett's multiple comparisons test. In all figures, $^*P < 0.05$, $^{**}P < 0.01$, $^{***}P < 0.001$, $^{****}P < 0.0001$, a $P$ value of more than 0.05 was considered non-significant.

# Data availability

This study includes no data deposited in external repositories. All algorithms and software used in this study are publicly available and links are provided in the corresponding sections of the methods.

The source data of this paper are collected in the following database record: biostudies:S-SCDT-10_1038-S44318-024-00270-7.

# Peer review information

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

## Acknowledgements

We are grateful to Zhigang Yang from the College of Physics and Optoelectronic Engineering, Shenzhen University, for supplying the MitoESq-635 dye and thank Peng Xi and Zhaoyang Wu for the performance of STED. We also thank Yingqi Guo from Kunming Biological Diversity Regional Center for Instrument, Kunming Institute of Zoology, Chinese Academy of Science for her assistance in sample preparation and performing electron microscopy. These works were supported by grants from the National Natural Science Foundation of China (91649119 and 92049105), the Ministry of Science and Technology of China (2015CB755605), Peking University Health Science Center (BMU2019YJ001), and the Applied Basic Research Key Project of Yunnan (E039030401) to JL. This work was also supported by grants from the National Natural Science Foundation of China (NSFC 81821092) to LL and (NSFC 8240460) to WX. The work was also supported, in part, by grants from the following: The Hong Kong Research Grants Council (RGC)-General Research Fund (GRF) (PI: GRF16100219 and GRF16100718); Alzheimer's Association Research Fellowship (PI: AARF-17-531566); Collaborative Research Fund (CRF) (Co-I: C4033-19EF); CUHK-Improvement on Competitiveness in Hiring New Faculties Funding Scheme (PI: Ref. 133) and CUHK-School of Life Sciences Startup funding to H-MC.

## Author contributions

**Wandi Xiong**: Data curation; Validation; Investigation; Methodology; Writing—original draft; Writing—review and editing. **Kaiyu Xu**: Data curation; Software; Formal analysis; Validation; Investigation; Visualization; Methodology. **Jacquelyne Ka-Li Sun**: Resources; Data curation; Software; Formal analysis; Validation; Investigation; Visualization; Methodology; Writing—review and editing. **Siling Liu**: Formal analysis; Validation; Investigation; Visualization; Methodology. **Baizhen Zhao**: Data curation; Formal analysis; Validation; Investigation; Visualization; Methodology; Writing—review and editing. **Jie Shi**: Resources; Formal analysis; Validation; Visualization; Writing—review and editing. **Karl Herrup**: Resources; Visualization; Methodology; Writing—review and editing. **Hei-Man Chow**: Resources; Software; Formal analysis; Funding acquisition; Validation; Investigation; Visualization; Methodology; Writing—review and editing. **Lin Lu**: Resources; Data curation; Software; Funding acquisition; Validation; Investigation; Visualization; Writing—review and editing. **Jiali Li**: Conceptualization; Resources; Data curation; Software; Formal analysis; Supervision; Funding acquisition; Validation; Investigation; Visualization; Methodology; Writing—original draft; Project administration; Writing—review and editing.

Source data underlying figure panels in this paper may have individual authorship assigned. Where available, figure panel/source data authorship is listed in the following database record: biostudies:S-SCDT-10_1038-S44318-024-00270-7.

## Disclosure and competing interests statement

The authors declare no competing interests.

# Expanded View Figures

**Figure EV1.  *LncMtDloop* is an evolutionarily conserved mitochondrial lncRNA, related to Fig. 1.**                                                        ▶

(**A**) BLAST search and mapping of this sequence with the NONCODE and UCSC databases affirms that *lncMtDloop* (AC027613.1) originates from the mitochondrial DNA (mtDNA), specifically located at chrM: 15,356–16,294 bp. (**B**) Examination of the sequence and evolutionary conservation of *lncMtDloop* across various species at the genomic locus of the *lncMtDloop* gene. (**C**) PhyloCSF analysis confirms the non-coding nature of *lncMtDloop*. The visualization employs a color scheme where the sense strand is depicted in green, indicating possible individual open reading frames. Conversely, the antisense strand is shown in red, also with potential open reading frames. The X-axis signifies the location along the RNA nucleotide sequence, while the Y-axis represents the PhyloCSF score. (**D**) Evaluation of coding potential scores for both coding genes, such as *Actb*, *Gapdh*, *Atp5b*, *Cox IV*, and non-coding genes like *Hotair*, *Xist*, and *lncMtDloop*. This assessment encompasses human, Macaca mulatta, and mouse species. (**E**) Utilization of a coding potential assessment tool to determine the coding probability of specific genes—*Actb*, *Gapdh*, *Atp5b*, *Cox IV* as well as non-coding genes *Hotair*, *Xist*, and *lncMtDlo*op—in human, *Macaca mulatta*, and mouse species. (**F**) Prediction of the secondary structures of *lncMtDloop* based on minimum free energy (MFE). The color scale bar, characterized by shades of red, indicates predictions with a high level of confidence. Source data are available online for this figure.

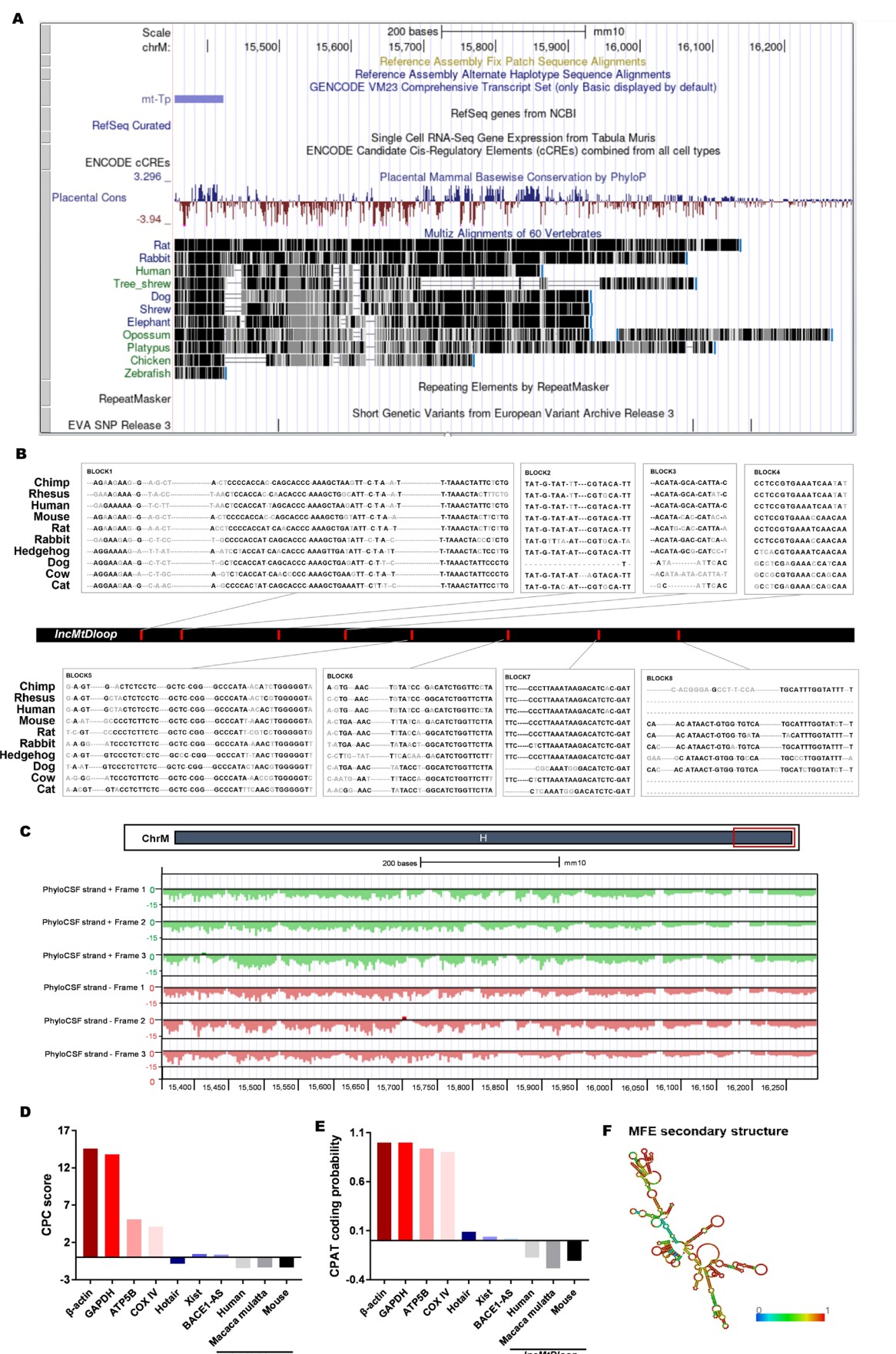

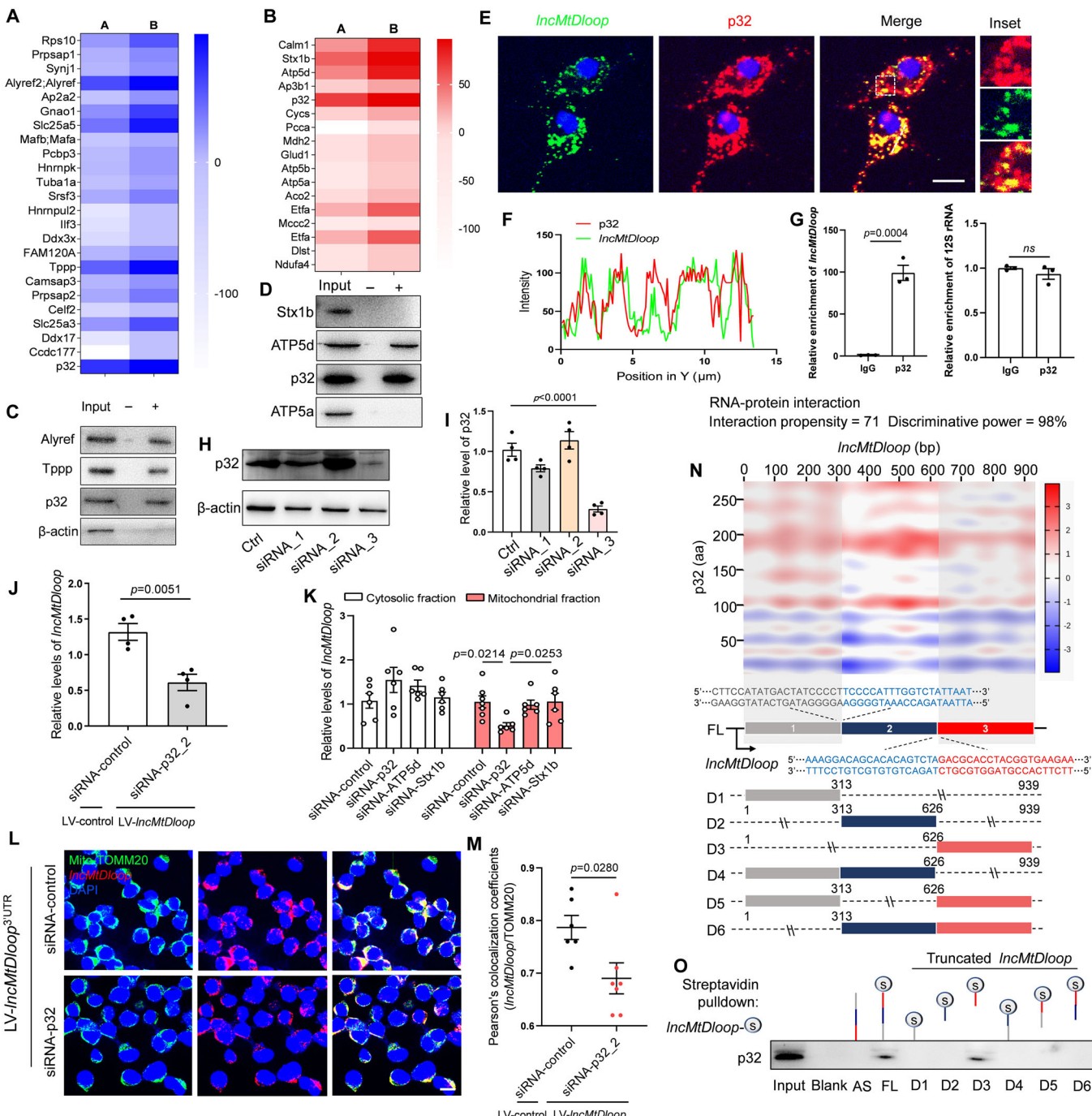

**Figure EV2.  Contribution of p32 to *lncMtDloop* localization within mitochondria, related to Fig. 2.**

(A) Predicted interaction propensity of *lncMtDloop* with binding partners in cytosolic preparations based on catRAPID, corresponding to mass spectrum results. The left column displays the interaction propensity scores of proteins, while the right column represents their discriminative power in a centesimal system. (B) Predicted interaction propensity of *lncMtDloop* with binding partners in mitochondrial preparations based on catRAPID, corresponding to mass spectrum results. The left column displays the interaction propensity scores of proteins, while the right column represents their discriminative power in a centesimal system. (C, D) RNA pull-down assays utilizing biotinylated *lncMtDloop* and antisense-*lncMtDloop* in cytosolic preparations (C) and mitochondrial (D), followed by western blot analysis of Alyref, Tppp, Stx1b, ATP5d, and p32. "+" represents *lncMtDloop* probe, and "−" represents anti-*lncMtDloop* control. (E) Representative images displaying the co-localization of *lncMtDloop* (green) and p32 (red) using RNAscope ISH and immunostaining in neurons. Scale bars, 10 μm. (F) Quantification of fluorescence signal intensities for *lncMtDloop* and p32 along the white dotted line in panel (E), measured using Image J. (G) RT-qPCR analysis for detectable *lncMtDloop* from RIP analysis of p32 antibody. 12 s RNA was used as control. Results were normalized to the control IgG RIP group. Bars represent mean ± SEM, with $n = 3$ independent experiments per group, unpaired $t$-test. (H, I) Knockdown of p32 in N2a cells using three siRNAs. The western blot analysis of p32 was performed. Bars represent mean ± SEM, $n = 3$ repetitions per group, unpaired $t$-test. (J) RT-qPCR analysis demonstrating the effect of p32 knockdown on *lncMtDloop* distribution in mitochondrial fractions. Bars represent mean ± SEM, with $n = 4$ repetitions per group, unpaired $t$-test. (K) RT-qPCR analysis revealing the impact of p32, ATP5d, and Stx1b knockdown on *lncMtDloop* distribution in mitochondrial fractions. N2a cells were infected with LV-*lncMtDloop* followed by siRNA-p32 transfection. Bars represent mean ± SEM, with $n = 6$ repetitions per group, one-way ANOVA with Dunnett's multiple comparisons. (L) Representative images showing the co-localization of *lncMtDloop* (red) and ATP5a (green) in N2a cells. LV-*lncMtDloop*-infected N2a cells were transfected with siRNA against p32. Scale bars, 10 μm. (M) Quantification of co-localization as observed in panel K). Bars represent mean ± SEM, with $n = 6$ regions of interest (ROI) per group, unpaired $t$-test. (N) Heatmap illustrating the interaction propensity of *lncMtDloop* binding to p32, as predicted by catRAPID. The X-axis indicates RNA nucleotide sequence location, while the Y-axis indicates protein residue location. Red shades indicate predictions with high interaction propensity. Bottom: Schematic map depicting the construction of six truncated *lncMtDloop* mutants to identify its protein-binding motifs. (O) In vitro transcription of the indicated six *lncMtDloop* truncates, followed by RNA pull-down assays using streptavidin-tagged RNA. Source data are available online for this figure.

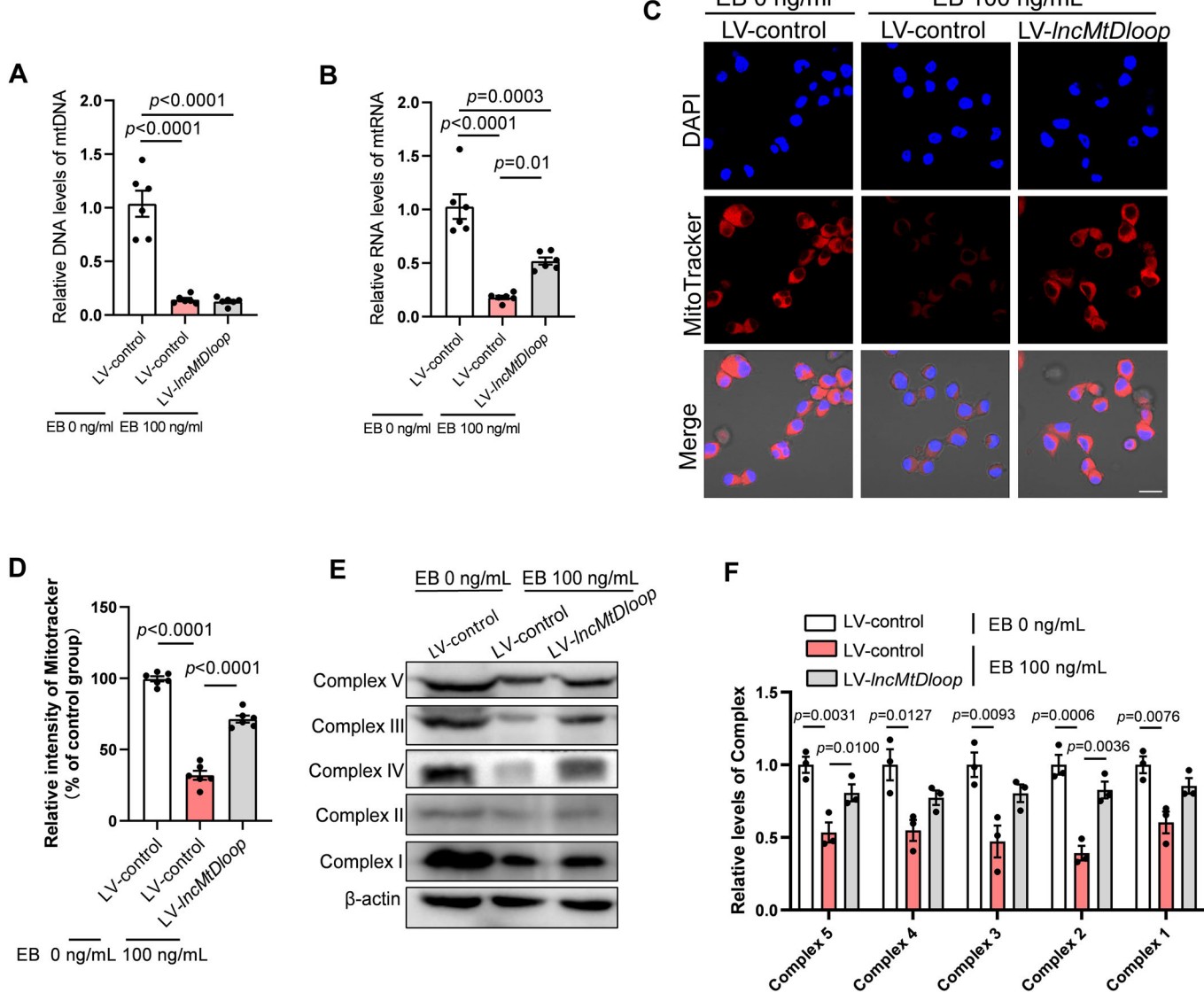

**Figure EV3. *lncMtDloop* promotes mtDNA transcription in mtDNA depletion assay, related to Fig. 3.**

(A) Determination of mtDNA copy number by qPCR. Error bars denote mean ± SEM, with $n = 6$ repetitions per group, one-way ANOVA followed by Dunnett's multiple comparisons test. (B) Determination of mtRNA copy number by RT-qPCR. Error bars denote mean ± SEM, with $n = 6$ repetitions per group, one-way ANOVA followed by Dunnett's multiple comparisons test. (C) Representative images depicting the MitoTrack signals in the N2a cell. Scale bars, 20 μm. (D) Quantification of fluorescence signal intensities for mitochondria. Error bars denote mean ± SEM, with $n = 6$ repetitions per group, one-way ANOVA followed by Dunnett's multiple comparisons test. (E) Western blot analysis depicting the levels of OXPHOS subunit proteins upon treatment with LV-*lncMtDloop* in primary cultured neurons w/o EB treatment. (F) Relative intensities of signals for OXPHOS subunit proteins. Error bars denote mean ± SEM, with $n = 3$ repetitions per group, one-way ANOVA followed by Dunnett's multiple comparisons test. Source data are available online for this figure.

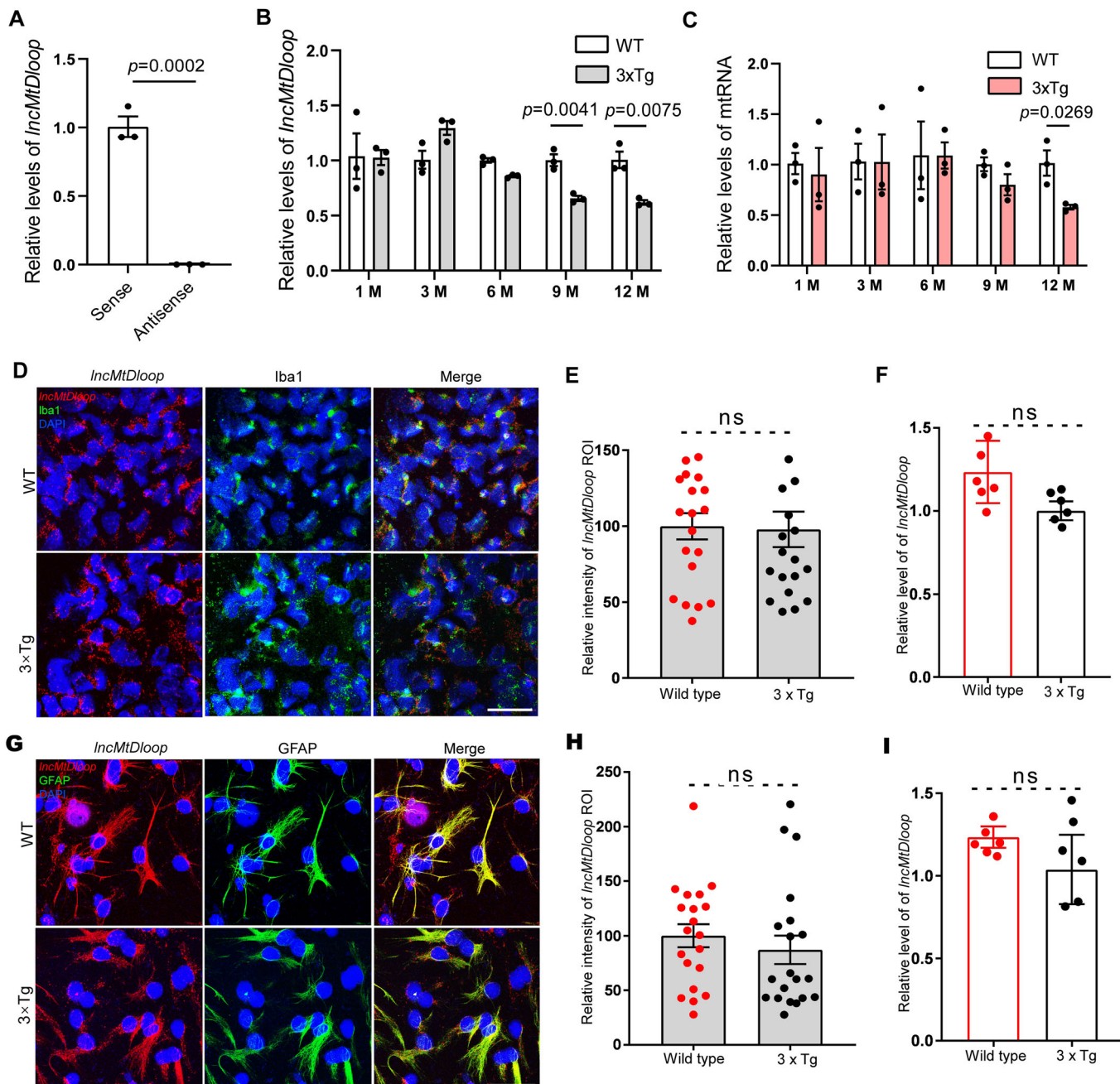

**Figure EV4.  Decreased expression of *lncMtDloop* differentially occurs in distinct neural cell types of AD, related to Fig. 4.**

(**A**) Determination of *lncMtDloop* levels by RT-qPCR. Antisense was used as control. Error bars denote mean ± SEM, with $n = 3$ repetitions per group, unpaired *t*-test. (**B**) Determination of *lncMtDloop* levels by RT-qPCR. Error bars denote mean ± SEM, with $n = 3$ mice per group, unpaired *t*-test. (**C**) Determination of mtRNA copy number by RT-qPCR. Error bars denote mean ± SEM, with $n = 3$ mice per group, unpaired *t*-test. (**D**) Representative images of RNAscope ISH and immunostaining showing few changes in levels of *lncMtDloop* (red) expression in 3xTg hippocampal microglia (Iba1, green) cultures at DIV14. Scale bars, 50 μm. (**E**) Relative intensities of *lncMtDloop* fluorescent signals are illustrated in (**D**). Bars indicate mean ± SEM, *n* (cells) = 17–19 per group. By unpaired *t*-test. (**F**) RT-qPCR analysis of *lncMtDloop* expression in wild type and 3xTg hippocampal microglia cultures at DIV14. Bars = mean ± SEM, $n = 6$ repetitions per group, unpaired *t*-test. (**G**) Representative images of RNAscope ISH and immunostaining showing few changes in levels of *lncMtDloop* (red) expression in 3xTg hippocampal astrocyte (GFAP, green) cultures at DIV14. Scale bars, 50 μm. (**H**) Relative intensities of *lncMtDloop* fluorescent signals are illustrated in (**G**). Bars indicate mean ± SEM, *n* (cells) = 20 per group. By unpaired *t*-test. (**I**) RT-qPCR analysis of *lncMtDloop* expression in wild type and 3xTg hippocampal astrocyte cultures at DIV14. Bars = mean ± SEM, $n = 6$ repetitions per group, unpaired *t*-test.

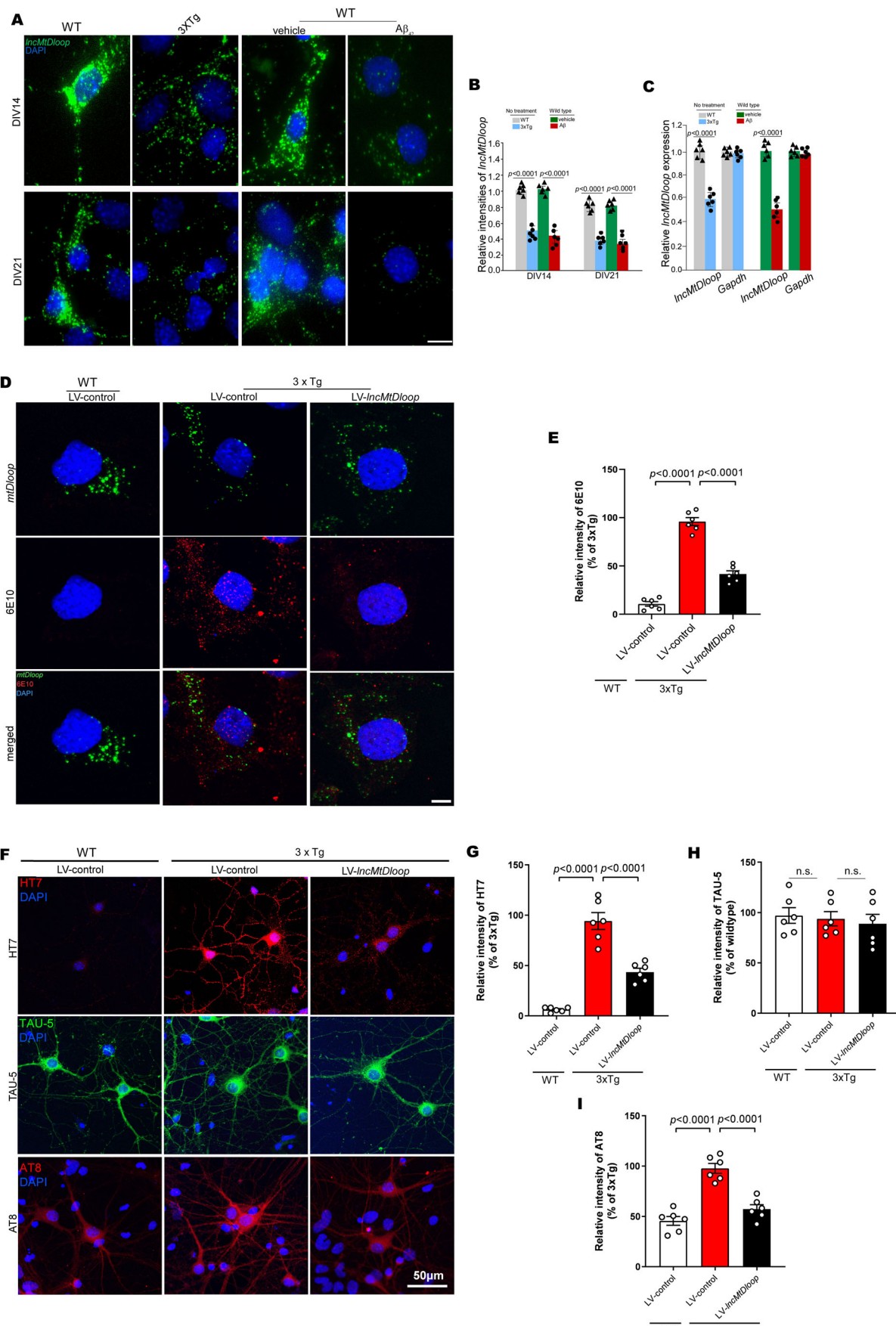

**Figure EV5.** *LncMtDloop* **expression is mutually related to AD pathology, related to Fig. 5.**

(A) Representative image of RNAscope ISH showing *lncMtDloop* expression. Wild-type (WT) and 3 × Tg primary hippocampal neurons at DIV14 and DIV21 were performed RNAscope ISH with mouse-specific probe sets against *lncMtDloop* with the treatment of Aβ$_{42}$. Scale bars, 10 μm. (B) Quantification of *lncMtDloop* fluorescent intensity illustrated in (A). Bars indicate mean ± SEM, $n = 6$ images per group, by unpaired *t*-test. (C) RT-qPCR analysis of *lncMtDloop* expression in hippocampal neurons of wild type and 3 × Tg at DIV14 with Aβ$_{42}$ treatment. Bars indicate mean ± SEM, $n = 6$ repetitions per group, by unpaired *t*-test. (D) Representative images of RNAscope ISH and immunostaining showing *lncMtDloop* expression (green) and Aβ production (red). Scale bars, 5 μm. WT and 3xTg primary hippocampal neurons were infected with LV-*lncMtDloop* or LV-control at DIV5, cells were prepared for tests at DIV14. (E) Quantification of fluorescence intensities of 6E10 signals illustrated in (D). Bars indicate mean ± SEM, $n = 6$ images per group, by one-way ANOVA with Dunnett's multiple comparisons test. (F) Representative IF images showing HT7, TAU-5, and AT8 in wild type and 3xTg primary hippocampal neurons. Wild type and 3xTg primary hippocampal neurons were infected with LV-*lncMtDloop* at DIV5, cells were collected and prepared for tests at DIV14. (G–I) Relative intensities of HT7, TAU-5, and AT8 fluorescent signals illustrated in (F). Bars indicate mean ± SEM, $n = 6$ images per group, one-way ANOVA with Dunnett's multiple comparisons test. Source data are available online for this figure.

