## [Peer Review File · The EMBO Journal]

The mitochondrial long non-coding RNA *IncMtloop* regulates mitochondrial transcription and suppresses Alzheimer's disease

Wandi Xiong, Kaiyu Xu, Jacquelyne Ka-Li Sun, Siling Liu, Baizhen Zhao, Jie Shi, Karl Herrup, Hei-Man Chow, Lin Lu, and Jiali Li

Corresponding author(s): Jiali Li (jiali.li@hmhn.org) , Hei-Man Chow (heimanchow@cuhk.edu.hk), Lin Lu (linlu@bjmu.edu.cn)

Review Timeline:

Submission Date:	26th Oct 23
Editorial Decision:	28th Nov 23
Revision Received:	25th Mar 24
Editorial Decision:	3rd May 24
Revision Received:	27th Aug 24
Accepted:	9th Sep 24

Editor: Kelly Anderson

Transaction Report:

Dear Dr. Li,

Thank you for submitting your manuscript for consideration by the EMBO Journal. It has now been seen by three referees whose comments are shown below.

Given the referees' positive recommendations, I would like to invite you to submit a revised version of the manuscript, addressing the comments of all three reviewers. I should add that it is EMBO Journal policy to allow only a single round of revision, and acceptance of your manuscript will therefore depend on the completeness of your responses in this revised version. It would be good to discuss your plan to address the referee concerns and I am available to do so in the coming weeks by email or zoom.

Thank you for the opportunity to consider your work for publication. I look forward to your revision.

Yours sincerely,

Kelly M Anderson, PhD
Editor, The EMBO Journal
k.anderson@embojournal.org

We realize that it is difficult to revise to a specific deadline. In the interest of protecting the conceptual advance provided by the work, we recommend a revision within 3 months (26th Feb 2024). Please discuss the revision progress ahead of this time with the editor if you require more time to complete the revisions. Use the link below to submit your revision:

Referee #1:

In this paper, Xiong et al. studied the role of IncMtDloop in controlling mitochondrial homeostasis in neuronal cells. Complementary to this, the authors showed some partial evidence that this lncRNA molecule (originating from mtDNA) contributes to cognitive decline, neurodegeneration, and mitochondrial dysfunction in an AD-related mice model. Most of the findings discussed in this paper are interesting and, without question, represent an interesting advance in understanding mitochondrial function regulation in mammals. However, in the present form, this manuscript requires additional studies and effort to be considered for publication in the EMBO Journal.

General comments

1. The evidence that relates the possible contribution of IncMtDloop with mtDNA regulation seems very solid, as the authors showed studies in different species. However, IncMtDloop contribution to mitochondrial dynamics regulation and AD-neurodegeneration is unclear and requires additional data.

2. The manuscript's abstract is unclear and requires re-writing. The critical elements of IncMtDloop are not included, and information regarding Alzheimer's disease is unclear.

3. In the manuscript, the authors presented several sentences establishing a certain novelty of the studies discussed and references's interpretations that are not correct:

-In the introduction, the authors novelty propose the possible interplay between TFAM and neurodegeneration shown in AD. However, several manuscripts already suggest this train of thought.

-This sentence: "A reciprocal relationship has been proposed to exist between the severity of AD-related pathologies and the extent of mitochondrial dysfunction (Kerr et al., 2017; Kingwell, 2019)" should be revised. Mitochondrial dysfunction has been extensively showed in AD. However, there is still no clear evidence that a progressive level of mitochondrial impairment is correlated with AD severity.

-In this sentence: "A discernible reduction in IncMtDloop was observed in the brain tissues of both AD patients and the 3xTg mice, aligning with the trajectory of Alzheimer's pathogenesis". The data presented in the manuscript have not demonstrated this effect.

-In this sentence: "While the connection between malfunctions in nuclear genome-encoded genes responsible for mitochondrial function and brain aging is well-established." The authors should revise this statement. There are still some elements that need to be found to explain how mitochondrial function could be affected by aging.

4. A careful read of the manuscript showed several sentences that are not referenced. Also, several grammar and syntax mistakes must be corrected.

5. It is curious why the authors presented mitochondrial dynamics changes induced by IncMtDloop in the AD mice model without showing expression /activity of mitochondrial dynamics regulators (Dlp1-Fis-1; Mnf1/2, Opa1).

Figure's comments

Fig1. Immunofluorescence studies require cell morphology controls. Also, the authors must include additional data using another mitochondrial protein control to solidify their observations.

Also, it is mandatory to present a complete colocalization analysis using Manders or any other factor calculation to discuss the

differences shown in these figures appropriately. This criterion must be included in all immunofluorescence studies.

Fig. 2F. Immunofluorescence controls for cell morphology must be provided.

Fig. 2D. This figure lacks mitochondrial extracts and proper controls. The authors must include western blot studies for VDAC or other mitochondrial quality control.

Fig 3E. The authors must provide full-resolution western blot images.

Fig. 4A. Immunofluorescence controls for cell morphology must be provided.

Fig. 4K-L. It is not clear to this reviewer (see graph) how the authors calculated their experimental n? What we see in the graph represents independent experiments? .

Fig. 4M. Western blot images must be improved. The images are not clear and look pixelated.

Fig. 5H. To appropriately quantify mitochondrial fission/fusion events, the authors must set a mitofusion assay using another methodology, like mito-Kendra construct expression. Using Mito-Dendra, it is possible to adequately measure all mitochondrial fusion/fission events simultaneously. Also, it is imperative to include another mitochondrial morphology control.

Fig. 5N. How the authors interpret OCR data is intriguing. If we observe, the curves for both conditions do not show significant differences. However, the quantification and analysis of mitochondrial basal capacity show statistical differences between both conditions. This issue must be addressed.

Fig. 7. A. A control for AAV infection in mice tails must be provided.

Fig 7K. The authors must evaluate these findings because it is unclear whether significant differences in LTP activity are seen.

Referee #2:

Comments on Xiong et al „The mtDNA-derived IncMtDloop promotes mitochondrial homeostasis maintenance and implications in AD”

In their manuscript entitled "The mtDNA-derived IncMtDloop promotes mitochondrial homeostasis maintenance and implications in AD" Xiong et al characterize the function of a long non-coding RNA (lncRNA), previously identified in the same lab to be relevant for brain development and aging in rhesus macaques. Here, the lncRNA was shown to originate from the mitochondrial D-loop region, the only regulatory region contained within mtDNA, and aptly renamed IncMtDloop. The authors propose that IncMtDloop interacts with the mitochondrial transcription factor A (TFAM), thereby enhancing mitochondrial gene expression. Furthermore, they observe decreased expression of IncMtDloop in the brains of human Alzheimer's patients and mouse brains of an AD model. By introducing allotropic IncMtDloop into neuronal cultures and mice, the authors observed a significant improvement in mitochondrial homeostasis and morphology as well as cellular and organismal well-being.

This study targets a very interesting - and currently underappreciated - aspect of mitochondrial biology, i.e. the potential regulation of mitochondrial gene expression by non-coding RNAs. Interestingly, a recent study identified the well-known non-coding mitochondrial 7S RNA as a key regulator of mitochondrial gene expression, as it regulates feedback inhibition of the mitochondrial RNA polymerase (Zhu et al, 2022; PMID: 35662414). Thus, further identification of related ncRNA species and their function is of great significance. In addition, IncMtDloop might show clinical relevance in AD models. The authors provide a large amount of various data sets of high quality. However, in its current state, the study lacks experimental evidence to fully support the claims made here and move beyond correlation. One major concern is the lack of detailed, appropriate assays to indeed prove a direct function of IncMtDloop in the regulation of mitochondrial gene expression. It cannot be excluded that all in vitro and in vivo observations are due to other cellular effects, especially given the fact that a large number of other potential interaction partners was identified for IncMtDloop. Thus, clear evidence is lacking on how specifically the expression of this lncRNA may rescue the highly complex aberrations found in AD models.

The study proposes a very interesting finding and deserves publication, thus I recommend major revision to address the detailed comments found below.

Major concerns:

IncMtDloop

In spite of its high importance in this study, IncMtDloop itself is rather poorly characterized. Information on its size and sequence

is largely based on sequencing data and bioinformatic predictions alone. More biochemical evidence is required.

- A number of other lncRNAs have been described in mitochondrial DNA (lncND5, lncND6 and lncCytb, Rackham et al, 2011, PMID:22028365). Similar to what is known about them, the authors need to include information on the sequence, especially 3' and 5' transcript ends (e.g. based on PARE or PolyA analyses) of lncMtDloop and clearly depict the position of this lncRNA in the D-loop region. To compare the abundance of mitochondrial lncRNAs, northern blots of lncMtDloop in comparison to the other lncRNAs are required. Those should be probed for both sense and antisense RNA to get an idea of their respective abundance.
- More information on the potential generation of lncMtDloop is required. Based on their scheme, lncMtDloop appears to be generated after termination of H-strand transcription which is usually terminated before by yet unknown mechanisms. Interestingly, a previous study (Jemt et al, 2015; PMID: 26253742; which should be included in the discussion) has already described an antisense D-loop transcript generated upon recovery from mtDNA depletion, a state in which H-strand termination is reduced. The authors need to clarify if these two transcripts are the same or at least similar. In line with this, the study would benefit from including a similar mtDNA depletion/recovery assay. Furthermore, it is interesting to know how this lncRNA species may be processed - do the transcript ends contain punctuation elements? Did the authors observe an interaction with mtRNA processing enzymes such as MRPP1, MRPP3 or ELAC2 (to name a few) or the key RNA binding protein LRPPRC in their RNA-IP studies? The authors should discuss this point.
- The co-localisation between lncMtDloop and Mito-ATP5 appears rather poor in primary neurons (Figure 1B). Large extramitochondrial portions can be observed, the authors should comment on what they might represent. Given their hypothesis regarding regulation of mitochondrial transcription, it would be interesting to see if the lncMtDloop colocalizes with active mitochondrial nucleoids. In situ hybridization in combination with BrU labelling and nucleoid staining will help to quantify the correlation between lncMtDloop and mtRNA synthesis.

TFAM interaction and recruitment

- The authors employ RNA pulldown assays to identify potential binding partners of lncMtDloop, using the respective antisense RNA as a control. They manage to pulldown a large quantity of proteins, both in cytosolic and mitochondrial fractions. Curiously, the authors focus on mitochondrial transcription, although other interactions partners corresponding to other pathways, e.g. cellular respiration, were more highly enriched. Please clarify this rationale. Furthermore, the authors observe a strong interaction with key cytosolic proteins. A key question that strongly relates to the in cellulo and in vivo observations remains: How do the authors rule out that lncMtDloop effects on other pathways are responsible for the observed changes?
- The data on a direct lncMtDloop:TFAM interaction suffers from a lack of adequate assays and controls. TFAM, the major mitochondrial packaging protein, binds mtDNA specifically and non-specifically. However, TFAM was shown to bind more complex RNA structures such as 4-way junctions and mitochondrial tRNAs, albeit at a lower affinity (Brown et al, 2015; PMID: 26545237). It is thus not surprising that TFAM might pull-down certain, more complex RNA species. Furthermore, this might explain the observation with truncated constructs - do the truncations lead to the loss of secondary structure? To fully appreciate the proposed interaction, it is necessary to know other mitochondrial RNA species detected after pull-down. When performing qRT-PCR after TFAM pull-down, what were the detection levels of e.g. mitochondrial mRNAs, tRNAs, 7S RNA or the other mitochondrial lncRNAs in comparison?
- The EMSA data is also lacking, as only overexpression of lncMtDloop was included. The authors need to perform EMSA assays in which the concentration of lncMtDloop is titrated. As controls, the same assay needs to be performed in the presence of 7S RNA, other mitochondrial lncRNAs and dsDNA.
- A direct effect of lncMtDloop on mitochondrial transcription cannot be concluded from the qPCR data alone, as the RNA species binds numerous targets within mitochondria, all of which might generate secondary effects on mitochondrial gene expression. It is absolutely necessary to perform in vitro transcription assays in the presence of RNA to substantiate the main hypothesis, similar to what was done for 7S RNA (Zhu et al, 2022). Assays need to be performed using both HSP and LSP templates to conclude a global effect. Furthermore, assays need to be performed titrating concentrations of lncMtDloop as well as control RNA (e.g. antisense), other lncRNAs and 7S RNA as controls.

RNA import assay

The authors utilize allotropic expression of lncMtDloop to increase cellular levels of their target RNA species. It should be noted that RNA import into mitochondria is a very controversial topic that has only been shown in a few selected studies in very limited circumstances.

- The authors should include other mitochondrial lncRNAs as controls when performing their PNPase pull-down to assess targeting abundance, they should have a similar hairpin and be targeted.
- As the authors state, the RNA import assay employing the MRPS12 UTR directs lncMtDloop to the mitochondrial surface. This does not necessarily mean that it gets imported into the mitochondrial matrix which is a prerequisite for the proposed mode of action. Here, RNA gets amplified from mitochondrial fractions, thus it might well be on the outer surface. The authors need to prove matrix localization of their lncMtDloop, e.g. by coupling RNase digest on isolated, intact mitochondria to subsequent RNA isolation and northern blotting.

AD model systems

Interestingly, the authors observe a clear decrease in the levels of lncMtDloop in various AD models, pointing to a clinical relevance of their observations. Furthermore, the authors provide an extensive panel of high-quality data in cellular systems and in vivo showing that morphological and functional alterations of mitochondria as well as physiological parameters can be rescued by the expression of lncMtDloop. The main problem here is that it remains unclear if the observed effects are due to a direct effect on mitochondrial gene expression.

- While there is a clear effect on mitochondrial respiration and morphology, this might be due to an uncharacterized function of

IncMtDloop. In their initial RNA-IP some of the major hits in mitochondria such as cellular respiration were much more significantly enriched. The authors detect an effect on Cytochrome C oxidase, how do they exclude that there is no direct effect of IncMtDloop on cellular respiration? Were perhaps Complex IV assembly factors pulled down? Disturbances in OXPHOS frequently lead to changes in cristae and mitochondrial morphology, similar to what was observed here. Furthermore, the data sets shown do not provide clear evidence that IncMtDloop gets imported into the mitochondrial matrix to be able to exert a function in mitochondrial gene expression. Thus, it is possible that IncMtDloop, attached to the mitochondrial surface, might interact with some of the cytosolic partners observed in the RNA-IP, exerting effects on mRNA metabolism and cytoplasmic translation (categories enriched). To address these concerns, the authors should re-evaluate their RNA-IP for a potential direct modulation of cellular respiration and consider this possibility in their conclusions and discussion.

- The authors observe a decrease in the transcript levels of mitochondrial mRNAs and correlate this with the observed mitochondrial and cellular dysfunctions. However, it has been shown that depletion of mitochondrial RNAs, even to about 30% of controls, is well-tolerated in vivo (Lagouge et al, 2015; PMID: 26247782). Similarly, neurons tolerate disturbances of mitochondrial dysfunction for a very long time in vivo (Sorensen et al, 2001; PMID: 11588181). How do the authors reconcile their findings with these observations? Western blotting of OXPHOS protein levels is needed to determine the protein levels of mtDNA-encoded/OXPHOS subunits. Is the rate of mitochondrial transcription increased after overexpression of the IncMtDloop as a compensatory response? This can be analyzed by e.g. in organello transcription in cell lines. Their qPCR data argues for a Complex IV deficiency which is also observed in their activity assay - is Complex I also disturbed?
- Is the decrease in IncMtDloop in AD models consistent with a general decrease in mtDNA levels? It would be interesting to know if a stable ratio of mtDNA:IncRNA is maintained or if the ratio is shifted over time, correlating with pathology.

Referee #3:

Xiong et al. report a new long non-coding mitochondrial DNA that shows to be a critical regulator of mitochondrial homeostasis through regulation of TFAM, the major mitochondrial transcriptional factor, and with relevance in an Alzheimer's disease model. The research is well designed and both the recovery of mitochondrial network dynamics and pathological features in 3xTg AD mice after IncMtDloop expression are convincing. However, the 3xTg model is a very AD aggressive model and it would be interesting to understand the mtDNA transcription phenotype in early stages of the disease where a treatment strategy is likely to be more efficient. Moreover, the reader would benefit of a more detailed description of results (e.g. behavior) and detailed methods and analysis.

Major comments:

- The authors show no changes in mtRNA expression in macaque and mice during aging. However, I wonder if that is true in the disease model. Several studies show increased transcription of mtDNA genes in early AD/MCI patients and pre-symptomatic AD models (e.g., PMID: 15075441, 27793643, 37907591). This hyper mitochondrial metabolism may be a compensatory mechanism at early stages. It would be very interesting to see if young/early stage 3xTg show any changes in IncMtDloop and mtRNA levels and when this phenotype initiates.
- In figure 5G, the authors show some representative STED pictures of mitochondrial morphology but is not easy to understand if the analysis in K and L was done using STED or confocal. By reading the text it seems it was done using STED but pictures in 5H seem to have confocal resolution. The fig 5H would also benefit of having a zoom in of the fission event along time - it's difficult to follow any event along time in the representative pictures shown.
- Methods should state how fission events were calculated (e.g. automatized or manual quantification) and in which region of the neurons (soma only?). Same for TEM images. How did the authors selected the cut off to what is tubular and short tubular?
- The results description about memory performance show be better described/explained. E.g., at some point the authors say "contextual and cued fear conditioning text was carried out." And then continue to MWM description without saying the results of the previous text.

Minor comments:

- Please add references to the sentences: (page 5) "(...) a common structure among well-documented lncRNAs"; (page 6) "Mitochondrial genome homeostasis (...) in metabolic disorders, aging and neurodegenerative diseases."
- In page 13, please rephrase the sentence: "To explore whether the loss of IncMtDloop is connected to decreased mtDNA content and reduced mtRNA expression." - it seems unfinished.
- Tubular in Fig 6G, H is misspelled.
- It's difficult to say that the organelle pointed in Fig. 6D is a mitochondrion. There is no double membrane or visible cristae and it's darker than normal. It could be a lysosome.
- In page 15 authors say that injections of AAV-PHP.eB-IncMtDloop-3UTRMRPS12 were done through mouse tail vein but in Fig S2 it seems it was injected in hippocampus.
- Mitophagy modulators would fit better associated with peripheral fission events described in Fig 5.

- In Fig 7B add mention to the staining used.
- Fig 7E-H would benefit of having a title/legend for hippocampus vs PFC.
- No ethics are described for mice experiments.
- Software used in MWM analysis?

We highly appreciate the possibility to submit a revised version of our manuscript to *The EMBO Journal*. We are also grateful for the positive response and constructive feedback that we received from the referees. Their insightful comments helped us to substantially strengthen the manuscript and we hope that you will find it suitable for publication in *The EMBO Journal*. Please see below for our point-by-point response to the issues raised.

Referee #1:

In this paper, Xiong et al. studied the role of lncMtDloop in controlling mitochondrial homeostasis in neuronal cells. Complementary to this, the authors showed some partial evidence that this lncRNA molecule (originating from mtDNA) contributes to cognitive decline, neurodegeneration, and mitochondrial dysfunction in an AD-related mice model. Most of the findings discussed in this paper are interesting and, without question, represent an interesting advance in understanding mitochondrial function regulation in mammals. However, in the present form, this manuscript requires additional studies and effort to be considered for publication in the EMBO Journal.

General comments

1. The evidence that relates the possible contribution of lncMtDloop with mtDNA regulation seems very solid, as the authors showed studies in different species. However, lncMtDloop contribution to mitochondrial dynamics regulation and AD-neurodegeneration is unclear and requires additional data.

Response: Thank the referee for your valuable suggestion. The requested data has been added accordingly (Fig. EV7). Indeed, our data demonstrates that levels of lncMtDloop were significantly lower among the 3xTg primary neurons, as well as when WT neurons were exposed to exogenous A β 42. Furthermore, ectopic expression of lncMtDloop in the same 3xTg-derived culture reduced the accumulation of A β and the levels of

phosphorylated tau, indicating a potential interplay between lncMtDloop and A β pathologies.

2. The manuscript's abstract is unclear and requires re-writing. The critical elements of lncMtDloop are not included, and information regarding Alzheimer's disease is unclear.

Response: We greatly appreciate the referee's comprehensive review. We have revised the abstract and are confident that it is now clearer.

3. In the manuscript, the authors presented several sentences establishing a certain novelty of the studies discussed and references's interpretations that are not correct:

-In the introduction, the authors novelty propose the possible interplay between TFAM and neurodegeneration shown in AD. However, several manuscripts already suggest this train of thought.

Response: Thank you for the referee's diligent review. While several manuscripts have already suggested the interplay between TFAM and neurodegeneration in Alzheimer's disease (AD), there is limited literature exploring the regulatory mechanisms by which TFAM modulates mitochondrial function and contributes to neurodegenerative disorders. In our introduction, we have highlighted our renaming of AC027613.1 as "lncMtDloop" and elucidated its significant roles in orchestrating TFAM-dependent mtDNA transcription and maintaining overall mitochondrial homeostasis.

-This sentence: "A reciprocal relationship has been proposed to exist between the severity of AD-related pathologies and the extent of mitochondrial dysfunction (Kerr et al., 2017; Kingwell, 2019)" should be revised. Mitochondrial dysfunction has been extensively showed in AD. However, there is still no clear evidence that a progressive level of mitochondrial impairment is correlated with AD severity.

Response: We appreciate the referee's suggestion. In the revised version, we have rewritten sentences accordingly.

-In this sentence: "A discernible reduction in IncMtDloop was observed in the brain tissues of both AD patients and the 3xTg mice, aligning with the trajectory of Alzheimer's pathogenesis". The data presented in the manuscript have not demonstrated this effect.

Response: We value the referee's suggestion. In the revised version, we have deleted sentences and adjusted IncMtDloop's connection with AD pathogenesis in the discussion.

-In this sentence: "While the connection between malfunctions in nuclear genome-encoded genes responsible for mitochondrial function and brain aging is well-established." The authors should revise this statement. There are still some elements that need to be found to explain how mitochondrial function could be affected by aging.

Response: We appreciate the referee's suggestion, and in response, we have revised the sentences with adding several new literatures.

4. A careful read of the manuscript showed several sentences that are not referenced. Also, several grammar and syntax mistakes must be corrected.

Response: Thank you to the referee for your meticulous review. We apologize for any mistakes in our manuscript. We have thoroughly revised the sentences and grammar throughout the paper accordingly.

5. It is curious why the authors presented mitochondrial dynamics changes induced by IncMtDloop in the AD mice model without showing expression /activity of mitochondrial dynamics regulators (Dlp1-Fis-1; Mnf1/2, Opa1).

Response: Thank you to the referee for your thorough review. In response, we have conducted western blot analysis to illustrate the expression of mitochondrial dynamics proteins, including Drp1 (dynamin-related protein 1) and Opa1 (optic atrophy type 1) accordingly (Fig 5N-P).

Figure's comments

Fig1. Immunofluorescence studies require cell morphology controls. Also, the authors must include additional data using another mitochondrial protein control to solidify their observations.

Response: Thank you to the referee for your careful review. We conducted RNAscope *in situ* hybridization to visualize the expression signal of *lncMtDloop*, which is specific and highly sensitive for detecting *lncMtDloop* signals. Subsequently, we performed immunofluorescence staining to verify the colocalization of *lncMtDloop* and mitochondria.

Regarding the cell morphology controls for the immunofluorescence studies, we encountered some confusion. If it pertains to using bright-field (phase contrast microscope) imaging or labeling the cells with another marker, we would like to clarify that due to parameter setting issues of microscopy, we regretfully did not save the bright-field images of immunofluorescence. We refer to several published papers related to ncRNAs (Liu et al., *Nature Metabolism*, 2021; Wu et al., *Science*, 2021; Zhao et al., *Cell*, 2020), as well as our previous publications (Xu et al., *Nature Communications*, 2020; Liu et al., *Genome Research*, 2017), none of which include phase contrast images. However, to demonstrate cell morphology, we labeled the neurons with MAP2 (a neuronal marker) and combined this with RNAscope ISH (shown in the low-power field to exhibit the overall morphology of the neuron) (as shown in the revised Fig 4K).

Additionally, to strengthen our findings, we designed an antisense probe as a negative control and selected COXIV as another mitochondrial protein control. Our results indicate minimal signal from the antisense probe and little or no merged signals with

the mitochondrial marker. In contrast, IncMtDloop displayed clear merged signals with the COX IV marker shown as below.

Also, it is mandatory to present a complete colocalization analysis using Manders or any other factor calculation to discuss the differences shown in these figures appropriately. This criterion must be included in all immunofluorescence studies.

Response: Thank you to the referee for your careful review. We have incorporated various colocalization analyses in the figures with distinct experimental purposes. To visualize the fluorescence signal of colocalization, we utilized ImageJ to calculate the intensity based on dashed white lines, as described in previous publications (Zhao et al., Cell, 2020; Liu et al., Nature Metabolism, 2021). Examples of this approach can be seen in Fig 1B-G, Fig 3F-G, and Fig EV2E in the revised version. Additionally, to quantify the co-occurrence correlation among different experimental groups, we employed ImageJ to quantify the intensity-based Pearson's colocalization coefficient, as demonstrated in Fig EV2K-L and Fig S1A-B.

Fig. 2F. Immunofluorescence controls for cell morphology must be provided.

Response: Thank you to the referee for your careful review. Regarding Figure 1, unfortunately, we did not save the bright-field images of immunofluorescence due to issues with the microscopy parameters. In reference to various published papers related to ncRNAs (Liu et al., *Nature Metabolism*, 2021; Wu et al., *Science*, 2021; Zhao et al., *Cell*, 2020) and our previously published papers on ncRNAs (Xu et al., *Nature Communications*, 2020; Liu et al., *Genome Research*, 2017), none of them include phase contrast images. However, to depict cell morphology using an alternative method, we labeled the neurons with MAP2 (a neuronal marker) and combined this with RNAscope ISH, as shown in the low-power field to exhibit the overall morphology of the neuron (as depicted in the above).

Fig. 2D. This figure lacks mitochondrial extracts and proper controls. The authors must include western blot studies for VDAC or other mitochondrial quality control.

Response: Thank the referee for your careful review. The requested VDAC1 as a quality control has been performed accordingly (Fig 2E and 3D)

Fig 3E. The authors must provide full-resolution western blot images.

Response: Thank the referee for your careful review. The requested full-resolution WB (revised Fig 2E) images has been provided in the Source Data as well as below accordingly.

Fig. 4A. Immunofluorescence controls for cell morphology must be provided.

Response: Thank the referee for your careful review. Regarding Fig 4A, the situation parallels that of Figure 1. While we recognize the potential value of incorporating cell

morphology data to strengthen our findings, we unfortunately encountered challenges in retaining the bright-field images of immunofluorescence due to issues with the microscopy parameters. Additionally, integrating immunohistochemistry (IHC) and RNAscope in situ hybridization, especially in human paraffin brain slices, poses methodological hurdles. Despite our diligent efforts, we were unable to achieve the desired qualitative results in this regard. Nevertheless, we believe that leveraging lower magnitude data, as depicted in the revised Fig 4A, may help to partially address the absence of detailed cell morphological data. We hope the referee will appreciate the rationale behind our approach.

Fig. 4K-L. It is not clear to this reviewer (see graph) how the authors calculated their experimental n? What we see in the graph represents independent experiments?

Response: Thank the referee for your meticulous review. As indicated in the figure legends, Figure 4K serves as the representative image from RNAscope ISH, illustrating *IncMtDloop* expression in wild-type (WT) and 3×Tg primary hippocampal neurons at DIV14. Figure 4L quantifies the fluorescent intensity of *IncMtDloop* as depicted in Figure 4K, measured using Image J software. Figure 4M showcases the Northern blot images displaying *IncMtDloop* transcripts in primary hippocampal neurons of wild type and 3×Tg at DIV14. Finally, Figure 4N presents the relative intensity results corresponding to the Northern blot findings in Figure 4M.

Fig. 4M. Western blot images must be improved. The images are not clear and look pixelated.

Response: Thank the referee for your careful review. Indeed, the images are not western blot images but rather northern blot RNA images. We have now added clearer images to replace the previous ones as well as full-length gel images in the source data accordingly.

Fig. 5H. To appropriately quantify mitochondrial fission/fusion events, the authors must

set a mitofusion assay using another methodology, like mito-Kendra construct expression. Using Mito-Dendra, it is possible to adequately measure all mitochondrial fusion/fission events simultaneously. Also, it is imperative to include another mitochondrial morphology control.

Response: Thank the referee for your careful review. Fig. 5H (the revised Figure S2B) and Fig.5I comprises time-lapse images captured over a 5-minute duration in living primary neurons. We utilized the MitoESq-635 probe, developed by the Xi laboratories, which is well-suited for long-term, high-resolution STED nanoscopy (as documented in Yang et al., Nature Communications, 2020; Liu et al., Nature, 2017), to investigate the dynamic changes in mitochondrial fusion and fission over time. It's important to note that other probes or constructs are prone to quenching and are not suitable for long-term imaging under STED conditions.

Response: Thank for the referee's careful review. We have thoroughly re-examined the data and quantification methods and have confirmed their accuracy (revised Fig. 5R). The original data and analysis method are provided below for reference.

Fig. 7. A. A control for AAV infection in mice tails must be provided.

Response: Thank the referee for your careful review. Indeed, we employed AAV-PHP.eB vectors carrying allotropic IncMtDloop fused with the 3'UTR of MRPS12 (designated as SYN: AAV-PHP.eB-IncMtDloop-3'UTRMRPS12). As a control, AAV-PHP.eB vectors carrying an empty vector fused with the 3'UTR of MRPS12 were utilized (designated as SYN: AAV-PHP.eB-vector-3'UTRMRPS12).

Fig 7K. The authors must evaluate these findings because it is unclear whether significant differences in LTP activity are seen.

Response: Thank the referee for your careful review. We agree that the apparent lack of trend in the data may be attributed to normalization to a baseline, which can obscure the pattern. Nonetheless, upon reevaluation of the data and quantification methods, we have confirmed their accuracy.

Referee #2:

Comments on Xiong et al „The mtDNA-derived lncMtDloop promotes mitochondrial homeostasis maintenance and implications in AD"

In their manuscript entitled "The mtDNA-derived lncMtDloop promotes mitochondrial homeostasis maintenance and implications in AD" Xiong et al characterize the function of a long non-coding RNA (lncRNA), previously identified in the same lab to be relevant for brain development and aging in rhesus macaques. Here, the lncRNA was shown to originate from the mitochondrial D-loop region, the only regulatory region contained within mtDNA, and aptly renamed lncMtDloop. The authors propose that lncMtDloop interacts with the mitochondrial transcription factor A (TFAM), thereby enhancing mitochondrial gene expression. Furthermore, they observe decreased expression of lncMtDloop in the brains of human Alzheimer's patients and mouse brains of an AD model. By introducing allotropic lncMtDloop into neuronal cultures and mice, the authors observed a significant improvement in mitochondrial homeostasis and morphology as well as cellular and organismal well-being.

This study targets a very interesting - and currently underappreciated - aspect of

mitochondrial biology, i.e. the potential regulation of mitochondrial gene expression by non-coding RNAs. Interestingly, a recent study identified the well-known non-coding mitochondrial 7S RNA as a key regulator of mitochondrial gene expression, as it regulates feedback inhibition of the mitochondrial RNA polymerase (Zhu et al, 2022; PMID: 35662414). Thus, further identification of related ncRNA species and their function is of great significance. In addition, lncMtDloop might show clinical relevance in AD models. The authors provide a large amount of various data sets of high quality. However, in its current state, the study lacks experimental evidence to fully support the claims made here and move beyond correlation. One major concern is the lack of detailed, appropriate assays to indeed prove a direct function of lncMtDloop in the regulation of mitochondrial gene expression. It cannot be excluded that all in vitro and in vivo observations are due to other cellular effects, especially given the fact that a large number of other potential interaction partners was identified for lncMtDloop. Thus, clear evidence is lacking on how specifically the expression of this lncRNA may rescue the highly complex aberrations found in AD models.

The study proposes a very interesting finding and deserves publication, thus I recommend major revision to address the detailed comments found below.

Response: We sincerely value the referee's positive comments and constructive suggestions, which have significantly contributed to enhancing the quality of our manuscript.

Major concerns:

lncMtDloop

In spite of its high importance in this study, lncMtDloop itself is rather poorly characterized. Information on its size and sequence is largely based on sequencing data and bioinformatic predictions alone. More biochemical evidence is required.

Response: Thank the referee for your careful review. We assessed the relative information of lncMtDloop using the UCSC Genome Browser. The table below and Figure EV 1A illustrate the size, sequence, and sequence conservation among species.

- A number of other lncRNAs have been described in mitochondrial DNA (lncND5, lncND6 and lncCytb, Rackham et al, 2011, PMID:22028365). Similar to what is known about them, the authors need to include information on the sequence, especially 3' and 5' transcript ends (e.g. based on PARE or PolyA analyses) of lncMtDloop and clearly depict the position of this lncRNA in the D-loop region. To compare the abundance of mitochondrial lncRNAs, northern blots of lncMtDloop in comparison to the other lncRNAs are required. Those should be probed for both sense and antisense RNA to get an idea of their respective abundance.

Response: Thank the referee for your thorough review. We utilized the UCSC Genome Browser to characterize the relative information of lncMtDloop. The table below and Figure EV 1A delineate its size (939 bp), position (15,356-16,294 bp), and sequence conservation among species.

In addition, we used antisense RNA as controls to quantify the abundance of lncMtDloop, but observed minimal signals of antisense RNA, even with faint PCR band signals. Consequently, in subsequent experiments, we opted to use the WT group as the control (as shown in the figure below).

- More information on the potential generation of lncMtDloop is required. Based on their scheme, lncMtDloop appears to be generated after termination of H-strand transcription which is usually terminated before by yet unknown mechanisms. Interestingly, a previous study (Jemt et al, 2015; PMID: 26253742; which should be included in the discussion) has already described an antisense D-loop transcript generated upon recovery from mtDNA depletion, a state in which H-strand termination is reduced. The authors need to clarify if these two transcripts are the same or at least similar. In line with this, the study would benefit from including a similar mtDNA depletion/recovery assay. Furthermore, it is interesting to know how this lncRNA species may be processed - do the transcript ends contain punctuation elements? Did the authors observe an interaction with mtRNA processing enzymes such as MRPP1, MRPP3 or ELAC2 (to name a few) or the key RNA binding protein LRPPRC in their RNA-IP studies? The authors should discuss this point.

Response: Thank the referee for your valuable suggestions. In the study by Jemt et al. (as shown in the figure below), the antisense D-loop is positioned nearby at 15,986-16,116, relative to a short transcript of lncMtDloop (15,356-16,294 bp). This proximity suggests the potential absence of the binding element for the target protein.

In response to the reviewer's suggestion, we conducted an mtDNA depletion/recovery assay. Our aim was to explore the regulatory role of lncMtDloop in mtDNA transcription. N2a cells were treated with ethidium bromide (EB) for 8 weeks, a well-established method for inducing mtDNA depletion (Khozhukhar et al., 2023; Swerdlow, 2007). Following EB treatment, both mtDNA and mtRNA copy numbers significantly decreased, accompanied by reduced levels of mitochondrial complex (OXPHOS subunit proteins) and MitoTracker signaling. Of note, ectopic expression of lncMtDloop notably restored mtRNA levels without affecting mtDNA copy number and further augmented OXPHOS subunit protein levels (refer to Fig EV5 in the revised version). These findings indicate that lncMtDloop enhances the affinity and efficacy of TFAM interaction with the mtDNA promoter, thereby promoting transcription (refer to Fig EV4D in the revised version).

Moreover, in our previous study (Liu et al., 2017), we employed comprehensive analyses of RNA-seq and CAGE-seq (cap analysis of gene expression and sequencing) to characterize lncRNA expression in macaques. We generated cDNA libraries from polyadenylated RNA extracted from macaque brain tissue, enriching for full-length lncRNAs possessing both a 5'-cap and 3'-poly(A) tail. Thus, the captured lncMtDloop represents a processed and matured RNA with a 5'-cap, alternative splicing, and polyadenylation. For the RNA pull-down assay, we evaluated the mass spectrometry (MS) results in conjunction with catRAPID predictions. The figure below displays the proteins pulled down by RNA that exhibit high interaction scores with lncMtDloop. In the mitochondrial fraction, lncMtDloop demonstrated binding affinity for protein groups predominantly associated with carbon metabolism, cellular respiration, and mitochondrial transcription pathways, with no discernible presence of mtRNA processing enzymes.

- The co-localisation between lncMtDloop and Mito-ATP5 appears rather poor in primary neurons (Figure 1B). Large extramitochondrial portions can be observed, the authors should comment on what they might represent. Given their hypothesis regarding regulation of mitochondrial transcription, it would be interesting to see if the lncMtDloop colocalizes with active mitochondrial nucleoids. In situ hybridization in combination with BrU labelling and nucleoid staining will help to quantify the correlation between lncMtDloop and mtRNA synthesis.

Response: We appreciate the valuable suggestions from the referee. Indeed, there are some extramitochondrial signals observed. These signals could potentially represent impurities or escaped mtDNA transcripts due to mitochondrial permeability transition pore (mPTP) dysfunction. Given that this constitutes a small portion, our primary focus remains on elucidating the regulatory function of lncMtDloop within mitochondria. In response to the reviewer's suggestion, we conducted BrU labeling to quantify lncMtDloop and mtRNA synthesis. However, due to the lipid bilayer structure of mitochondria, BrU uptake into mitochondria proved challenging, resulting in the absence of fluorescence signals in mitochondria with BrU staining. As indicated by the anti-Brdu product manual, there were also no signals detected in mitochondria (refer to the figure below). We acknowledge that there are obstacles hindering our ability to quantify lncMtDloop and mtRNA synthesis within the limited timeframe. We hope to optimize staining methods for BrU staining in future experiments.

TFAM interaction and recruitment

- The authors employ RNA pulldown assays to identify potential binding partners of lncMtDloop, using the respective antisense RNA as a control. They manage to pulldown a large quantity of proteins, both in cytosolic and mitochondrial fractions. Curiously, the authors focus on mitochondrial transcription, although other interactions partners corresponding to other pathways, e.g. cellular respiration, were more highly enriched. Please clarify this rationale. Furthermore, the authors observe a strong interaction with key cytosolic proteins. A key question that strongly relates to the in cellulo and in vivo observations remains: How do the authors rule out that lncMtDloop effects on other pathways are responsible for the observed changes?

Response: Thank the referee for your careful review. While it's true that many RNA-pulled proteins may be enriched in cellular respiration or other pathways, our analysis with the catRAPID algorithm indicates that these proteins exhibit low interaction scores with lncMtDloop. In contrast, TFAM demonstrates a strong binding affinity with lncMtDloop. Moreover, considering the specific position of lncMtDloop in the mitochondrial genome, we hypothesize that it is more likely to have a regulatory function in mtDNA homeostasis, leading us to select TFAM as the potential binding protein. Given that mtDNA encodes essential components of the OXPHOS complexes, which play a crucial role in neuronal ATP synthesis, we perceive mitochondrial transcription as the upstream response, with other pathways including cellular respiration serving as downstream responders. Therefore, our primary focus lies on the mitochondrial transcription pathway.

- The data on a direct lncMtDloop: TFAM interaction suffers from a lack of adequate assays and controls. TFAM, the major mitochondrial packaging protein, binds mtDNA specifically and non-specifically. However, TFAM was shown to bind more complex RNA structures such as 4-way junctions and mitochondrial tRNAs, albeit at a lower affinity (Brown et al, 2015; PMID: 26545237). It is thus not surprising that TFAM might pull-down certain, more complex RNA species. Furthermore, this might explain the observation with truncated constructs - do the truncations lead to the loss of secondary structure? To fully appreciate the proposed interaction, it is necessary to know other mitochondrial RNA species detected after pull-down. When performing qRT-PCR after TFAM pull-down, what were the detection levels of e.g. mitochondrial mRNAs, tRNAs, 7S RNA or the other mitochondrial lncRNAs in comparison?

Response: Thank the referee for your thorough review. We employed 12s rRNA as a control, as it is also transcribed from mtDNA, and discovered that TFAM can indeed pull down a small portion of 12s rRNA. Given TFAM's role as the major mitochondrial packaging protein, it is rational to expect its binding with mtDNA transcripts. However, our findings revealed that TFAM significantly pulls down a larger portion of lncMtDloop compared to 12s rRNA, indicating a strong binding affinity between TFAM and lncMtDloop.

Indeed, the stem-loop structure plays a crucial role in lncRNA functionality. Utilizing the structure predicted by minimum free energy and the binding sites predicted by catRAPID, we designed truncated constructs while ensuring the preservation of the stem-loop structure to maintain structural integrity.

- The EMSA data is also lacking, as only overexpression of lncMtDloop was included. The authors need to perform EMSA assays in which the concentration of lncMtDloop is titrated. As controls, the same assay needs to be performed in the presence of 7S RNA, other mitochondrial lncRNAs and dsDNA.

Response: Thank the referee for your careful review. We have extensively researched a collection of papers detailing EMSA (Electrophoretic Mobility Shift Assay) methods. According to protocols outlined in studies such as Yadav et al., published in *Molecular Cell* (2022), and Lei et al., published in *Nature Communications* (2021), it is necessary to perform titration with purified protein in the EMSA assay.

In our study, the OE-lncMtDloop group refers to the lysate extract from N2a cells overexpressing lncMtDloop as an RNA species. We adopted methodologies outlined in publications such as Wang et al., published in *Cell Death & Differentiation* (2021), and utilized N2a mitochondrial extract components.

Nevertheless, in pursuit of a dose-dependent effect, we also prepared extracts with varying concentrations, employing 12s rRNA as a control to mitigate non-specific binding (as shown in the figure below). The outcomes revealed that even a single copy of extracts was adequate to differentiate from the control groups.

- A direct effect of lncMtDloop on mitochondrial transcription cannot be concluded from the qPCR data alone, as the RNA species binds numerous targets within mitochondria, all of which might generate secondary effects on mitochondrial gene expression. It is absolutely necessary to perform in vitro transcription assays in the presence of RNA to substantiate the main hypothesis, similar to what was done for 7S RNA (Zhu et al, 2022). Assays need to be performed using both HSP and LSP templates to conclude a global effect. Furthermore, assays need to be performed titrating concentrations of lncMtDloop as well as control RNA (e.g. antisense), other lncRNAs and 7S RNA as controls.

Response: Thank the referee for your insightful suggestion. We acknowledge that

further studies would be beneficial to uncover the intricate details of interaction and regulatory mechanisms. However, establishing in vitro transcription assays necessitates a mature system for protein expression and purification, which we currently lack due to time constraints. We have taken note of relevant articles such as those by Wang et al. (Cell Stem Cell, 2015; Cell Death & Differentiation, 2021) and Lei et al. (Nature Communications, 2021), which demonstrate that techniques such as ChIP, EMSA, and luciferase reporter assays can elucidate regulatory mechanisms. Our focus primarily lies in elucidating the role and regulatory function of lncMtDloop in AD conditions rather than delving into basic biochemical research. In the future, we aspire to dedicate more time to overcome this challenge and delve deeper into the regulatory mechanisms of lncMtDloop from a fundamental research perspective. As it is known, the H strand of mtDNA encodes 12 polypeptides crucial for OXPHOS function, while the L strand encodes a single polypeptide (MT-ND6). To comprehensively assess the global effects of both HSP and LSP, we conducted additional studies, including evaluations of transcript and protein levels of OXPHOS subunit proteins predominantly encoded by mtDNA (as shown in the revised Fig 5F-H).

As mentioned in the previous review, we extensively researched various papers detailing EMSA methods. According to Yadav et al. (Molecular Cell, 2022) and Lei et al. (Nature Communications, 2021), titration of purified protein is necessary for the EMSA assay. In our study, the OE-lncMtDloop group denotes the lysate extract from N2a cells overexpressing lncMtDloop. We followed methodologies outlined by Wang et al. (Cell Death & Differentiation, 2021) and utilized N2a mitochondrial extract as components. To ensure a dose-dependent effect, we prepared extracts with varying concentrations and employed 12s rRNA as a control to mitigate non-specific binding. Our results indicated that even a single copy of extracts was sufficient to distinguish from the control groups. Additionally, we used antisense RNA as another control to further rule out non-specific binding (as shown in the figures below)..

RNA import assay

The authors utilize allotropic expression of lncMtDloop to increase cellular levels of their target RNA species. It should be noted that RNA import into mitochondria is a very controversial topic that has only been shown in a few selected studies in very limited circumstances.

- The authors should include other mitochondrial lncRNAs as controls when performing their PNPase pull-down to assess targeting abundance, they should have a similar hairpin and be targeted.

Response: Thank the referee for your careful review. Indeed, other mitochondrial lncRNAs, such as ND5, ND6, among others, were found to exhibit low interaction properties with PNPASE according to catRAPID prediction. In our study, we utilized 12s rRNA as a control for comparative analysis (as shown in the figure below).

- As the authors state, the RNA import assay employing the MRPS12 UTR directs lncMtDloop to the mitochondrial surface. This does not necessarily mean that it gets imported into the mitochondrial matrix which is a prerequisite for the proposed mode of action. Here, RNA gets amplified from mitochondrial fractions, thus it might well be

on the outer surface. The authors need to prove matrix localization of their lncMtDloop, e.g. by coupling RNase digest on isolated, intact mitochondria to subsequent RNA isolation and northern blotting.

Response: Thank the referee for your careful review. In our research hypothesis, we packaged the lncMtDloop MRPS12-3'UTR into LV virus to infect the cells. Subsequently, lncMtDloop MRPS12-3'UTR was transcribed in the nucleus and transported into the cytosolic fraction. Guided by MRPS12 3'UTR, the free lncMtDloop in the cytosolic fraction can be transported to the mitochondrial surface. Interacting with p32, cytosolic free lncMtDloop is facilitated to the mitochondrial surface. Subsequently, lncMtDloop binds to PNPASE with a specific stem-loop sequence and is transported into the mitochondria. We conducted Western blot analysis to verify the matrix localization of lncMtDloop. Malate dehydrogenase (MDH) served as the mitochondrial matrix marker for characterization of the mito-matrix location (as shown in the figure below). We employed a mitochondrial isolation kit to isolate and collect mitochondria. Significant enrichment of MDH was observed in the mitochondrial fraction compared to other fractions. We considered RNAscope-IF dual staining to visualize matrix localization. However, MDH is also distributed in the cytosolic fraction, posing challenges in obtaining specific colocalization signals.

AD model systems

Interestingly, the authors observe a clear decrease in the levels of lncMtDloop in various AD models, pointing to a clinical relevance of their observations. Furthermore, the

authors provide an extensive panel of high-quality data in cellular systems and in vivo showing that morphological and functional alterations of mitochondria as well as physiological parameters can be rescued by the expression of lncMtDloop. The main problem here is that it remains unclear if the observed effects are due to a direct effect on mitochondrial gene expression.

Response: Thank the referee for your careful review. In our mtDNA depletion assay, we observed that overexpression of lncMtDloop can rescue mtDNA transcription, thereby maintaining mitochondrial genome homeostasis and influencing OXPHOS activity. Considering that mtDNA encodes crucial components of the OXPHOS complexes, pivotal for cellular ATP synthesis, this restoration of mtDNA transcription is significant. Notably, high-energy demands are imperative for processes such as neuronal signal transmission (e.g., Ach) and synaptic plasticity. Consequently, disturbances in mtDNA transcription render neurons particularly susceptible to energy deficits. By delving into the pathogenesis of Alzheimer's disease (AD) through the lens of mitochondrial genome homeostasis, we gain fresh insights into potential therapeutic targets. Hence, we propose an indirect relationship between lncMtDloop and mtDNA expression in AD, delineated by the pathway: lncMtDloop → TFAM → mtDNA → OXPHOS subunits → neuronal energy supply → synaptic plasticity in AD.

- While there is a clear effect on mitochondrial respiration and morphology, this might be due to an uncharacterized function of lncMtDloop. In their initial RNA-IP some of the major hits in mitochondria such as cellular respiration were much more significantly enriched. The authors detect an effect on Cytochrome C oxidase, how do they exclude that there is no direct effect of lncMtDloop on cellular respiration? Were perhaps Complex IV assembly factors pulled down? Disturbances in OXPHOS frequently lead to changes in cristae and mitochondrial morphology, similar to what was observed here. Furthermore, the data sets shown do not provide clear evidence that lncMtDloop gets imported into the mitochondrial matrix to be able to exert a function in mitochondrial gene expression. Thus, it is possible that lncMtDloop, attached to the mitochondrial

surface, might interact with some of the cytosolic partners observed in the RNA-IP, exerting effects on mRNA metabolism and cytoplasmic translation (categories enriched). To address these concerns, the authors should re-evaluate their RNA-IP for a potential direct modulation of cellular respiration and consider this possibility in their conclusions and discussion.

Response: Thank you to the referee for your meticulous review. Indeed, many RNA-pulled proteins are enriched in cellular respiration or other pathways. However, according to the catRAPID algorithm prediction, these proteins exhibit low interaction scores with lncMtDloop, whereas TFAM demonstrates a strong binding ability with lncMtDloop. Moreover, considering the specific position of lncMtDloop in the mitochondrial genome, we hypothesize that it is more likely to play a regulatory role in mtDNA homeostasis. Since mtDNA encodes essential components of the OXPHOS complexes critical for neuronal ATP synthesis, we consider mitochondrial transcription as the upstream effect, with other pathways including cellular respiration responding downstream to disturbances in mtDNA homeostasis. Therefore, our primary focus lies on the mitochondrial transcription pathway. Regarding cytosolic partners observed in the RNA pull-down, it's worth noting that these RNA-pulled proteins are identified in vitro rather than under endogenous conditions. To explore the possible mitochondrial trafficking pathway of lncMtDloop, we incubated a portion of cytosolic lysate and mitochondrial lysate with in vitro transcribed lncMtDloop (both sense and antisense). However, under endogenous conditions, lncMtDloop is predominantly located in mitochondria, with minimal distribution in the cytosolic fraction. As a result, our focus primarily centers on mitochondria-related pathways to investigate the regulatory mechanism of lncMtDloop, while we refer to the cytosolic results to interpret the mitochondrial trafficking mechanism of lncMtDloop.

- The authors observe a decrease in the transcript levels of mitochondrial mRNAs and correlate this with the observed mitochondrial and cellular dysfunctions. However, it has been shown that depletion of mitochondrial RNAs, even to about 30% of controls,

is well-tolerated in vivo (Lagouge et al, 2015; PMID: 26247782). Similarly, neurons tolerate disturbances of mitochondrial dysfunction for a very long time in vivo (Sorensen et al, 2001; PMID: 11588181). How do the authors reconcile their findings with these observations? Western blotting of OXPHOS protein levels is needed to determine the protein levels of mtDNA-encoded/OXPHOS subunits. Is the rate of mitochondrial transcription increased after overexpression of the *IncMtDloop* as a compensatory response? This can be analyzed by e.g. in organello transcription in cell lines. Their qPCR data argues for a Complex IV deficiency which is also observed in their activity assay - is Complex I also disturbed?

Response: Thank the referee for your careful review. We have investigated the changes in *IncMtDloop* and mtDNA transcript levels over time. Our findings revealed a 40% decrease in *IncMtDloop* expression in the whole brain of 3xTg mice compared to WT mice during the aging process, particularly notable at 9 to 12 months (refer to Fig EV6B). Conversely, mtRNA copy numbers displayed a delayed decline, beginning at 12 months (as depicted in revised version Fig EV6C). This suggests a dose-dependent effect of mtRNA depletion. Throughout the aging process, the accumulation of reactive oxygen species (ROS) and DNA damage may contribute to progressive mtRNA depletion. Given that *IncMtDloop* serves as a regulatory factor in mtDNA transcription, its decline preceding alterations in mtRNA exacerbates its dysregulatory impact on mtDNA homeostasis.

In addition, we also conducted Western blot analysis of OXPHOS subunit proteins to assess alterations in mtDNA-encoded protein levels (the revised Fig 5G and H). Our findings revealed a relative decline in complexes I, II, III, IV, and V. Interestingly, overexpression of *IncMtDloop* almost completely rescued their deficiency. We do not interpret this as a compensatory response. Through extensive molecular biology experiments and mtDNA depletion assays in previous sections, we discovered that *IncMtDloop* specifically regulates mtDNA transcription but not its replication. Furthermore, our findings suggest that *IncMtDloop* plays a role in modulating

mitochondrial OXPHOS activity in AD neurons.

- Is the decrease in IncMtDloop in AD models consistent with a general decrease in mtDNA levels? It would be interesting to know if a stable ratio of mtDNA:IncRNA is maintained or if the ratio is shifted over time, correlating with pathology.

Response: Thank the referee for your valuable suggestions. In the revised version, we have delved into the relationship between mtRNA, IncRNA, and pathology over time. We conducted time-point screenings at 1 month, 3 months, 6 months, 9 months, and 12 months to observe changes in IncMtDloop and mtRNA levels during the early stages. Our findings revealed a decline in IncMtDloop expression in the whole brain of 3xTg mice during the aging process, particularly notable at 9 to 12 months (refer to Fig EV6B). Conversely, mtRNA copy numbers displayed a delayed decline, commencing at 12 months (as depicted in the revised version Fig EV6C).

Referee #3:

Xiong et al. report a new long non-coding mitochondrial DNA that shows to be a critical regulator of mitochondrial homeostasis through regulation of TFAM, the major mitochondrial transcriptional factor, and with relevance in an Alzheimer's disease model. The research is well designed and both the recovery of mitochondrial network dynamics and pathological features in 3xTg AD mice after IncMtDloop expression are convincing. However, the 3xTg model is a very AD aggressive model and it would be interesting to understand the mtDNA transcription phenotype in early stages of the disease where a treatment strategy is likely to be more efficient. Moreover, the reader would benefit of a more detailed description of results (e.g. behavior) and detailed methods and analysis.

Major comments:

- The authors show no changes in mtRNA expression in macaque and mice during aging. However, I wonder if that is true in the disease model. Several studies show increased transcription of mtDNA genes in early AD/MCI patients and pre-symptomatic AD models (e.g., PMID: 15075441, 27793643, 37907591). This hyper mitochondrial metabolism may be a compensatory mechanism at early stages. It would be very interesting to see if young/early stage 3xTg show any changes in IncMtDloop and mtRNA levels and when this phenotype initiates.

Response: We appreciate the valuable suggestions. In response, we conducted time-point screening at intervals of 1 month, 3 months, 6 months, 9 months, and 12 months to observe changes in IncMtDloop and mtRNA levels during the early stages. Our findings revealed a decline in IncMtDloop expression in the whole brain of 3xTg mice during the aging process, particularly noticeable at 9 to 12 months (as shown in the revised Fig EV6B). Conversely, mtRNA copy numbers began to decline with a delay, starting at 12 months (as depicted in the revised Fig EV6C).

- In figure 5G, the authors show some representative STED pictures of mitochondrial morphology but is not easy to understand if the analysis in K and L was done using STED or confocal. By reading the text it seems it was done using STED but pictures in 5H seem to have confocal resolution. The fig 5H would also benefit of having a zoom in of the fission event along time - it's difficult to follow any event along time in the representative pictures shown.

Response: Thank the referee for your careful review. Indeed, to explore the dynamic changes of mitochondrial fusion and fission over time, we utilized the MitoESq-635 probe provided by the Xi laboratories. This probe is specifically designed for long-term, high-resolution STED nanoscopy, as documented in publications by Yang et al. (*Nature Communications*, 2020) and Liu et al. (*Nature*, 2017). It's worth noting that confocal imaging presents challenges for maintaining long-term live imaging.

Additionally, we have replaced the previous image (the revised Fig. S2B) with a zoom in version (the revised Fig. 5I), along with the inclusion of new appendix videos S1-4. These videos exclusively reveal mitochondrial dynamics.

Methods should state how fission events were calculated (e.g. automatized or manual quantification) and in which region of the neurons (soma only?). Same for TEM images. How did the authors selected the cut off to what is tubular and short tubular?

Response: Thank the referee for your careful review. Indeed, we analyzed fission events through manual quantification (comparing the screenshots with last previous time point to statistically analyze the proportion of peripheral and midzone fission), calculating them within the region of the neuronal soma (We screened a large view encompassing the soma and nucleus, as depicted in the figure below). The same methodology was applied to TEM images. We utilized the aspect ratio as a cutoff to distinguish between tubular (>2) and short tubular (≤ 2) structures.

- The results description about memory performance show be better described/explained. E.g., at some point the authors say "contextual and cued fear conditioning text was carried out." And then continue to MWM description without saying the results of the previous text.

Response: Thank the referee for your valuable suggestion. We have thoroughly revised the description of the results to ensure clarity. In addition, we have expanded upon certain sections to provide more detailed explanations and enhance understanding.

Minor comments:

- Please add references to the sentences: (page 5) "(...) a common structure among well-documented lncRNAs"; (page 6) "Mitochondrial genome homeostasis (...) in metabolic disorders, aging and neurodegenerative diseases."

Response: Thank the referee for your careful review. We have added the references in the revised version.

- In page 13, please rephrase the sentence: "To explore whether the loss of IncMtDloop is connected to decreased mtDNA content and reduced mtRNA expression." - it seems unfinished.

Response: Thank the referee for your careful review. We apologized for the mistake.

The sentence has been rephrased accordingly.

- Tubular in Fig 6G, H is misspelled.

Response: Thank the referee for your careful review. The misspelled tubular has been corrected accordingly.

- It's difficult to say that the organelle pointed in Fig. 6D is a mitochondrion. There is no double membrane or visible cristae and it's darker than normal. It could be a lysosome.

Response: Thank the referee for your careful review. It appears that mitochondria in Fig. 6D may be mistaken for a lysosome due to its appearance. Upon closer inspection with image zoom, the double membrane of the mitochondria can be observed, although the matrix appears dark. To ensure accuracy, we have replaced the image in the revised Fig. 6D.

- In page 15 authors say that injections of AAV-PHP.eB-IncMtDloop-3UTRMRPS12 were done through mouse tail vein but in Fig S2 it seems it was injected in hippocampus.

Response: Thank the referee for your careful review. In this study, we stated two methods of AAV infection: tail-vein injection and brain stereotactic microinjection. The Schaffer collateral-CA1 synapse serves as a well-established model for long-term potentiation (LTP) and synaptic plasticity, predominantly situated around the hippocampus region. Given that the hippocampus is the most susceptible region to pathological changes in Alzheimer's disease (AD), we opted for localized stereotaxic injection of AAV9-IncMtDloop-3'UTRMRPS12 into the hippocampal area. This approach enabled us to directly assess the impact of this allotropic intervention on hippocampal neuron firing.

- Mitophagy modulators would fit better associated with peripheral fission events described in Fig 5.

Response: Thank the referee for your valuable suggestion. Since this part of the study was conducted on mouse brain tissue, which is directly relevant to Fig. 6, where abnormal mitochondrial morphology is depicted, it is more appropriate to describe the connection between abnormal mitochondrial morphology and untimely mitophagy in Fig. 6.

- In Fig 7B add mention to the staining used.

Response: Thank the referee for your careful review. The requested staining marker 6E10 has been labeled accordingly (Fig. 7B).

- Fig 7E-H would benefit of having a title/legend for hippocampus vs PFC.

Response: Thank the referee for your careful review. The requested title and legend for hippocampus vs PFC have been added accordingly.

- No ethics are described for mice experiments.

Response: Thank the referee for your careful review. The requested ethics description of mice experiments has been added in the Methods accordingly.

- Software used in MWM analysis?

Response: Thank the referee for your careful review. The software information of MWM has been provided in Appendix Table S9 accordingly.

Dear Dr. Li,

Congratulations on a great revision! Overall, the referees have been positive. However, Referee 2 and 3 have a few remaining concerns. While we realize this would require extra work and add a few more weeks before final publication, we feel this will greatly strengthen your important study.

When you submit your revision, please also take care of the following editorial items and add this also to your point-by-point response:

1. Please provide ORCID ID for author Lu.
2. Please provide the following funding info online in eJP: the Ministry of Science and Technology of China (2015CB755605), Peking University (BMU2019YJ001), the key basic project of Yunnan province (E039030401), The Hong Kong Research Grants Council (RGC)-General Research Fund (GRF) (PI: GRF16100219 and GRF16100718), Collaborative Research Fund (CRF) (Co-I: C4033-19EF), CUHK-Improvement on Competitiveness in Hiring New Faculties Funding Scheme (PI: Ref. 133) and CUHK-School of Life Sciences Startup funding.
3. Please provide up to five keywords, which may or may not appear in the title, should be given in alphabetical order, below the abstract, each separated by a slash (/).
4. Please remove the author contribution section from the main manuscript.
5. All figures should be referred to in the main manuscript and in chronological order. Please add callouts for Appendix Table S5-S9 and for the movies.
6. Please provide an author checklist.
7. 7 EV figures are in one PDF file; we can accommodate 5 as expanded view which means up to 5 can be uploaded separately and their legends should be provided in the ms file; the rest of the figures should go to the Appendix file.
8. Please provide page numbers in the table of contents on the title page for the appendix file.
9. Please change the size of the synopsis image to 550 pixels wide by 200-600 pixels high and provide in jpeg, TIFF or png format.
10. Please remove the inclusion and diversity section from the main manuscript.
11. Please update the names of the videos to Movie EV1-EV3.
12. "Appendix Video S1-4. Fusion and Fission of mitochondrial living images (seen in the separated files)" needs to be removed from the Appendix since each movie needs to be zipped with a corresponding legend (which can be provided in a readme.txt file) so that we have one zip folder per movie; the mentioning of the movies should also be removed from the Appendix ToC.
13. Please note that the figure 6e-f; EV 2h; does not contain any quantification graph, kindly rectify the statistics related information in the figure legend appropriately.
14. Please note that the figure 5a-b does not contain any statistical parameter, kindly rectify the statistical test related information in the figure legend appropriately.
15. Please note that the legends for figures 5k-l is not provided in the sequential manner (legend for figure 5k is provided before legend of figure 5l). This needs to be rectified.
16. Please note that the legends for figures 6b-d is not provided in the sequential manner (legend for figure 6d is provided before legend of figure 6b-c). This needs to be rectified.
17. Please indicate the statistical test used for data analysis in the legends of figures EV 4b-c.
18. Please note that in figures 2h-i; 6g-h; 7g-h; there is a mismatch between the annotated p values in the figure legend and the annotated p values in the figure file that should be corrected.
19. Please note that information related to n is missing in the legends of figures 5q; 6g-h; EV 7g-i.

20. Although 'n' is provided, please describe the nature of entity for 'n' in the legends of figures 2b, f, h-i; 3e, j, l, o; 4l, n-o; 5d-f, h, o-p, r-t; EV 2g, i-j; EV 5a-b, d, f; EV 6a-c, f, i; EV 7b-c, e.

21. Please define the error bars in the legend of figure 5q.

22. Please define the scale bar for figures EV 2k; EV 5c.

23. Please add a scale bar and its definition for figures 6b-c.

24. Please define the red arrows in the legend of figure 6a, d.

25. Please define the white arrows in the legend of figure 5j.

Thank you for the opportunity to consider your work for publication and I look forward to reading your revision.

Warm wishes,
Kelly

Kelly M Anderson, PhD
Editor, The EMBO Journal
k.anderson@embojournal.org

Use the link below to submit your revision:

Referee #1:

The authors answered all criticism raised by this reviewer. In its present form, this manuscript improved and deserves to be published

Referee #2:

Comments on Xiong et al „The mtDNA-derived lncMtDloop promotes mitochondrial homeostasis maintenance and implications in AD" - Revision 1

The authors of the manuscript entitled "The mtDNA-derived lncMtDloop promotes mitochondrial homeostasis maintenance and implications in AD" have considered major concerns and largely improved the revised version. Still, key concerns have not been addressed in sufficient detail, especially regarding biochemical evidence related to lncMtDloop expression and its direct effect on mtDNA expression. The authors should address these concerns (see details below) and modify their manuscript accordingly, before acceptance.

Major concerns:

lncMtDloop

My previous review stated clearly that "more biochemical evidence is required" concerning the characterization of lncMtDloop and that "northern blots in comparison to the other lncRNAs are required". This key piece of evidence is still missing, although northern blotting is a method employed in this study (see Figure 4). To fully characterize their very central lncRNA in sufficient detail, the authors need to perform northern blotting of lncMtDloop, 7S RNA and at least one further lncRNA (e.g. lncND5), probing sense and antisense RNA to compare abundance and biochemical size. On the other hand, the authors very clearly show the expression of lncMtDloop in their mouse models by PCR, this image should be included in the study and supplemented by showing at least 7S RNA expression.

TFAM interaction and recruitment

I would like to thank the authors for clarification of their approach on why they chose to study TFAM as the main interaction partner, compared to other hits. An additional sentence clarifying that other proteins identified had a lower catRAPID score, giving an example, would help in the text.

In Figure 3E, the authors chose to include 12 rRNA - a very abundant and stable mtRNA species - as a control and noticed clear differences in TFAM binding. The graphs should be adapted to show the same scale on the y-axis, so the difference in enrichment becomes directly evident. Here, the enrichment of one other lncRNA, preferable 7S RNA, needs to be included to

appreciate IncMtDloop binding compared to another lncRNA, as requested in my previous review.

The study largely improved by including EMSA assays; however, some questions remain. In the rebuttal letter, the authors show a EMSA titration assay (which should be included in the main study). In lane 2, a strong band shift is already seen, even without addition of the lncRNAs - how do the authors explain this? In their other EMSA, a band shift is already seen upon overexpression of the antisense IncMtDloop, which is contrary to their other assay. Why is this the case? Please discuss. My previous review clearly asked for in vitro transcription assays to substantiate the conclusions made on a direct role of IncMtDloop on mtDNA transcription (as done in the Zhu et al, 2022 paper), data that has not been included here. However, I understand that establishing a working in vitro system takes time and expertise. The authors should thus very clearly state these limitations of their study - due to lack of direct evidence - in the result and discussion section of the manuscript and more carefully interpret the conclusions drawn throughout the manuscript. Nonetheless, their finding on increased mtDNA transcription, but not mtDNA replication after mtDNA depletion is very interesting and merits further investigation, I strongly advise the authors to address this in future study and highly recommend supporting in vitro studies.

AD model systems

The authors have now included western blot analysis of OXPHOS complexes in their mouse models and observe a relative OXPHOS decline that is rescued upon expression of IncMtDloop. Densitometric quantification is partly difficult, especially upon overexposure of blots. Here, for some reason, the LV control band is continuously smaller compared to the others in the source data, making comparison by intensity difficult. Especially complex IV is very hard to distinguish in the blot shown here, which is overexposed and of poor quality. Is there a less over-exposed version available? Better images are required for densitometry. Furthermore, I was wondering how the authors reconcile their findings with published work showing that mt-mRNA depletion is well-tolerated in vivo and that neurons are generally fine upon TFAM-knockout - they still need to discuss this point made in my previous review.

Minor points:

- The authors should possibly modify their statement on IncMtDloop sizes - are size differences of hundreds of base pairs differences still slight?
- Figures 5A and E - could the authors clarify in the figure legends which probes were used for mtDNA and mtRNA?

Referee #3:

I acknowledge the effort of the authors to extensively answer the reviewers questions. However, there are still minor points that should be addressed.

- Regarding the role of mitochondrial fission in Fig 5, the authors say they analyzed perinuclear mitochondria/ mitochondria in the soma. I wonder if decreased levels of peripheral fission induced by IncMtDloop can affect the entry of mitochondria in the axon and anterograde transport of mitochondria, that are highly dependent on peripheral fission. How does it look the distribution of mitochondria in the axon and dendrites after IncMtDloop expression?

- In agreement with Reviewer #1, the IncMtDloop and Mito-ATP5 colocalization is not strong. In fact, this is particularly evidence in neurons, while the other cells show a much clear colocalization. The authors suggested that mtDNA transcripts could have escaped to the cytosol and, as far as I know, both cytosolic mtDNA and mtRNA are known to induce neuroinflammation. So I wonder if IncMtDloop can have a role in mitochondria-induced neuroinflammation. The authors should check if there are any differences in the cytosolic versus mitochondrial levels of IncMtDloop in hippocampal WT vs 3xTg neurons.

- The authors didn't really answer how they decided the cut off for mitochondrial morphology: tubular vs short tubular vs fragmented. $\{less\ than\ or\ equal\ to\}2 >2$, seems very arbitrary. The authors should create a cut off based on the statistical quartiles of the aspect ratio of all mitochondria measured.

- Regarding description in the text for AAV injections: authors should add a) for hippocampus injections and b) c) for tail injections when they refer to Appendix Fig S3. Otherwise it's confusing.

Response to the Editor's comments

1. Please provide ORCID ID for author Lu.

Response: *The ORCID ID for author Lu is 0000-0003-0742-9072.*

2. Please provide the following funding info online in eJP: the Ministry of Science and Technology of China (2015CB755605), Peking University (BMU2019YJ001), the key basic project of Yunnan province (E039030401), The Hong Kong Research Grants Council (RGC)-General Research Fund (GRF) (PI: GRF16100219 and GRF16100718), Collaborative Research Fund (CRF) (Co-I: C4033-19EF), CUHK-Improvement on Competitiveness in Hiring New Faculties Funding Scheme (PI: Ref. 133) and CUHK-School of Life Sciences Startup funding.

Response: *All funding information has been provided online in eJP.*

3. Please provide up to five keywords, which may or may not appear in the title, should be given in alphabetical order, below the abstract, each separated by a slash (/).

Response: *The five keywords are Alzheimer's disease, mitochondrial homeostasis, mtDNA, IncMtDloop, and TFAM.*

4. Please remove the author contribution section from the main manuscript.

Response: *The author contribution section has been removed from the main manuscript accordingly.*

5. All figures should be referred to in the main manuscript and in chronological order. Please add callouts for Appendix Table S5-S9 and for the movies.

Response: *Thanks for your careful review. The requested corrections have been made accordingly.*

6. Please provide an author checklist.

Response: *The author checklist has been provided accordingly.*

7. 7 EV figures are in one PDF file; we can accommodate 5 as expanded view which means up to 5 can be uploaded separately and their legends should be provided in the ms file; the rest of the figures should go to the Appendix file.

Response: *Thanks for your careful review. The requested changes regarding EV Figures have been corrected accordingly.*

8. Please provide page numbers in the table of contents on the title page for the appendix file.

Response: *Thanks for your careful review. The requested page numbers have been added accordingly.*

9. Please change the size of the synopsis image to 550 pixels wide by 200-600 pixels high and provide in jpeg, TIFF or png format.

Response: *The requested change in the synopsis has been made accordingly.*

10. Please remove the inclusion and diversity section from the main manuscript.

Response: *Thanks for your careful review. The inclusion and diversity section has been removed accordingly.*

11. Please update the names of the videos to Movie EV1-EV3.

Response: *The names of the videos to Movie EV1-EV3 have been updated accordingly.*

12. "Appendix Video S1-4. Fusion and Fission of mitochondrial living images (seen in the separated files)" needs to be removed from the Appendix since each movie needs to be zipped with a corresponding legend (which can be provided in a readme.txt. file) so that we have one zip folder per movie; the mentioning of the movies should also be removed from the Appendix ToC.

Response: *Thanks for your careful review. The requested changes in the information in the videos have been corrected accordingly.*

13. Please note that figure 6e-f; EV 2h; does not contain any quantification graph, kindly rectify the statistics-related information in the figure legend appropriately.

Response: *Thanks for your careful review. The requested quantifications in Figure 6e-f and EV 2h have been provided accordingly.*

14. Please note that the figure 5a-b does not contain any statistical parameter, kindly rectify the statistical test-related information in the figure legend appropriately.

Response: *Thanks for your careful review. We have revalued the statistical test and make sure there is no statistical significance in these two figures, which are consistent with our description in the manuscript.*

15. Please note that the legends for figures 5k-l is not provided in the sequential manner (legend for figure 5k is provided before legend of figure 5l). This needs to be rectified.

Response: *Thanks for your careful review. The sequential manner of the legends for figures 5k-l has been corrected accordingly.*

16. Please note that the legends for figures 6b-d is not provided in the sequential manner (legend for figure 6d is provided before legend of figure 6b-c). This needs to be rectified.

Response: *Thanks for your careful review. The sequential manner of the legends for figures 6b-d has been corrected accordingly.*

17. Please indicate the statistical test used for data analysis in the legends of figures EV 4b-c.

Response: *Thanks for your careful review. These enrichment analyses were performed using Metascape. The requested description of enrichment analyses has been provided accordingly.*

18. Please note that in figures 2h-i; 6g-h; 7g-h; there is a mismatch between the annotated p values in the figure legend and the annotated p values in the figure file that should be corrected.

Response: *Thanks for your careful review. The mismatch between the annotated p values in the figure legend and file has been corrected accordingly.*

19. Please note that information related to n is missing in the legends of figures 5q; 6g-h; EV 7g-i.

Response: *Thanks for your careful review. The missing "n" information in the legends of figures 5q; 6g-h; EV 7g-i has been provided accordingly.*

20. Although 'n' is provided, please describe the nature of entity for 'n' in the legends of figures 2b, f, h-i; 3e, j, l, o; 4l, n-o; 5d-f, h, o-p, r-t; EV 2g, i-j; EV 5a-b, d, f; EV 6a-c, f, i; EV 7b-c, e.

Response: *Thanks for your careful review. The requested nature of the entity for “n” in all figures has been provided accordingly.*

21. Please define the error bars in the legend of figure 5q.

Response: *Thanks for your careful review. The error bars in Figure 5q have been defined accordingly.*

22. Please define the scale bar for figures EV 2k; and EV 5c.

Response: *Thanks for your careful review. The scale bars for figures EV 2k; and EV 5c have been defined accordingly.*

23. Please add a scale bar and its definition for figures 6b-c.

Response: *Thanks for your careful review. The requested scale bar and its definition in Figures 6b-c have been provided accordingly.*

24. Please define the red arrows in the legend of figure 6a, d.

Response: *Thanks for your careful review. The red arrows of Figure 6a, d have been defined in the legend accordingly.*

25. Please define the white arrows in the legend of figure 5j.

Response: *Thanks for your careful review. The white arrows of Figure 5j have been defined in the legend accordingly.*

Response to the reviewers' comments

Referee #1:

The authors answered all criticism raised by this reviewer. In its present form, this manuscript improved and deserves to be published

Response: *We greatly appreciate the referee's positive comments, as they serve to affirm the value and quality of our work. These encouraging remarks inspire us to continue our efforts and reinforce our commitment to rigorous research and scientific excellence.*

Referee #2:

Comments on Xiong et al „The mtDNA-derived lncMtDloop promotes mitochondrial homeostasis maintenance and implications in AD" - Revision 1

The authors of the manuscript entitled "The mtDNA-derived lncMtDloop promotes mitochondrial homeostasis maintenance and implications in AD" have considered major concerns and largely improved the revised version. Still, key concerns have not been addressed in sufficient detail, especially regarding biochemical evidence related to lncMtDloop expression and its direct effect on mtDNA expression. The authors should address these concerns (see details below) and modify their manuscript accordingly, before acceptance.

Major concerns:

lncMtDloop

My previous review stated clearly that "more biochemical evidence is required" concerning the characterization of lncMtDloop and that "northern blots in comparison to the other lncRNAs are required". This key piece of evidence is still missing, although northern blotting is a method employed in this study (see Figure 4). To fully characterize their very central lncRNA in sufficient detail, the authors need to perform northern blotting of lncMtDloop, 7S RNA and at least one further lncRNA (e.g. lncND5), probing sense and antisense RNA to compare abundance and biochemical size. On the other hand, the authors very clearly show the expression of lncMtDloop in their mouse models by PCR, this image should be included in the study and supplemented by showing at least 7S RNA expression.

Response: *We greatly appreciate the referee's suggestion. Using both sense and antisense primers, we have comparatively analyzed the expression levels of lncMtDloop. Notably, faint PCR band signals and Ct values were observed with the antisense primers, leading us to focus on probing the sense RNA to detect the abundance of lncMtDloop. In addition, the requested experiment has been completed accordingly (Appendix Fig. S5).*

TFAM interaction and recruitment

I would like to thank the authors for clarification of their approach on why they chose to study TFAM as the main interaction partner, compared to other hits. An additional sentence clarifying that other proteins identified had a lower catRAPID score, giving an example, would help in the text.

Response: *We greatly appreciate the referee's suggestion. We have added a sentence to the revised manuscript to clarify that other proteins had lower catRAPIDS scores.*

In Figure 3E, the authors chose to include 12 rRNA - a very abundant and stable mtRNA species - as a control and noticed clear differences in TFAM binding. The graphs should be adapted to show the same scale on the y-axis, so the difference in enrichment becomes directly evident. Here, the enrichment of one other lncRNA, preferable 7S RNA, needs to be included to appreciate lncMtDloop binding compared to another lncRNA, as requested in my previous review.

Response : *We greatly appreciate the referee's suggestion. We have adjusted the graphs to ensure consistency in scale on the y-axis. Additionally, as requested in the previous review, we compared the enrichment of lncMtDloop and 7S RNA in the RIP experiment. Our findings indicate little difference in 7S RNA enrichment.*

The study largely improved by including EMSA assays; however, some questions remain. In the rebuttal letter, the authors show an EMSA titration assay (which should be included in the main study). In lane 2, a strong band shift is already seen, even without the addition of the lncRNAs - how do the authors explain this? In their other EMSA, a band shift is already seen upon overexpression of the antisense lncMtDloop, which is contrary to their other assay. Why is this the case? Please discuss.

Response: *We express our sincere gratitude to the referee for the careful review of our manuscript. In response to the comments, we have meticulously verified and repeated the EMSA assay. Firstly, we apologize for the incorrect labels in the previous EMSA figure (left figure below). The corrected figure is now provided (right figure below). Upon re-examination, we have further confirmed that the signals (red box) observed in lane 2 and lane 5 are noise artifacts. The new data from repeated assays verified this claim, as shown in Appendix Fig S4A. Regarding the shifted bands found under the conditions without the addition of lncMtDloop overexpression or with the overexpression of the antisense of lncMtDloop, as stated in the previous version, the EMSA was performed using lysate extracts from N2a cells overexpressing lncMtDloop. Therefore, even without additional overexpression, endogenous lncMtDloop remains, leading to the appearance of mildly shifted bands. This occurs despite the OE-antisense of lncMtDloop causing a competitive effect on the binding of endogenous lncMtDloop to the labeled probe compared to without its overexpression (Appendix Fig S4A). Additionally, we have included data from a more thorough titration of lncMtDloop in the Appendix (Appendix Fig S4B).*

My previous review clearly asked for in vitro transcription assays to substantiate the conclusions made on a direct role of lncMtDloop on mtDNA transcription (as done in the Zhu et al, 2022 paper), data that has not been included here. However, I understand that establishing a working in vitro system takes time and expertise. The authors should thus very clearly state these limitations of their study - due to lack of direct evidence - in the result and discussion section of the manuscript and more carefully interpret the conclusions drawn throughout the manuscript. Nonetheless, their finding on increased mtDNA transcription, but not mtDNA replication after mtDNA depletion is very interesting and merits further investigation, I strongly advise the authors to address this in future study and highly recommend supporting in vitro studies.

Response: *We greatly appreciate the referee's careful and insightful review, which has significantly inspired and guided our work. We acknowledge the limitations of our study and have addressed these in the revised manuscript. In particular, we recognize the importance of the reviewer's observation regarding the direct role of lncMtDloop on mitochondrial DNA (mtDNA) transcription, as well as the intriguing finding of increased mtDNA transcription without a corresponding increase in mtDNA replication. These insights highlight critical areas for further investigation. We agree that extensive additional research is required to fully elucidate the mechanisms underlying these phenomena. Future studies should focus on exploring the regulatory pathways of lncMtDloop in greater detail, examining its potential interactions with other molecular players involved in mtDNA transcription and replication, and assessing the broader implications of these findings in the context of cellular function and disease. By addressing these points, we aim to contribute to a more comprehensive understanding of the role of lncMtDloop in mitochondrial biology.*

The authors have now included western blot analysis of OXPHOS complexes in their mouse models and observe a relative OXPHOS decline that is rescued upon expression of lncMtDloop. Densitometric quantification is partly difficult, especially upon overexposure of blots. Here, for some reason, the LV control band is continuously smaller compared to the others in the source data, making comparison by intensity difficult. Especially complex IV is very hard to distinguish in the blot shown here, which is overexposed and of poor quality. Is there a less over-exposed version available? Better images are required for densitometry.

Response: *We greatly appreciate the referee's careful review. In response, we have replicated the results using a 10% separation gel and exposed them with precise cutting separation. We have included improved images, showcasing a less over-exposed version, in the revised manuscript. We believe this version presents a higher-quality representation of our findings.*

Furthermore, I was wondering how the authors reconcile their findings with published work showing that mt-mRNA depletion is well-tolerated in vivo and that neurons are generally fine upon TFAM-knockout - they still need to discuss this point made in my previous review.

Response: *We greatly value the referee's careful review and have addressed this point in the revised manuscript. In response to the suggestions, we have followed the studies from the Larsson Lab with interest, which are experts in mtDNA or mitochondria in aging (Lagouge et al., PLoS Genet, 2015; Sorensen, J Neuroscience, 2001; Filograna, FEBS Lett, 2021; Kauppila, Cell Metabolism, 2017). It is noted that the authors mainly focused on the alteration of the heart and liver (Lagouge, 2015). While there was a 50~70% reduction of mtRNA and no pathological reaction observed in KO mice, it is important to highlight that it may be one-sided to conclude that KO mice are healthy without demonstrating pathological sections (HE staining) or physiological and biological indicators, especially considering no data showing relevant alteration (HE staining, behavior test) of the brain or CNS system. Similar observations have been made in AD patients or AD mice models, which may appear healthy compared with the control group without showing apparent pathological alterations in appearance. However, upon detection of relevant indicators (mitochondrial OCR, ECAR; HE staining; behavior test, etc.), pathological alterations may become evident. Notably, related work has shown that approximately 50% reduction of mtDNA can initially lead to Tau phosphorylation acutely (Swerdlow, Journal of Alzheimer's Disease, 2020).*

As mentioned by the reviewer, neurons can tolerate disturbances of mitochondrial dysfunction for a long time in vivo, and neurons are generally fine upon TFAM knockout. Our findings also align with this, as we observed a 30%-50% reduction in brain tissues, similar to the Sorensen group (Sorensen, 2001). Approximately 40% reduction of mtDNA and mtRNA in TFAM KO mice in the first 5 months of life did not lead to overt behavioral disturbances or histological changes. However, at around 5–5.5 months of age, MILON mice started to show signs of disease, followed by death shortly thereafter. The onset of overt disease in MILON mice coincides with the initiation of rapidly progressive neurodegeneration. Our experimental models have reached 12-15 months during progressive neurodegeneration, and the accumulation of chronic factors during neurodegeneration may exacerbate the deregulated impact on mitochondrial homeostasis and physiological status.

References:

- 1. Marie Lagouge., et. al., SLIRP Regulates the Rate of Mitochondrial Protein Synthesis and Protects LRPPRC from Degradation. PLoS Genet, 2015.*
- 2. Sorensen L., et. al., Late-onset corticohippocampal neurodepletion attributable to catastrophic failure of oxidative phosphorylation in MILON mice. J Neurosci, 2001.*
- 3. Roberta Filograna., et. al., Mitochondrial DNA copy number in human disease: the more the better? FEBS Lett, 2021.*
- 4. Kauppila TES., et. al., Mammalian Mitochondria and Aging: An Update. Cell Metab, 2017. Weidling IW., et. al., Mitochondrial DNA Manipulations Affect Tau Oligomerization. J Alzheimers Dis, 2020.*

Minor points:

- The authors should possibly modify their statement on lncMtDloop sizes - are size differences of hundreds of base pairs differences still slight?

Response: *We greatly appreciate the referee's careful review. We acknowledge that there are differences in the sizes of lncMtDloop among different species. To assess conservation, we selected 10 species and compared them using the UCSC Genome Browser. Our analysis revealed high conservation in the corresponding block region across these species. We have accordingly adjusted the statement regarding lncMtDloop sizes in the revised manuscript.*

- Figures 5A and E - could the authors clarify in the figure legends which probes were used for mtDNA and mtRNA?

Response: *We greatly appreciate the referee's careful review. As we mentioned in the figure legends, Figure 5A indicates the relative RNA of mtRNA in the hippocampus and PFC of the rhesus macaque, and Figure 5E indicates the relative RNA of mtRNA in the N2A cell. We conducted RT-qPCR to test the relative RNA level of mtRNA, which primers were shown in the Appendix Table S3. of*

Referee #3:

I acknowledge the effort of the authors to extensively answer the reviewers questions. However, there are still minor points that should be addressed.

- Regarding the role of mitochondrial fission in Fig 5, the authors say they analyzed perinuclear mitochondria/ mitochondria in the soma. I wonder if decreased levels of peripheral fission induced by lncMtDLoop can affect the entry of mitochondria in the axon and anterograde transport of mitochondria, that are highly dependent on peripheral fission. How does it look the distribution of mitochondria in the axon and dendrites after lncMtDLoop expression?

Response: *We greatly appreciate the referee's careful review. Neuronal mitochondrial biogenesis primarily occurs in the soma. Therefore, our analysis mainly focused on relevant parameters (such as length and morphology type) of perinuclear mitochondria in the soma of the neuron. In response to the reviewer's inquiry regarding whether decreased levels of peripheral fission induced by lncMtDLoop can affect the entry of mitochondria into the axon and anterograde transport of mitochondria, we have taken note of relevant studies from the Sheng et al. group (Cell, 2008; Cell Metabolism, 2020; Neuron, 2021; Neuron, 2022). Neurons are particularly susceptible to mitochondrial dysfunction, especially as axons are highly vulnerable to bioenergetic failure. Our initial evaluation primarily focused on the primary events occurring around the soma, with subsequent investigation into impaired mitochondria trafficking to sites within the axon, where high energy demand exists. Moving forward, we aim to conduct further research specifically targeting downstream events within axons induced by lncMtDloop.*

References:

1. Kang JS., et al., Docking of axonal mitochondria by syntaphilin controls their mobility and affects short-term facilitation. *Cell*, 2008.
2. Han Q., et al., Restoring cellular energetics promotes axonal regeneration and functional recovery after spinal cord injury. *Cell Metab*, 2020
3. Chamberlain KA., et al., Oligodendrocytes enhance axonal energy metabolism by deacetylating mitochondrial proteins through transcellular delivery of SIRT2. *Neuron*, 2021.

4. Cheng XT., et al., *Programming axonal mitochondrial maintenance and bioenergetics in neurodegeneration and regeneration. Neuron, 2022.*

- In agreement with Reviewer #1, the IncMtDloop and Mito-ATP5 colocalization is not strong. In fact, this is particularly evidence in neurons, while the other cells show a much clear colocalization. The authors suggested that mtDNA transcripts could have escaped to the cytosol and, I far as I know, both cytosolic mtDNA and mtRNA are known to induce neuroinflammation. So I wonder if IncMtDLoop can have a role in mitochondria-induced neuroinflammation. The authors should check if there are any differences in the cytosolic versus mitochondrial levels of IncMtDLoop in hippocampal WT vs 3xTg neurons.

Response: *Thank you for the referee's careful review and valuable suggestions. It has been established that mtDNA escaping into the cytosol triggers the cGAS-STING pathway. Previous studies suggest that cytosolic mtDNA progressively accumulates during aging and may be associated with Alzheimer's pathogenesis. As part of one of our ongoing projects, our latest preliminary data indicate an increasing trend in the accumulation of escaped mtDNA (dsDNA) in the cytosolic fraction of 3xTg neurons (Figures A and B below). We have observed cytosolic enrichment and a mitochondrial reduction of IncMtDloop expression in 3xTg neurons (Figure C below). Moreover, our preliminary data further show that BV2 cells treated with lipopolysaccharide (LPS) exhibit an increased proportion of mtDNA (dsDNA) and trigger the cGAS-STING pathway (Figure D below). This suggests a potential mechanism by mitigating escaped mtDNA-induced cGAS signaling via IncMtDloop may prevent neuroinflammation in AD brains. Therefore, we are now exploring the role of cytosolic mtDNA escaping from mitochondria on the cGAS-STING pathway in AD neurons. Specifically, we aim to conduct an in-depth study to elucidate if and how IncMtDloop is involved in the escaped mtDNA-mediated cGAS-STING pathway, which may be related to AD neuroinflammation. This will involve a detailed examination of the pathways and molecular interactions involved in this process to uncover novel therapeutic targets for mitigating neuroinflammation in AD.*

- The authors didn't really answer how they decided the cut off for mitochondrial morphology: tubular vs short tubular vs fragmented. $\{ \text{less than or equal to} \} 2 > 2$, seems very arbitrary. The authors should create a cut off based on the statistical quartiles of the aspect ratio of all mitochondria measured.

Response: *We greatly appreciate the referee's careful review. While there is limited literature defining the cut-off for mitochondrial morphology, we have referenced several articles (Sagasti, *Dis Model Mech*, 2014; McCarron, *J Physiol*, 2016; Reusch, *Mitochondrion*, 2016; Tan, *Redox Biol*, 2020) to guide our analysis. These studies have indicated that the mean length/width ratio is predominantly distributed around 2 (Figure below, left). Upon re-analysis of the data based on the statistical quartiles of the aspect ratio of all measured mitochondria, we found that the mean ratio also aligns closely with 2 (Figures below, right). Therefore, we have adopted a ratio of 2 as the cut-off to define the morphology of tubular and short tubular mitochondria. Following fission events, particularly peripheral fission, mitochondria may fragment into smaller parts. We have categorized these morphologies as fragmented types. Fragmented mitochondria typically exhibit smaller sizes, ambiguous membrane boundaries, and fractured notches compared to other morphologies.*

References:

1. Kelley C O'Donnell., et al., *Axon degeneration and PGC-1 α -mediated protection in a zebrafish model of α -synuclein toxicity. Dis Model Mech, 2014.*
2. Chalmers S., et al., *Age decreases mitochondrial motility and increases mitochondrial size in vascular smooth muscle. J Physiol, 2016.*
3. McClatchey PM., et al., *Fully automated software for quantitative measurements of mitochondrial morphology. Mitochondrion, 2016*
4. Tan LX., et al., *Complement activation, lipid metabolism, and mitochondrial injury: Converging pathways in age-related macular degeneration. Redox Biol, 2020*

- Regarding description in the text for AAV injections: authors should add a) for hippocampus injections and b) c) for tail injections when they refer to Appendix Fig S3. Otherwise, it is confusing.

Response: *We greatly appreciate the referee's careful review and valuable suggestion. The requested clarification of Appendix Fig S3 (now Appendix Fig S7) has been added accordingly.*

Dear Dr. Li,

Congratulations on an excellent manuscript, I am pleased to inform you that your manuscript has been accepted for publication in The EMBO Journal. Thank you for your comprehensive response to the referee concerns and for providing detailed source data. It has been a pleasure to work with you to get this to the acceptance stage.

I will begin the final checks on your manuscript before submitting to the publisher next week. Once at the publisher, it will take about three weeks for your manuscript to be published online. As a reminder, the entire review process, including referee concerns and your point-by-point response, will be available to readers.

I will be in touch throughout the final editorial process until publication. In the meantime, I hope you find time to celebrate!

Yours sincerely,

Kelly M Anderson, PhD
Editor, The EMBO Journal
k.anderson@embojournal.org
